

**GMED: Global Marine Environment Datasets for environment visualisation and**
**species distribution modelling**
Zeenatul Basher[1]* David A. Bowden[2], and Mark J. Costello[1]
[1] Institute of Marine Science, The University of Auckland, Auckland 1142, New Zealand
[2] Coasts and Oceans Centre, National Institute of Water and Atmospheric Research (NIWA), Private Bag
14901, Wellington, New Zealand
* Corresponding author:
Zeenatul Basher
Email address: z.basher@auckland.ac.nz



## Abstract

The Global Marine Environment Datasets (GMED) is a compilation of publicly available
climatic, biological and geophysical environmental layers featuring present, past and future
environmental conditions. Marine biologists increasingly utilize geo-spatial techniques with
modelling algorithms to visualize and predict species biodiversity at a global scale. Marine
environmental datasets available for species distribution modelling (SDM) have different
spatial resolutions and are frequently provided in assorted file formats. This makes data
assembly one of the most time-consuming parts of any study using multiple environmental
layers for biogeography visualization or SDM applications. GMED covers the widest available
range of environmental layers from a variety of sources and depths from the surface to the
deepest part of the ocean. It has a uniform spatial extent, high-resolution land mask (to
eliminate land areas in the marine regions), and high spatial resolution (5 arc-minute, c. 9.2 km
near equator). The free public online availability of GMED enables rapid map overlay of
species of interest (e.g. endangered or invasive) against different environmental conditions of
the past, present and the future, and expedites mapping distribution ranges of species using
popular SDM algorithms. GMED can be found at http://gmed.auckland.ac.nz/ (DOI: https://
10.6084/m9.figshare.5937268)



## 1 Introduction

Understanding how species distributions are related to environmental gradients is important for assessing the impacts of, for instance, threats to habitats from species invasions, and climate change (Millennium Ecosystem Assessment, 2005). Because sampled data on species' distributions are spatially biased (Phillips et al., 2009), species distribution models (Anderson et al., 2003), which predict the occurrence of suitable habitat based on correlations between species' records and environmental parameters (Elith and Leathwick, 2009), are used increasingly to predict distributions in un-sampled areas based on environmental variables. SDM's have a wide variety of uses in biogeography, ecology and conservation biology (Elith and Leathwick, 2009). Successful prediction of species ecological niche preference using SDM algorithms depends on both high-quality species occurrence records and related environmental information (Elith and Leathwick, 2009). In contrast to the wide adoption of SDM in terrestrial ecosystem studies, there are relatively fewer studies of marine species (Robinson et al., 2011). Predictions of geographic distributions of marine organisms using SDM include studies on fish (Guinotte et al., 2006; Wiley et al., 2003), coral reefs (Bridge and Guinotte, 2013; Davies and Guinotte, 2011; Rinne et al., 2014; Tittensor et al., 2009; Tong et al., 2013), jellyfish (Bentlage et al., 2013), crabs (Compton et al., 2010), benthic invertebrates (Basher et al., 2014; Basher and Costello, 2016; Compton et al., 2013; Dambach et al., 2012; Reiss et al., 2011; Saeedi et al., 2016), and algae (Downie et al., 2013; Graham et al., 2007; Tyberghein et al., 2012; Verbruggen et al., 2009). Application of SDM in the marine realm were restricted by issues compared with the terrestrial environment are the fewer marine species observation records (Kaschner et al., 2006), extensive spatio-thermal variability characterizing the ocean environment (Franklin and Miller, 2009; Valavanis et al., 2008), and complexities involved in processing environmental data for SDM applications (Tyberghein et al., 2012).

Marine environmental data are derived from direct measurement, remote-sensing, and numerical modelling for a range of variables associated with the ocean surface (e.g. currents, wave height), water column (e.g. temperature, salinity, nutrients), and sea floor (e.g. depth, slope, distance to shore) (Valavanis et al., 2008). Because available marine environmental datasets occur in assorted file formats and differ in their accuracy, and temporal and spatial resolution, it is common for a large portion of time in SDM studies to be spent on assembling compatible environmental data (Tyberghein et al., 2012). Among the commonly available marine environmental datasets, sea surface temperature observations are relatively consistent, accurate, well spatially resolved and have a long global time series. Chlorophyll-a



concentration has similarly good consistency apart from data gaps in the polar-regions, but has
only been available at global scales since 1997. In contrast, most of the deep-sea (i.e., below
surface layers) and less well sampled variables (e.g. dissolved oxygen and nutrient
concentrations) are patchy in their spatial distribution and cannot be measured from satellite
imagery. Generally, data accuracy will be poorer from more remote areas, which have less
primary data. Hence, continuous, global, layers for such variables are predicted from ocean
circulation models and by extrapolation of in situ sample data. Ocean circulation models
generally have relatively coarse resolution, primarily because of computational capacity, and
thus are often inadequate to gather environmental conditions on finer time and spatial scales
(Redfern et al., 2006). However, when available at finer resolution, ocean circulation models
can simulate realistic features and dynamics, such as variability in frontal and eddy structures
and its effect on biogeochemical fields (McGillicuddy et al., 2003).

85           WorldClim (http://www.worldclim.org), a global terrestrial climate environment

dataset is a freely available and widely accessible online repository that has served the need for
terrestrial SDM researchers. Initiatives to establish equivalent marine environment data
repositories include (1) the KGS mapper environmental dataset (Hexacoral project, Fautin and
Buddemeier, 2011), (2) Aquamaps (Kaschner et al., 2008), (3) the human impact on marine
ecosystems layers (Halpern et al., 2008), (4) Bio-Oracle (Tyberghein et al., 2012), and (5)
MARSPEC: Ocean climate layers for marine spatial ecology (Sbrocco and Barber, 2013).
However, except for Bio-Oracle, other datasets have not been widely adopted due to the
complexity of processing the data for modelling applications. Although Bio-Oracle has the
greater number of independent variables among the datasets, it lacks bathymetry and other
ecologically significant layers (e.g. slope) (Table 3). The accuracy and resolution of various
ocean circulation models and survey data are continually increasing, particularly through
assimilation of observations from global ocean observing programmes. Millions of marine
species observation records are available from the Global Biodiversity Information Facility
(GBIF, http://www.gbif.org) and Ocean Biogeographic Information Systems
(OBIS, http://www.iobis.org). A need for easier access to marine species occurrence records
and environmental data prompted the science community to launch the Group on Earth
Observations Biodiversity Observation Network (GEO
BON, https://www.earthobservations.org/geobon.shtml (Andrefouet et al., 2008), which aims
to consolidate biodiversity and earth observation data in a more readily accessible form.

105          Despite these advances, recent experience with developing compatible, comprehensive

environmental layers for use with SDM in the deep-sea (Basher et al., 2014) demonstrated that



considerable work is needed to collate and match environmental data layers from disparate
sources. Based on this experience, we have developed an extensive on-line repository of marine
environmental data layers with consistent resolution and global coverage that are ready to use
in SDM and other spatial analyses. The repository is called the Global Marine Environment
Dataset (GMED). This paper describes the source data and procedures used to generate GMED.

**2 Methods**

Development of the GMED layers followed three main steps: (1) compilation, quality control,
and land-masking of source data; (2) interpolation and projection to generate continuous data
surfaces at uniform resolution; and (3) evaluation of derived data layers against source data
(Fig. 1).

*2.1 Source data*
We compiled data from *in situ* measured, remote-sensed, and modelled datasets for a broad
range of quantitative environmental variables (Table 1). We extracted spatially interpolated i*n*
*situ* measured and remotely sensed data from Aquamaps (Kaschner et al., 2008), KGS mapper
environmental data (Hexacoral project, Fautin and Buddemeier, 2011), NOAA Ocean Color
(Feldman and McClain, 2009), and World Ocean Database 2009 (Boyer et al., 2009). Modelled
datasets were sourced from Bio-Oracle (Tyberghein et al., 2012), paleoclimatic reconstructions
from Peltier (1993) and Paul & Schafer-Neth (2003) and IPCC future climatology layers from
Jungclaus (2006), Tyberghein *et al.* (2012), and Kaschner *et al.*(2013). All compiled datasets
were converted into ESRI grid format before adding into ArcMap workspace for further
processing. Several of the deep-sea variables (e.g., bottom salinity, nutrients) had marine pixels
with 'no data' value. We calculated these missing pixel values using the 'raster calculator' in
ArcGIS, as the average value of the 12 surrounding (ocean) cells. Variable values were then
extracted from each raster grid into a single, global, five arc-minute point geo database. A
uniform land mask was then applied by extracting high-resolution land area from GEBCO 30
arc-second bathymetry (IOC et al., 2003) (Fig. 1).


*2.2 Interpolation and projection*
Methods used to produce smooth interpolated environmental surfaces may combine regression
analyses and distance-based weighted averages (Hartkamp et al., 1999). Such approaches





include: Gaussian weighting filter (Thornton et al., 1997), PRISM method (Daly et al., 2002),
Spline (Hijmans et al., 2005; New et al., 2002) and Inverse Distance Weighting and Kriging
(see Hartkamp et al., 1999, for an overview). We used Inverse Distance Weighting (IDW)
multivariate interpolation (Daly, 2006; Shepard, 1968) to generate environmental surfaces
using the "Spatial Analyst" extension in ArcGIS 10. We selected IDW instead of other
interpolation techniques because it is computationally efficient and its ability to interpolate
equal distance points has been demonstrated in other studies (Dirks et al., 1998; Joseph and
Kang, 2011; Lu and Wong, 2008). IDW interpolates environmental surfaces based on
surrounding measured values that determine the smoothness of the resulting surface
(interpolated values are decreased by distance weighting). In contrast, kriging, the other
commonly used method produces environmental surface based on statistical models and is
more suitable for capturing fine-scale local variability (Gong et al., 2014). IDW interpolation
was used with the default smoothing option in Spatial Analyst (p=2), which assigns the final
interpolated cell values as weighted averages of the values of 12 surrounding points.
Most currently-available datasets are provided in equidistant projections (same distance
from north to south in any pixel of the map). This may be suitable for some mapping
applications, however to measure species richness, abundance and density estimate in a
particular region, an equal–area projected (same area in any pixel of the map) dataset is
preferred (Elith et al., 2010; Tittensor et al., 2009). Following Tyberghein et al (2012), GMED
environmental rasters were interpolated into Behrmann equal area projection as well as WGS84
world geographic equidistant projection. Both equal area and geographically projected data
layers were converted into ASCII grid format before making them available for downloading
from the GMED website (Fig. 1). A spatially cropped version of the dataset is also generated
by cropping the northern extent of the dataset at 70°N because of limited sample data in the
Arctic.

*2.3 Descriptive statistics*
In ArcGIS, the "band statistics" tool was used to measure the standard deviation, standard error,
and coefficient of variation of each dataset. The same tool was used to calculate Pearson
correlation coefficients ($r$) for all pairwise comparisons between pixels in the datasets. To
compare GMED with other available datasets we calculated the range of values for depth,
temperature, salinity, and chlorophyll-*a* annual mean based on a 0.5° resolution grid. We
compared mean values of the above variables with KGS Environment Dataset (Fautin and
Buddemeier, 2011) and AquaMaps dataset (Kaschner et al., 2008).




*2.4 Quality assurance of interpolated data layers*

All of the primary datasets used in the GMED compilation had undergone quality control

checks by the primary data collectors and processors (Table 1). Here, we checked the

interpolation quality of the generated layers to ensure that no errors were introduced during

the re-interpolation process. We tested the interpolation quality for all of the data layers by

extracting interpolated values from 10,000 randomly generated evaluation points over the

global ocean area using the 'extract to points' tool in the ArcGIS 'Spatial Analyst' extension.

Coefficient of variations and standard errors of individual data layers were then calculated

from this point grid using the 'pastecs' package in R v2.15 (R Core Team, 2014) and

compared with values for these statistics derived from the original source layers (Table 1)  to

ensure no significant error was introduced with the interpolation process.

**3 Results**

After initial data cleaning, the primary GMED point grid had ca. 5.7 million data points. Sixty

global marine environment rasters were generated from these point records (Table 1). A

detailed description of the data layers, their sources and interpolated surface images are

available in the supporting materials sections (Table S2 and Appendix A).

*3.1 Comparison with other datasets*

Differences were observed in extreme values by comparison with the source datasets. For

instance, the GMED depth layer has maximum values of 10,415 m, compared to 8,672 m in

KGS Mapper and 8,586 m in AquaMaps (Fig. 2), and 10,977 m in a statistical analysis of

marine bathymetry (Costello et al., 2010; Costello et al., 2015). The sea surface temperature

(mean) layer has values ranging between −1 and 31°C, compared to KGS Mapper (−1.9 to 29.9

°C) and AquaMaps (-1.79 to 29.57 °C ). Maximum values were also higher in GMED than

other two datasets for Salinity (41 versus 40.3 and 40.02 PSS). In contrast, the maximum value

of chlorophyll-a in GMED was between the values of other two datasets (60.3 versus 64.5 and

56.7 mg.m$^{-3}$) (Fig. 3).

*3.2 Interpolation quality validation with source data*

Interpolation error of GMED's environment surface by comparison with the source data layers

was minimal, as assessed by consistent standard errors and coefficients of variation across most



of the datasets when verified using the random evaluation points (See Fig. S1 for details).
Depth, LGM depth, and primary productivity datasets showed higher standard error in the
GMED evaluation data than in the source datasets. These increases were probably due to
downgrading the spatial resolution of the interpolated surface into GMED's standard five arc-
minute resolution from their primary data resolution of 30 arc-second. Visual inspection of the
original source data layers revealed that the Arctic had more data gaps compared to the
Antarctic, which caused interpolation errors to be more visible in the higher latitudes of
northern hemisphere, especially above 70°N latitude (Appendix A Visualizations).

**218  4 Discussion**

GMED has 6 to 12 times higher spatial resolution than most previously available major marine
environment datasets, with the exception of Bio-Oracle, which is at the same resolution.
However, GMED has 30 more data layers than Bio-Oracle (Table 1 and Table 3). GMED
environmental surfaces were also derived from a more diverse set of sources than any other
publicly available data. Applications such as analyses of species' population densities will
benefit from equal-area projected dataset while rapid mapping of species will benefit from
more the commonly-used geographically projected equidistant dataset. The inclusion of depth,
slope, and several deep-sea variables with past and future climatic scenario layers in GMED
will enable researchers to model distributions of species across broad spatial and temporal
scales. We will integrate more data layers with GMED from climatic, anthropogenic variables
and modelled datasets as they become available in the future.

*231  4.1 Comparison with other datasets*

232       Existing marine environment datasets were compiled for specific objectives. For
example, AquaMaps, datasets represented long-term average of temporally varying
environmental variables (Ready et al., 2010). The KGS mapper marine datasets were developed
to enable environmental classification and to understand spatial and temporal patterns in
biogeochemistry and biogeography (Guinotte et al., 2006).  The Bio-Oracle dataset was
developed to facilitate modelling the distribution of shallow water marine species (Tyberghein
et al., 2012). Differences were observed in extreme values of GMED variables by comparison
with the source datasets. These effects were likely the result of the source data of these layers
being at higher spatial resolution than the source data of other datasets. As SDM results tend
to be influenced by correlated environmental factors (Jiménez-Valverde et al., 2009),
depending on the research questions researchers could use the Table S2 to decide on which




variables to use for their study to minimize this confounding correlation effect. GMED
provides the most comprehensive environmental dataset resource to date for support of SDM
applications. Table 3 gives a comparison of strengths and weaknesses of GMED by comparison
with other freely available marine environment datasets.


*4.2 Dataset extent and quality*
The comparatively high spatial resolution of GMED does not indicate that data quality is
necessarily high in all locations. The quality of the interpolated environmental surfaces is,
therefore, spatially variable and depends on local environmental variability and the quality and
density of the underlying raw observation datasets. GMED environmental surfaces may not
capture all the variation that occur at a resolution of 9 km considering the overall low density
of real-time ocean observations for most variables, and thus not capturing locally important
drivers such as fine scale bathymetric or environmental conditions.
The data layers derived from remotely sensed data included only information with the
highest available quality (from Level-3 processed data products, see Hooker and McClain,
2000 for details). However, even here, data gaps exist due to patchy temporal sampling of
ocean colour by MODIS and SeaWiFS sensors, sparse observational networks in the polar
regions (IPCC Climate Change, 2007), clouds, thick aerosols, inter-orbit gaps, sun glint, and
high solar zenith angles (Gregg and Casey, 2007). Filling these data gaps by interpolation
makes them disappear but may lead to unpredictable errors. The overall interpolation error was
small (Fig. S1), and the highest uncertainty (i.e. the highest predicted error) was in regions with
low data coverage at high latitudes in the Arctic, and some regions of Antarctica (Fig. S2)
(Kennedy, 2014). For example, chlorophyll-a, photosynthetically available radiation, and
diffuse attenuation, which are measured at relatively short wavelengths (in the visible
spectrum), cannot be accurately measured during the winter season at high latitudes due to high
solar zenith angles (Gregg and Casey, 2007). Surface temperature data do not suffer from this
effect because they are measured in longer wavelengths (the thermal infrared part of the
spectrum). Errors are also visible in some non-sampled areas in the middle of the oceans,
particularly for the less commonly reported variables e.g. the deep-sea and nutrient variables
(see layer visualization on Appendix A). Although interpolation and extrapolation of data for
pixels with missing data could create bias affecting the quality of interpolation with layers
created using remotely sensed data, our verification data indicates that the GMED layers are
reliable representations of the source data (Fig. S1).



The extent to which missing data could create a problem in analyses depends on the
application. The larger uncertainty in the prediction of areas with missing pixels may be offset
by a stronger gradient of dominant variables. We provide a cropped version (70°N top extent)
of the GMED dataset as well as a full version of dataset covering all latitudinal ranges. We
advise use of the cropped version of the dataset for any modelling exercise; the full extent
dataset should only be used with careful consideration of possible potential model anomalies
in the Polar Regions.
Although there was an overall agreement between all marine datasets in the tropical
and sub-tropical regions, differences shown in interpolated surface near the polar and coastal
areas were still large. This clearly indicates that some uncertainty exists about the true values
of any particular grid cell in these areas. The differences we found likely reflect the difference
between a pure statistical and a more mechanistic expert-driven approach in interpolation.
Future work focusing on model comparison in these geographic areas would be useful because
in our comparison the effects of interpolation method may be confounded with differences in
primary dataset resolution, used climate and depth data sources, and the temporal resolution of
datasets.
Marine species distribution models are susceptible to faulty predictions into land areas
when the underlying environmental data does not have a uniform land area. As we masked the
GMED datasets using land areas extracted from the very high-resolution (30 arc-second, ca.
930 m in equator) GEBCO data, model prediction in coastal areas should minimise such errors.
We made all data available ASCII Grid format, frequently used by common SDM algorithms
(e.g. MaxEnt, Random Forest, GARP). GMED is published in 5 arc-min (c. 9.2 km near
equator) resolution affording, (1) convenience of managing the rasters in common desktop
computing environments, (2) having sufficient resolution to model near-shore environments,
and (3) resolution fine enough to address species distribution questions at a global scale for
implementing management decisions.

**5 Data availability**

Full dataset in individual data layers with most recent updates are always available at:
http://gmed.auckland.ac.nz/
A snapshot associated with this manuscript stored at
DOI: https://doi.org/10.6084/m9.figshare.5937268



**6 Versions**

1.0 Initial public release of GMED

2.0 Six new data layers added to the repository (Aspects, Port Distance, Euphotic Layer

Bottom Depth, Total Suspended Matter, Particulate Organic Carbon, and Particulate

Inorganic Carbon)

**7 Conclusions**

We have compiled a comprehensive collection of 60 high-resolution marine environmental
data rasters, including layers representing the present, the Last Glacial Maximum, and future
climate scenario of year 2100. It is a freely available resource for marine species distribution
modelling and visualization applications. Its spatial resolution is 5 arc-min latitude-longitude,
which approximates to about 9.2 km x 9.2 km at equator. The gridded rasters are available for
download from the GMED website (http://gmed.auckland.ac.nz/). As more data become
available the collection should be expanded. GMED represents significant progress towards
the compilation of global scale marine environment data for users, particularly non-specialists
in such data such as biologists and ecologists. It enables users to rapidly overlay maps of past,
present and future environmental data on the distribution of species, and to use SDM to predict
potential distributions of vulnerable, endangered or invasive species. We welcome any
potential collaboration and contribution of new global data layers to GMED in future from
other researchers.



**Author Contributions**

ZB conceptualized the idea, compiled the data and created the figures. ZB prepared the manuscript with contribution from DB and MC. All authors contributed to the database compilation, analysis and editing of the manuscript.

**Acknowledgements**

The research was funded by the New Zealand Government under the New Zealand International Polar Year-Census of Antarctic Marine Life Project (IPY2007-01), a University of Auckland Doctoral Scholarship, and New Zealand Ministry of Business Innovation and Employment project CO1X1226. We gratefully acknowledge project governance during IPY200701 by the Ministry of Primary Industries Science Team and the Ocean Survey 20/20 CAML Advisory Group (Land Information New Zealand, Ministry of Primary Industries, Antarctica New Zealand, Ministry of Foreign Affairs and Trade, and National Institute of Water and Atmosphere Ltd). This is publication is a contribution to Group on Earth Observations – Biodiversity Observations Network.





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



Table 1. Source and description of data in GMED.

| Layer | Description | Unit | Original Spatial Resolution | Temporal Range | Derivatives | Primary Data Source |
|---|---|---|---|---|---|---|
| **Physical** | | | | | | |
| Depth | Water depth taken from GEBCO 08 Digital Atlas. | m | 30 arc-second | - | Mean | 1 |
| Slope | Slope derived from GEBCO 08 using ArcGIS Spatial Analyst. | degree | 5 arc-min (9.2 km) | - | - | - |
| Aspect (EW) | East/West Aspect of seafloor (sin(aspect in radians)) | radians | 5 arc-min (9.2 km) | - | - | 2, 3 |
| Aspect (NS) | North/South Aspect of seafloor (cos(aspect in radians)) | radians | 5 arc-min (9.2 km) | - | - | 2, 3 |
| Land distance | Distance to the nearest shoreline (water cells only) calculated using Euclidean distance formula in ArcGIS. | Kilometers | 5 arc-min (9.2 km) | - | - | 3 |
| Port Distance | Distance to nearest seaport, calculated using Euclidian distance formula in ArcGIS. | Euclidean distance | 5 arc-min (9.2 km) | - | - | 4 |
| Ice cover | Mean annual ice cover in percent as derived from the National Snow and Ice Data Centre. Missing cell values were interpolated and values for the ice shelves in the Antarctic were set to 1.5. | % (0-1.0) | 0.5° x 0.5° | 1979-2002 | Mean, Summer, Winter | 5 |
| Tide average | Tides, average of maximum amplitude. These tide model results are from a global 0.25° tide model, which assimilated tide estimates derived from the TOPEX/Poseidon altimeter. | m | 0.25° x 0.25° | - | Mean | 6 |
| Wave height | Height of waves in scaled discrete classes as provided by the Original LOICZ Database, for all coastal and oceanic cells. | m | 0.5° x 0.5° | - | Mean | 7 |
| Wind speed | Yearly variations of the surface marine atmosphere over the global oceans. | m·s⁻¹ | 0.5° x 0.5° | 1945-1989 | Mean | 8 |
| Surface current | Monthly average of Zonal velocity (UVEL), meridional velocity (VVEL) values in the ocean surface. | m·s⁻¹ | 0.25° x 0.25° | 2009-2010 | Mean | 9 |
| Euphotic Layer | Depth of the bottom of the Euphotic Layer i.e. | m | 2.5 arc-min (4km) | 1998-2013 | Mean | 10,11,12 |



| Bottom Depth | the depth for which the down-welling irradiance is 1% of its value at the surface. It characterizes the upper layer of the ocean, which can support phytoplankton photosynthesis. It depends on the turbidity of the water. | | | | | |
|---|---|---|---|---|---|---|
| Diffuse attenuation coefficient | The diffuse attenuation coefficient is an indicator of water clarity. It expresses how deeply visible light in the blue to the green region of the spectrum (490 nm) penetrates in to the water column. | $m^{-1}$ | 5 arc-min (9.2 km) | 2002 - 2009 | Mean | 13 |
| Temperature | Sea surface temperature is the temperature of the water at the ocean surface. This parameter indicates the temperature of the topmost meter of the ocean water column. | °C | 5 arc-min (9.2 km) | 2002 - 2009 | Mean, Minimum, Maximum, Range, Summer (May-Oct), Winter (Nov-Apr) | 13 |
| | Temperature of seabed. | °C | 1° x 1° | 1874-2000 | Mean | 14 |
| | Long term monitoring of temperature on multiple depth levels of the water column. | °C | 2° x 2° | 1871-2008 | Mean | 15 |
| Salinity | Salinity indicates the dissolved salt content in the ocean surface. | Parts per thousand | 1° x 1° | 1961-2009 | Mean | 16 |
| | Long term monitoring of Salinity on multiple depth levels of the water column. | Parts per thousand | 2° x 2° | 1871-2008 | Mean | 15 |
| Photosynthetically Active Radiation | Photosynthetically Active Radiation (PAR) indicates the quantum energy flux from the Sun (in the spectral range 400-700 nm) reaching the ocean surface. | Einstein/m²/day | 5 arc-min (9.2 km) | 1997-2009 | Mean | 13 |






**Chemical**

| | | | | | | |
|---|---|---|---|---|---|---|
| Chlorophyll -a | Chlorophyll A concentration indicates the concentration of photosynthetic pigment chlorophyll A (the most common "green" chlorophyll) in oceans. Please note that in shallow water these values may reflect any kind of autotrophic biomass. | mg·m⁻³ | 5 arc-min (9.2 km) | 2002 - 2009 | Mean, Minimum, Maximum, Range | 13 |
| | Chlorophyll-a concentration data consists of satellite measurements of global and regional ocean color data. | mg·m⁻³ | 5 arc-min (9.2 km) | 1997-2006 | Max, Mean, Summer (May-Oct) Max, Winter (Nov-Apr) max | 17 |
| Primary Productivity | Proportion of annual primary production in a cell. See reference for details about the productivity calculation methods. | mgC·m⁻²/day/cell | 5 arc-min (9.2 km) | - | Mean | 18, 19, 20 |
| pH | Measure of acidity in the ocean surface. | - | 1° x 1° | 1910-2007 | Mean | 16 |
| Total Suspended Matter | Total suspended matter concentration. It is a measure of the turbidity of the water. The product is useful typically for coastal waters where inorganic particle dominate over phytoplankton. | g.m⁻³ | 2.5 arc-min (4km) | 2002-2012 | Mean | 10, 11, 21 |

**Nutrients**

| | | | | | | |
|---|---|---|---|---|---|---|
| Calcite | Calcite concentration indicates the concentration of calcite (CaCO3) in oceans. | mol·m⁻³ | 5 arc-min (9.2 km) | 2002 - 2009 | Mean | 13 |
| Nitrate | This surface layer contains both [NO3] and [NO3+NO2] data i.e. mean chemically reactive dissolved inorganic nitrate and nitrate or nitrite. | µmol·l⁻¹ | 1° x 1° | 1922 - 1986 | Mean | 16, 22 |
| | Seabed Nitrate Concentration | µmol·l⁻¹ | 0.5° x 0.5° | 1874-2000 | Mean | 23 |
| Phosphate | Phosphorous Concentration surface and seabed. | µmol·l⁻¹ | 0.5° x 0.5° | 1874-2000 | Mean | 23 |
| Silicate | This variable indicates the concentration of silicate or ortho-silicic acid [Si(OH)4] in the ocean surface. | µmol·l⁻¹ | 1° x 1° | 1930 - 1986 | Mean | 16 |



| | | | | | | |
|---|---|---|---|---|---|---|
| | Seabed Silicate Concentration. | µmol·l⁻¹ | 0.5° x 0.5° | 1874-2000 | Mean | 23 |
| Dissolved Oxygen | Dissolved oxygen concentration [O2] in the surface. | ml·l⁻¹ | 1° x 1° | 1898 - 2009 | Mean | 16 |
| | Seabed Dissolved Oxygen Concentration | ml·l⁻¹ | 0.5° x 0.5° | 1874-2000 | Mean | 24 |
| Saturated Oxygen | Amount of dissolved oxygen as a percentage of maximum potential oxygen amount that could be present for the given temperature and salinity at standard atmospheric pressure (760 mmHg) (i.e., sea level). | ml·l⁻¹ | 0.5° x 0.5° | 1874-2000 | Mean | 24 |
| Utilized Oxygen | Apparent oxygen utilization (AOU) in ml/l - oxygen saturation concentration minus measured dissolved oxygen concentration. Both for surface and seabed. | ml·l⁻¹ | 0.5° x 0.5° | 1874-2000 | Mean | 16 |
| POC | Particulate Organic Carbon is an important component in the carbon cycle and serves as a primary food sources for aquatic food webs. | mg.m⁻³ | 2.5 arc-min (4km) | 1998-2013 | Mean | 10, 11, 25 |
| PIC | Particulate Inorganic Carbon or suspended calcium carbonate concentration | mg.m⁻³ | 2.5 arc-min (4km) | 1998-2013 | Mean | 10,11,26, 27 |
| **Past** | | | | | | |
| Last Glacial Maxima Depth | Water depth calculated from GEBCO 08 (using formula current depth-130 m; the average depth decrease mentioned in literature). | m | 30 arc-second | - | Mean | 1, 28 |
| Last Glacial Maxima Temperature | Sea surface temperature during last glacial maxima (22 thousand years ago) | °C | 1° x 1° | 19-22 cal.KYrBP | Mean | 29 |
| Last Glacial Maxima Salinity | Sea surface salinity during last glacial maxima (22 thousand years ago) | Parts per thousand | 1° x 1° | 19-22 cal.KYrBP | Mean | 29 |
| Last Glacial Maxima Ice Thickness | Thickness of ice sheets during last glacial maxima (22 thousand years ago) | km | 1° x 1° | 19-22 cal.KYrBP | Mean | 30 |




| **Future** | | | | | | |
|---|---|---|---|---|---|---|
| Temperature at 2100 | Future grids of monthly mean sea surface temperature, A1B (720 ppm stabilization) scenario. | °C | 1.25° x 1.25° | 2087–2096 | Mean | 31 |
| | Predicted seabed temperature for year 2100. | °C | 0.5° x 0.5° | 2090-2099 | Mean | 32 |
| Salinity at 2100 | Future grid of average monthly mean sea surface salinity | Parts per thousand | 2.75°x 3.75° | 2087–2096 | Mean | 31 |
| | Predicted seabed salinity for year 2100. | Parts per thousand | 0.5° x 0.5° | 2090-2099 | Mean | 32 |
| Primary productivity at 2100 | Predicted primary productivity for year 2100. | mgC·m⁻²·day⁻¹ | 0.5° x 0.5° | 2090-2099 | Mean | 32 |
| Ice Concentration at 2100 | Predicted ice cover (area proportion) for year 2100. | % (0-1) | 0.5° x 0.5° | 2090-2099 | Mean | 32 |

1. (IOC et al., 2003) ; 2.(Becker et al., 2009) ; 3. (Sbrocco and Barber, 2013); 4. (NGIA, 2014); 5. U.S. National
Snow and Ice Data Centre; (Cavalieri et al., 2003); 6. (Stewart, 2000); 7. KGS (Fautin and Buddemeier, 2011);
8. (Da Silva et al., 1994); 9. NASA JPL Laboratory; 10.(Fanton d'Andon et al., 2009); 11. (Maritorena et al.,
2010) ; 12. (Morel et al., 2007); 13. (Feldman and McClain, 2010); 14. (Stephens et al., 2002); 15. 20th Century
Reanalysis V2 data provided by the NOAA/OAR/ESRL PSD, Boulder, Colorado, USA; 16. (Boyer et al., 2009)
17. (Feldman and McClain, 2006); 18.(Bouvet et al., 2002); 19. (Hoepffner et al., 1999); 20. (Longhurst et al.,
1995); 21. (Doerffer and Schiller, 2007); 22. NOAA/NGDC Paleoclimatology Program, Boulder CO, USA. ; 23.
(Saving, 2006); 24. (Conkright et al., 2002); 25. (Stramski et al., 2008); 26. (Balch et al., 2005); 27. (Gordon et
al., 2001); 28. (Bintanja et al., 2005); 29. (Paul and Schäfer-Neth, 2003); 30. (Peltier, 1993); 31. Based on IPCC
(WCRP CMIP3) multi-model database (http://esg.llnl.gov:8080/index.jsp).UKMO-HadCM3 model. 32. IPSL
model, A2 scenario (http://icmc.ipsl.fr/)





Table 2. Descriptive statistics for the GMED environmental layers. All values are in annual means
and refer the ocean surface unless noted otherwise (see Table 1 for detailed layer descriptions).

| Layers | Minimum | Maximum | Mean | Std. Deviation | Std. Error | Co. Variation |
|---|---|---|---|---|---|---|
| **Physical** | | | | | | |
| Depth | -10293.65 | 0.00 | -3440.20 | 1738.53 | 0.72 | -0.51 |
| Slope | 0.00 | 21.65 | 0.98 | 1.22 | 0.00 | 1.24 |
| Aspect (East-West) | -98.94 | 99.94 | -0.03 | 34.27 | 0.01 | -1112.77 |
| Aspect (North-South) | -99.34 | 100.00 | 3.00 | 41.93 | 0.02 | 14.00 |
| Land Distance | 1.00 | 2774.45 | 665.51 | 554.33 | 0.23 | 0.83 |
| Port Distance | 0.00 | 64.16 | 15.63 | 12.36 | 0.01 | 0.79 |
| Ice Cover (Annual) | 0.00 | 1.50 | 0.12 | 0.27 | 0.00 | 2.18 |
| Ice Cover (May-Oct) | 0.00 | 1.50 | 0.12 | 0.28 | 0.00 | 2.29 |
| Ice Cover (Nov-Apr) | 0.00 | 1.50 | 0.11 | 0.28 | 0.00 | 2.56 |
| Wave Height | 0.00 | 7.00 | 0.28 | 0.99 | 0.00 | 3.51 |
| Wind Speed | 0.00 | 12.07 | 7.27 | 1.96 | 0.00 | 0.27 |
| Tide average | 0.00 | 6.40 | 0.46 | 0.45 | 0.00 | 0.97 |
| Current | -0.93 | 1.00 | 0.00 | 0.07 | 0.00 | 16.16 |
| Euphotic Layer Bottom Depth | 7.38 | 142.40 | 72.05 | 23.87 | 0.01 | 0.33 |
| Diffuse Attenuation Coefficient | 0.02 | 0.90 | 0.06 | 0.04 | 0.00 | 0.79 |
| Temperature | -1.00 | 31.54 | 14.40 | 10.94 | 0.00 | 0.76 |
| Temperature Maximum | -1.00 | 35.19 | 16.82 | 11.18 | 0.00 | 0.66 |
| Temperature Minimum | -2.00 | 30.76 | 12.47 | 10.68 | 0.00 | 0.86 |
| Temperature Range | 0.00 | 27.81 | 4.06 | 3.02 | 0.00 | 0.74 |
| Temperature (May-Oct) | -2.10 | 30.72 | 14.44 | 11.33 | 0.00 | 0.78 |
| Temperature (Nov-Apr) | -2.10 | 30.73 | 14.40 | 11.12 | 0.00 | 0.77 |
| Water Column Temperature | -2.30 | 26.03 | 5.55 | 3.63 | 0.00 | 0.65 |
| Seabed Temperature | -2.08 | 29.46 | 1.96 | 3.86 | 0.00 | 1.97 |
| Salinity | 0.00 | 41.00 | 33.60 | 2.50 | 0.00 | 0.07 |
| Water Column Salinity | 6.36 | 40.62 | 34.52 | 1.91 | 0.00 | 0.06 |
| Photosynthetically Active Radiation | 0.00 | 64.82 | 34.13 | 9.06 | 0.00 | 0.27 |
| | | | | | | |
| **Chemical** | | | | | | |
| Chlorophyll-a | 0.00 | 60.38 | 0.19 | 1.31 | 0.00 | 6.94 |
| Chlorophyll-a Max | 0.00 | 64.00 | 0.47 | 2.23 | 0.00 | 4.75 |
| Chlorophyll-a Min | 0.00 | 57.80 | 0.08 | 0.82 | 0.00 | 10.77 |
| Chlorophyll-a Range | 0.00 | 62.16 | 0.33 | 1.67 | 0.00 | 5.01 |
| Chlorophyll-a (May-Oct) Maximum | 0.03 | 64.57 | 0.67 | 2.08 | 0.00 | 3.12 |
| Chlorophyll-a (Nov-Apr) Maximum | 0.02 | 64.57 | 0.42 | 1.31 | 0.00 | 3.16 |
| Primary Productivity | 0.00 | 4875.00 | 370.03 | 277.80 | 0.11 | 0.75 |
| pH | 6.73 | 8.62 | 8.19 | 0.06 | 0.00 | 0.01 |
| Total Suspended Matter | 0.03 | 48.49 | 0.93 | 2.37 | 0.00 | 2.54 |



**Nutrient**

| | | | | | | |
|---|---|---|---|---|---|---|
| Calcite | 0.00 | 9.00 | 2.70 | 3.14 | 0.00 | 1.17 |
| Nitrate | 0.00 | 45.96 | 5.23 | 5.91 | 0.00 | 1.13 |
| Seabed Nitrate | 0.00 | 55.78 | 28.58 | 9.85 | 0.00 | 0.34 |
| Phosphate | 0.00 | 2.43 | 0.65 | 0.59 | 0.00 | 0.91 |
| Seabed Phosphate | 0.00 | 4.50 | 2.01 | 0.65 | 0.00 | 0.32 |
| Silicate | 0.00 | 69.00 | 9.59 | 13.26 | 0.01 | 1.38 |
| Seabed Silicate | 0.32 | 267.50 | 98.41 | 52.51 | 0.02 | 0.53 |
| Dissolved $O_2$ | 2.00 | 9.86 | 5.54 | 1.45 | 0.00 | 0.26 |
| Seabed Dissolved $O_2$ | 0.00 | 10.19 | 4.82 | 1.27 | 0.00 | 0.26 |
| Saturated $O_2$ | 76.05 | 113.11 | 100.08 | 3.25 | 0.00 | 0.03 |
| Seabed Utilized $O_2$ | -2.40 | 7.69 | 2.90 | 1.21 | 0.00 | 0.42 |
| Particulate Organic Carbon | 18.49 | 12898.87 | 89.23 | 118.74 | 0.05 | 1.33 |
| Particulate In-organic Carbon | 0.00 | 10808.54 | 142.70 | 212.35 | 0.09 | 1.49 |

**Past**

| | | | | | | |
|---|---|---|---|---|---|---|
| Depth | -10411.84 | 0.49 | -3836.29 | 1571.24 | 0.68 | -0.41 |
| Temperature | -1.56 | 28.59 | 14.76 | 10.47 | 0.00 | 0.71 |
| Salinity | 4.65 | 41.32 | 35.63 | 1.75 | 0.00 | 0.05 |
| Ice Thickness | 0.00 | 4735.79 | 31.25 | 262.76 | 0.11 | 8.41 |

**Future**

| | | | | | | |
|---|---|---|---|---|---|---|
| Temperature (A1B Scenario) | -1.61 | 35.05 | 18.04 | 10.91 | 0.00 | 0.60 |
| Temperature (A2 Scenario) | -2.19 | 31.91 | 17.58 | 11.12 | 0.00 | 0.63 |
| Seabed Temp | -2.08 | 31.33 | 2.43 | 4.25 | 0.00 | 1.75 |
| Salinity (A1B Scenario) | 3.37 | 40.05 | 34.37 | 1.99 | 0.00 | 0.06 |
| Salinity (A2 Scenario) | 3.37 | 40.05 | 34.37 | 1.99 | 0.00 | 0.06 |
| Seabed Salinity | 3.38 | 41.07 | 34.60 | 1.44 | 0.00 | 0.04 |
| Primary Productivity | 0.00 | 5004.00 | 354.76 | 277.07 | 0.12 | 0.78 |
| Ice Concentration | 0.00 | 1.50 | 0.05 | 0.16 | 0.00 | 3.16 |






Table 3. Comparison of features of freely-available online marine environment datasets. √ = Present,
× = Absent.

| | AQUAMAPS[1] | KGS[2] | HALPERN[3] | MARSPEC[4] | BIO-ORACLE[5] | GMED |
|---|---|---|---|---|---|---|
| **Resolution** | | | | | | |
| arc minute | 30' | 15-30' | 0.5' | 0.5'-10' | 5' | 5' |
| ca. km | 55 | 22-55 | 1 | 1-20 | 9 | 9 |
| **Uniform file format** | √ | √ | √ | √ | √ | √ |
| **Uniform land area mask** | × | √ | × | √ | √ | √ |
| **GIS-ready Format** (ASCII Grid or Raster) | × | × | √ | √ | √ | √ |
| **Common geographic extent** | √ | × | × | √ | √ | √ |
| **Suitable for coastal studies** | × | × | √ | √ | √ | √ |
| **High Resolution Land Mask** | × | × | × | √ | × | √ |
| **Bathymetry** | √ | √ | × | √ | × | √ |
| **Deep-Sea datasets** | √ | √ | × | × | × | √ |
| **Equal-area grids available** | × | × | × | × | √ | √ |
| **Future climate scenario** | √ | × | × | × | √ | √ |
| **Past climate condition** | × | × | × | √ | × | √ |
| **Descriptive statistics of dataset** | × | × | × | × | × | √ |
| **Individual dataset download option** | × | × | × | × | × | √ |

[1] AquaMaps (Kaschner et al., 2008), [2] KGS Hexacoral Project (Fautin and Buddemeier, 2011), [3] Global Map of
Human Impact on Marine Ecosystems (Halpern et al., 2008), [4] MARSPEC Ocean Climate Layers for Marine
Spatial Ecology (Sbrocco et al., 2013), [5]Bio-Oracle Marine SDM Raster (Tyberghein et al., 2012)

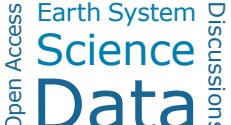


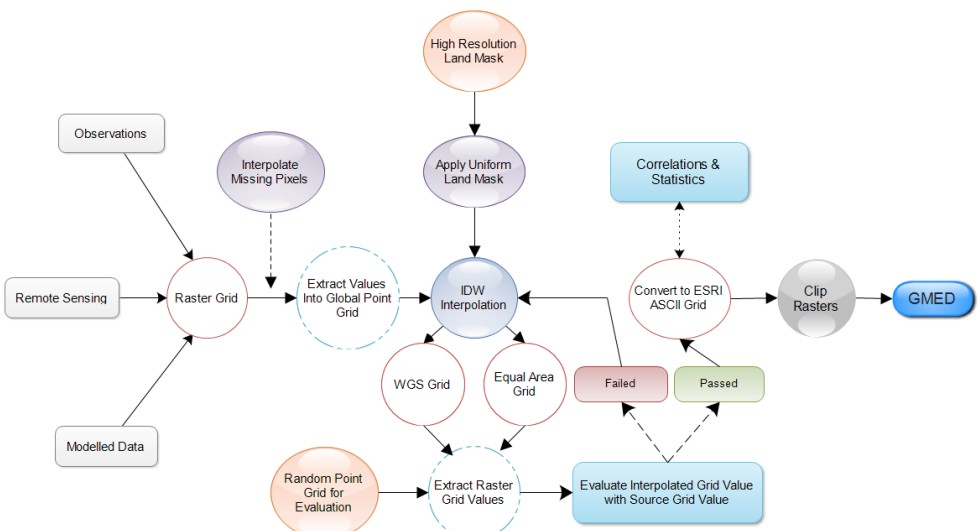

Figure 1. Data processing steps used to produce GMED.

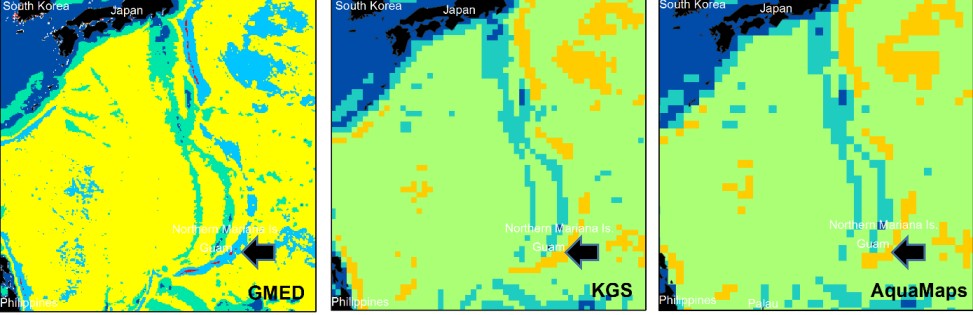

Figure 2. Comparison of Depth layers in GMED (left), KGS Mapper (middle) and AquaMaps (right).
The Mariana Trench near the east coast of Japan is more visible (black arrow) in GMED but barely
visible in both KGS Mapper and AquaMaps dataset.

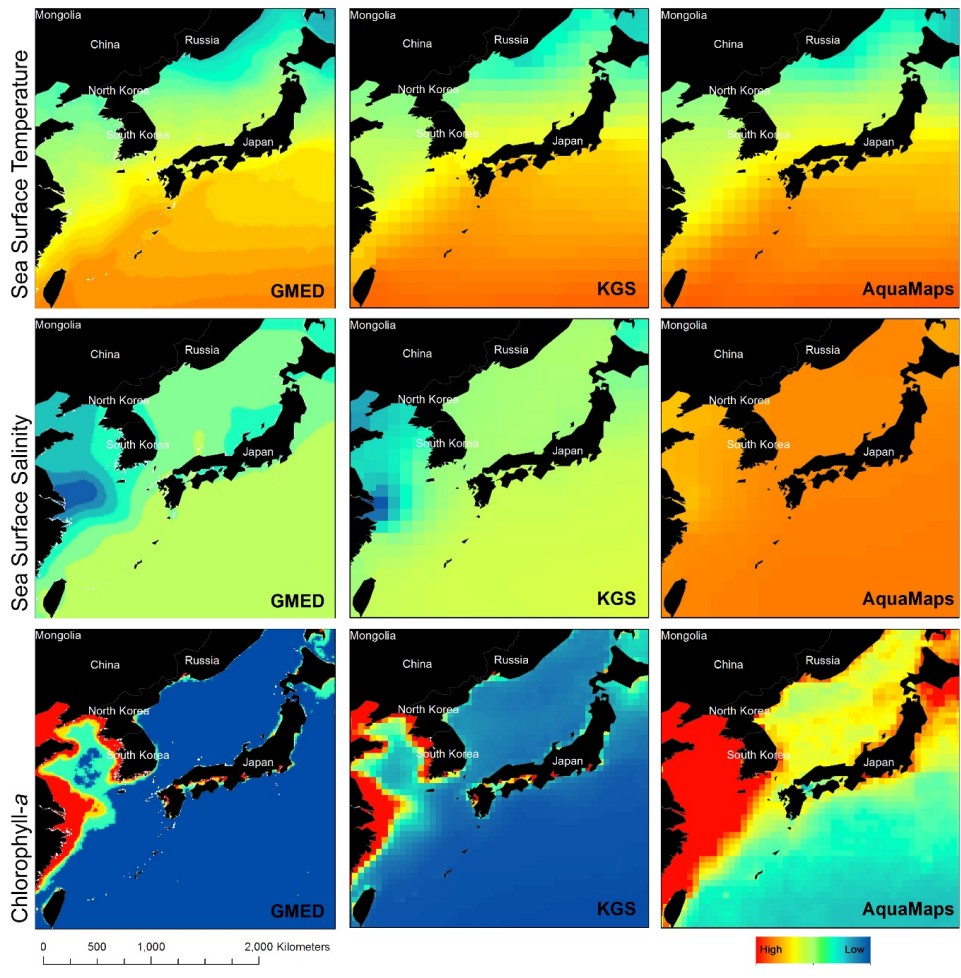

Figure 3. Comparison of mean surface temperature, salinity and chlorophyll-a of GMED with the
KGS Mapper and AquaMaps dataset. Data range high (red) to low (blue).



## Appendix A: Visualization of GMED Data Layers

### Physical

Figure A1. Depth

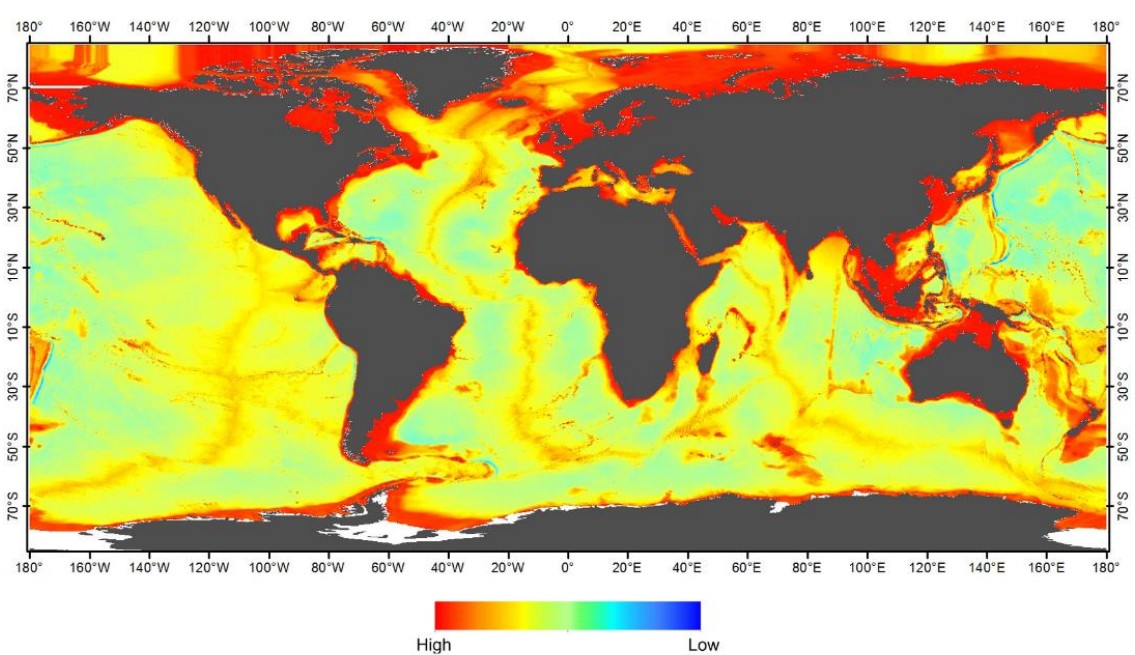

Figure A2. Slope

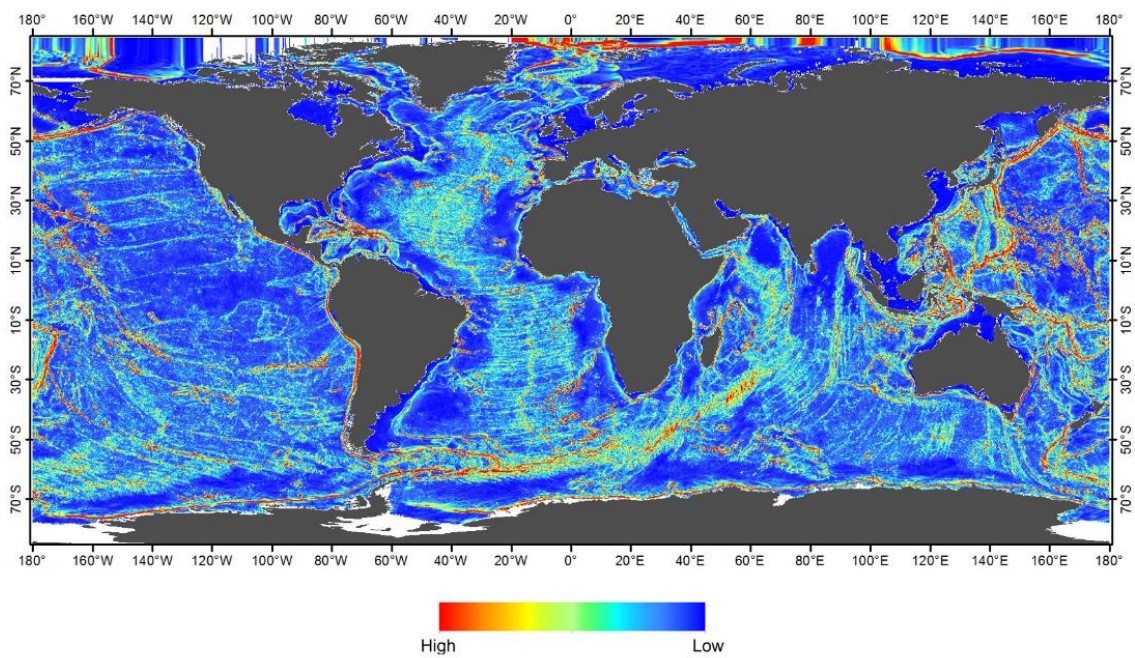



Figure A3. Aspect (East-West)

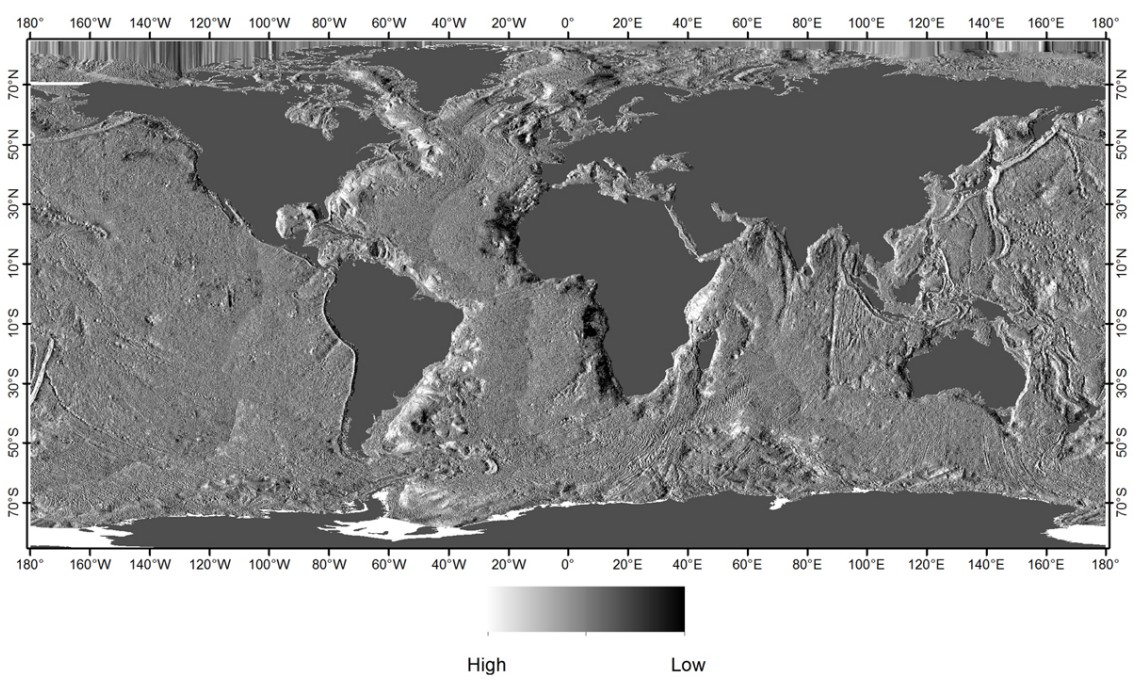

High                    Low

Figure A4. Aspect (North-South)

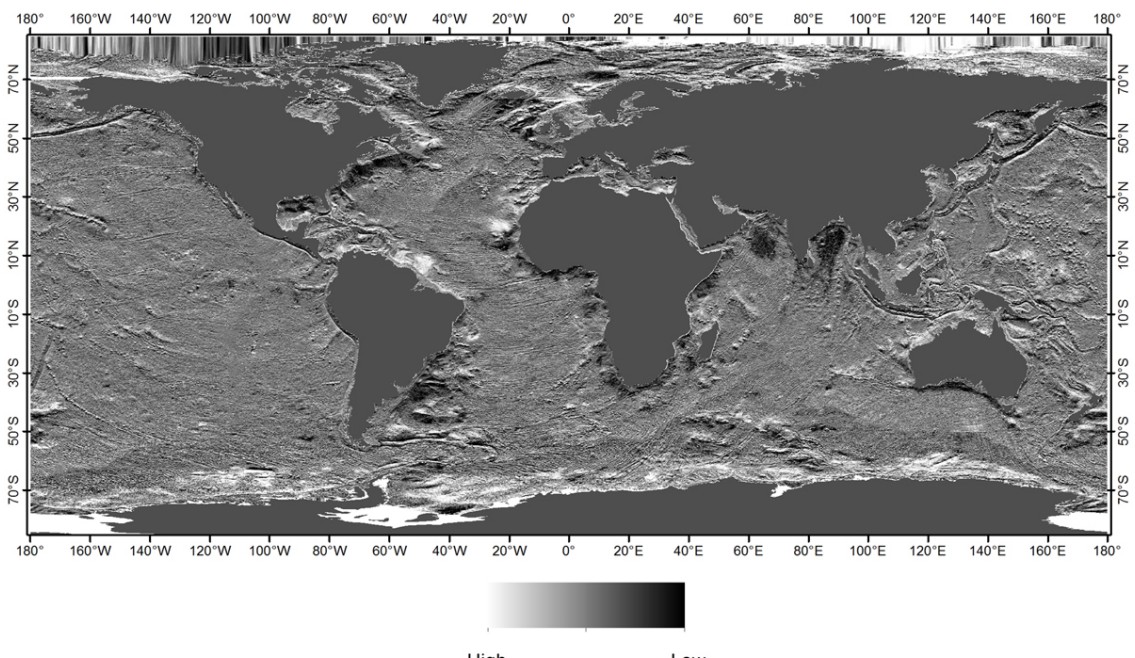

High                    Low



Figure A5. Land Distance

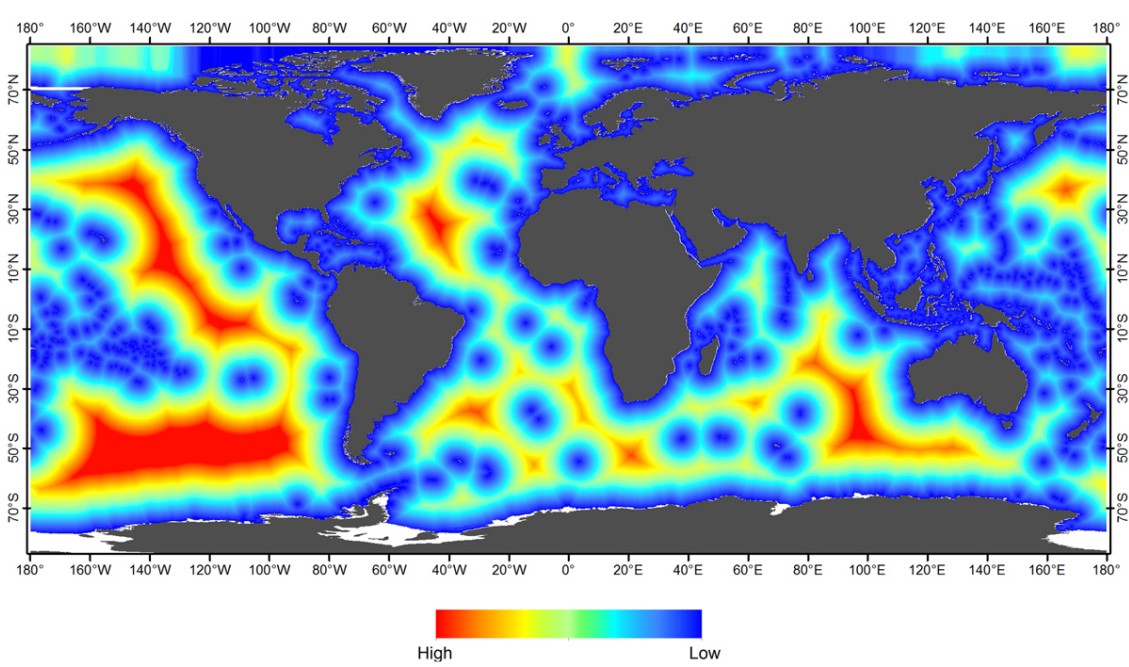

Figure A6. Port Distance



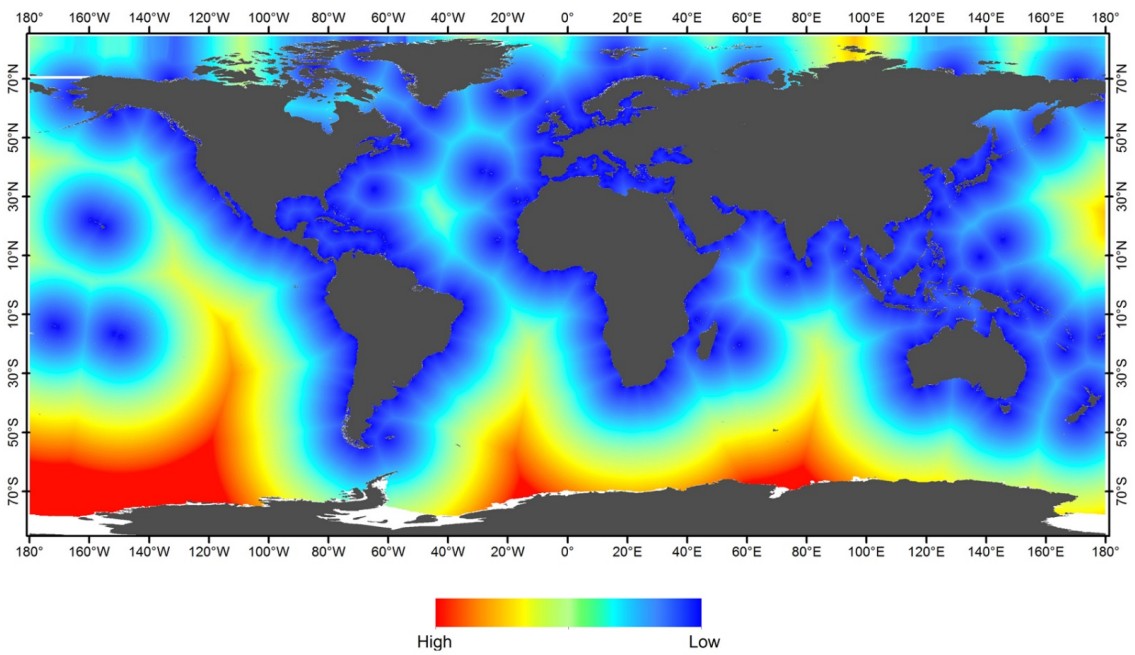

Figure A7. Ice cover  (Annual Mean)

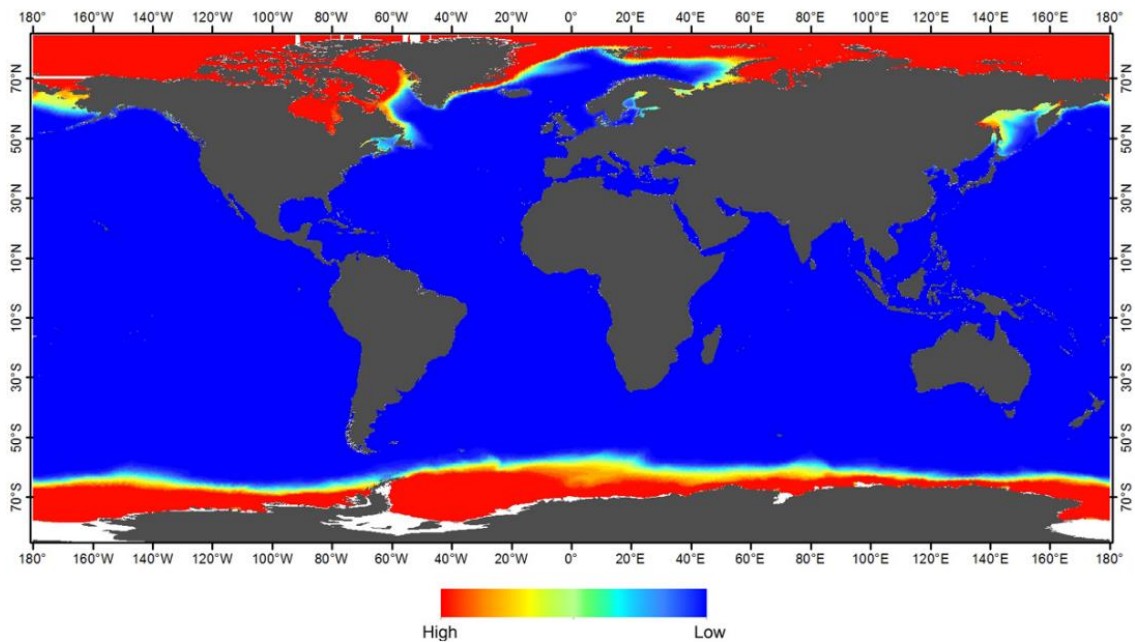

Figure A8. Ice Cover(May-Oct)



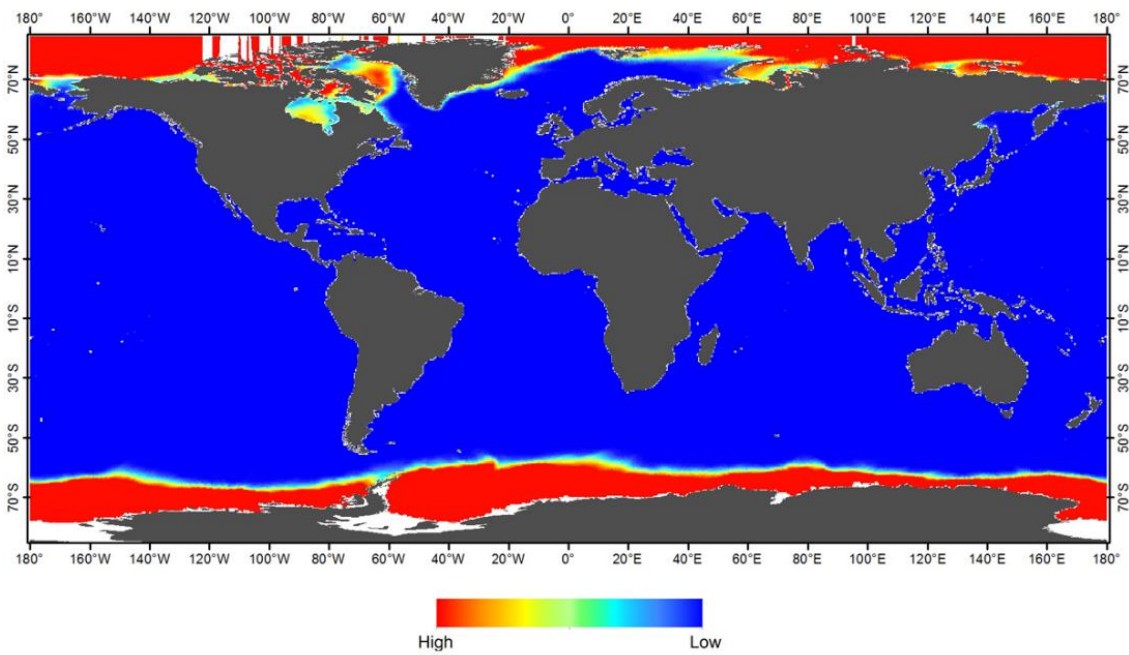



Figure A9. Ice Cover (Nov- Apr)

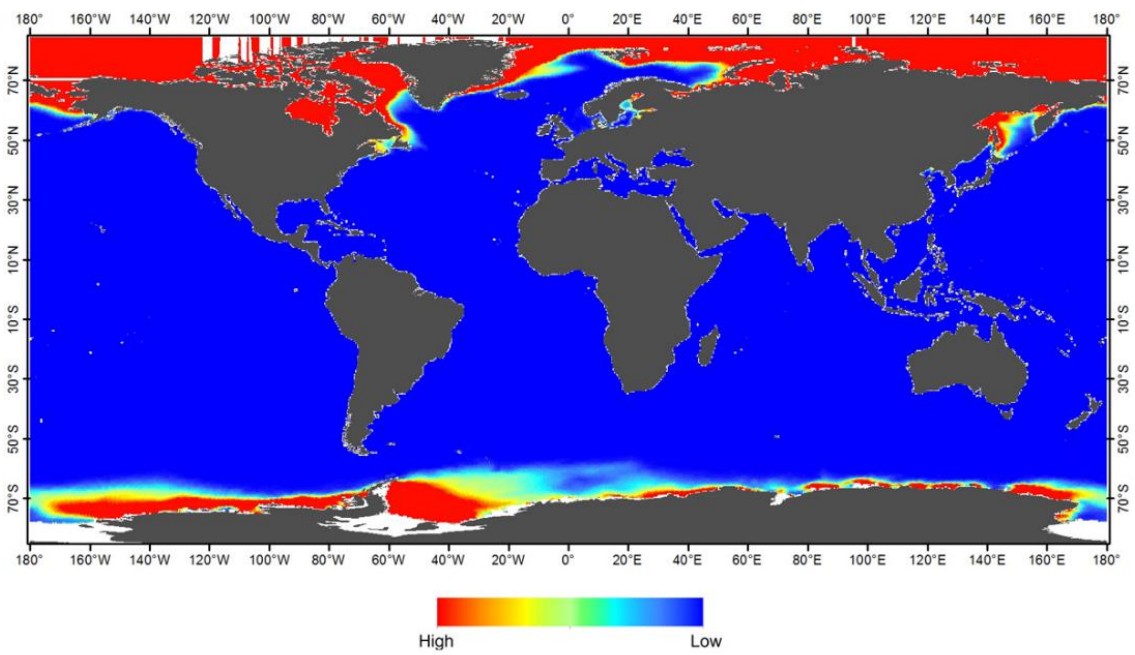

Figure A10. Wave Height

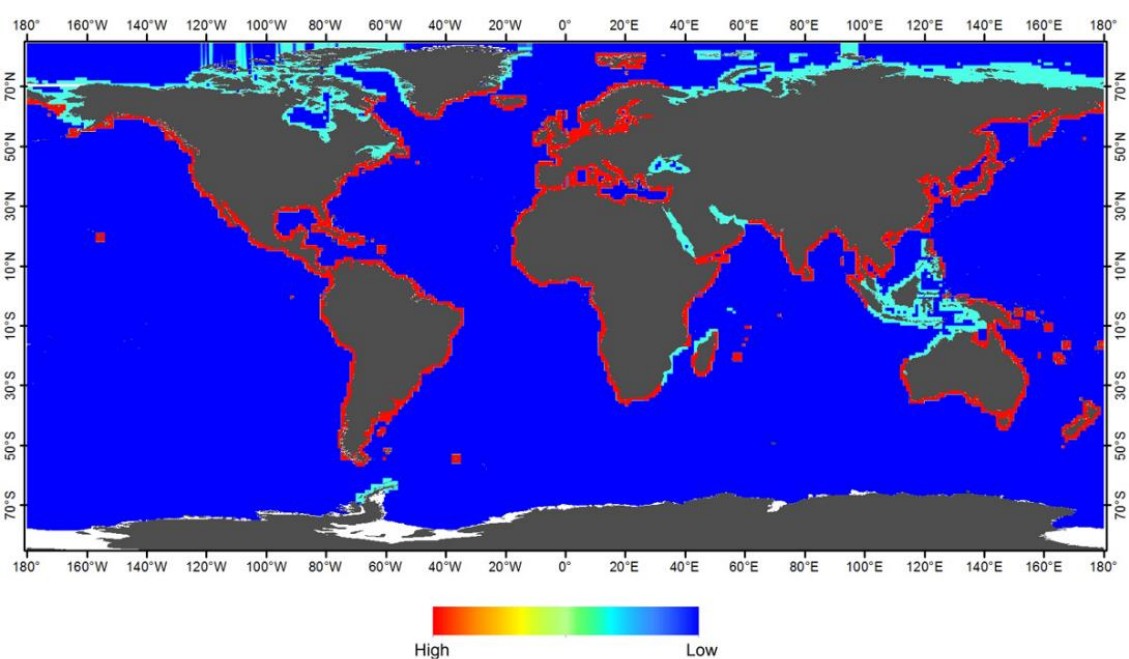





Figure A11. Wind Speed

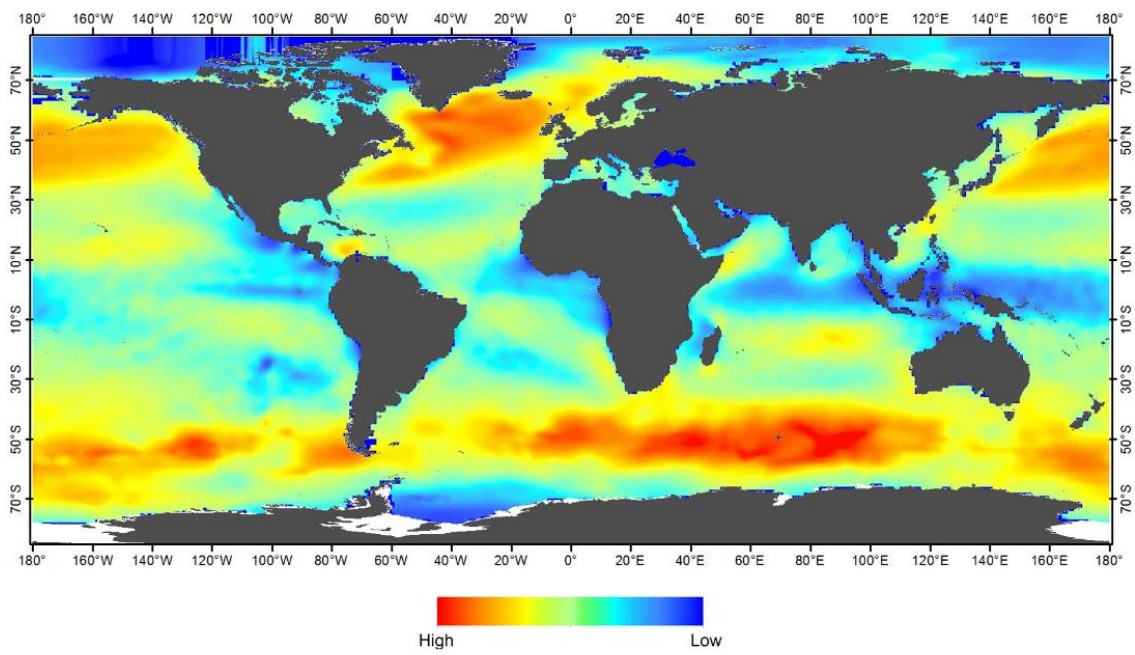

Figure A12. Tide average

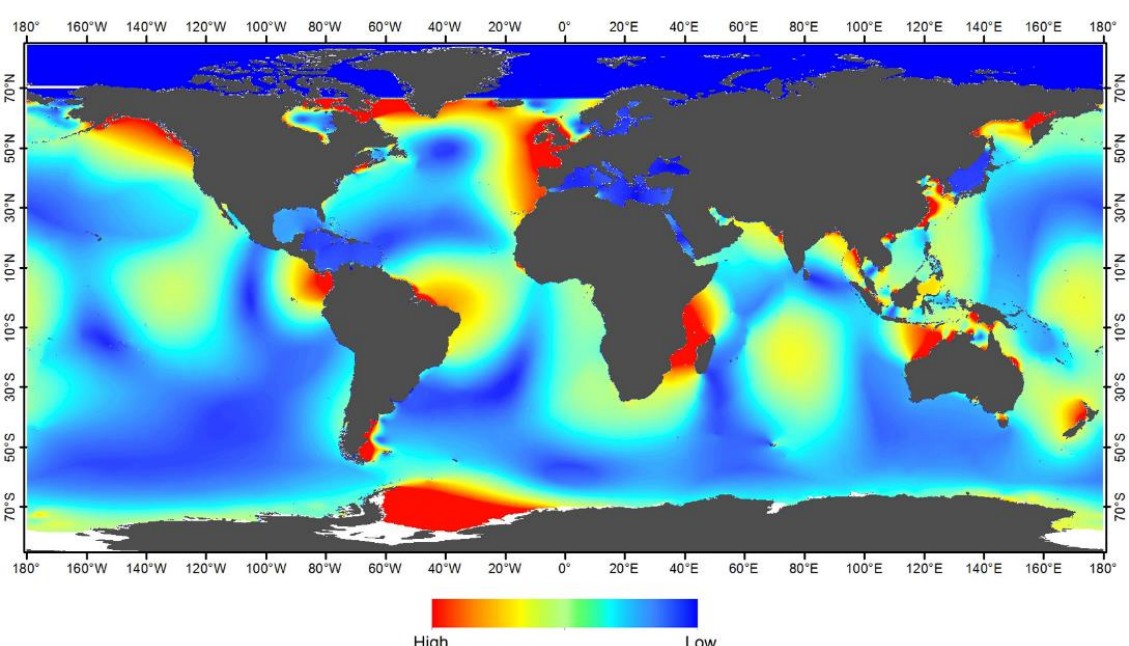

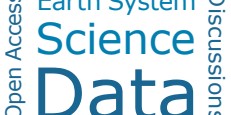

Figure A13. Surface Current

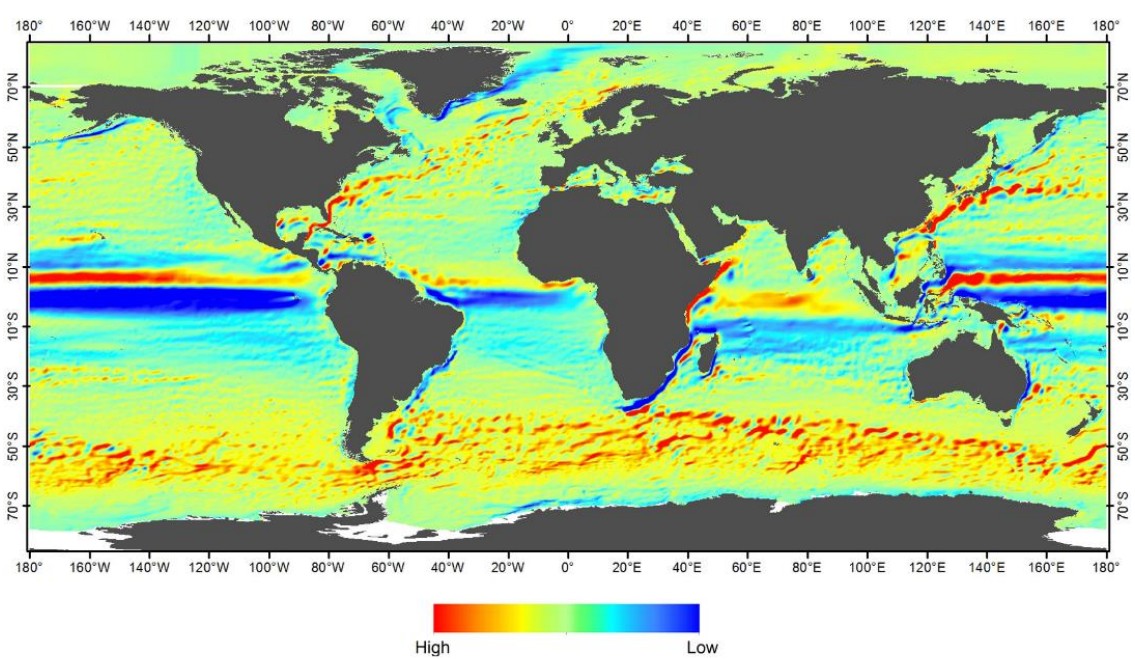

Figure A14. Euphotic Layer Bottom Depth

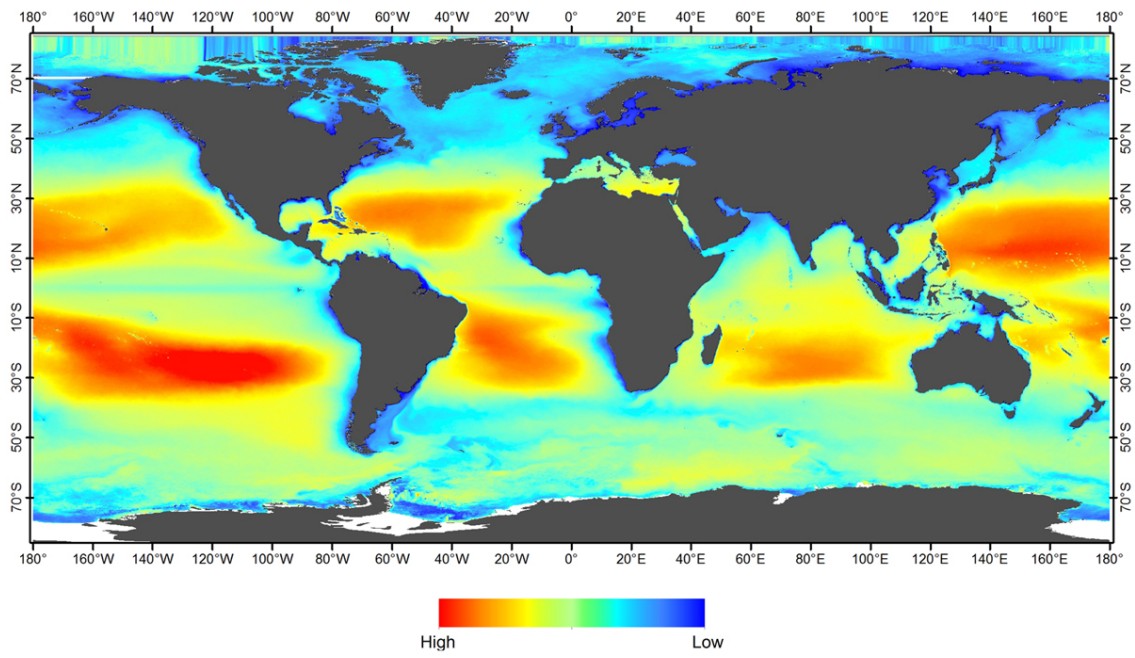



Figure A15. Diffuse Attenuation Coefficient

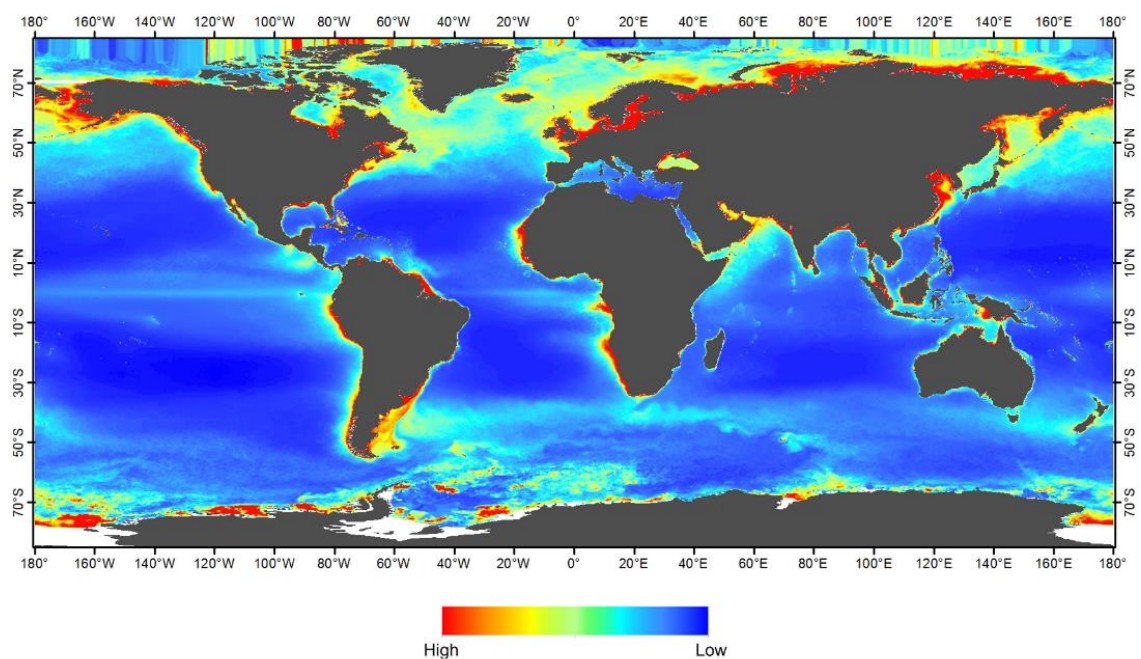

Figure A16. Sea Surface Temperature Mean



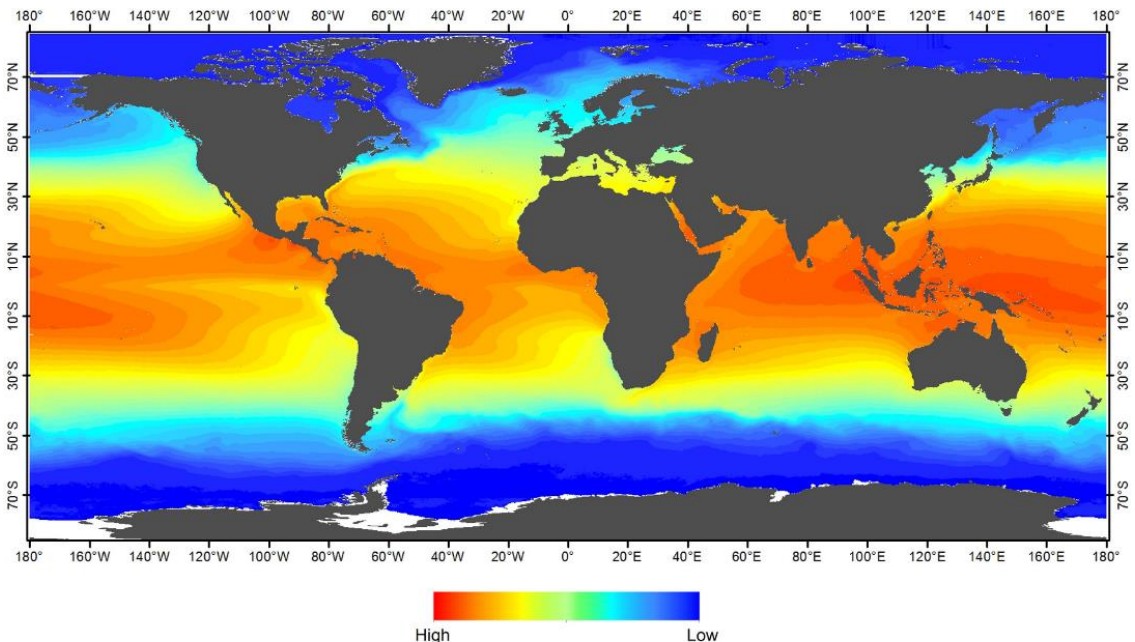

Figure A17. Sea Surface Temperature Maximum

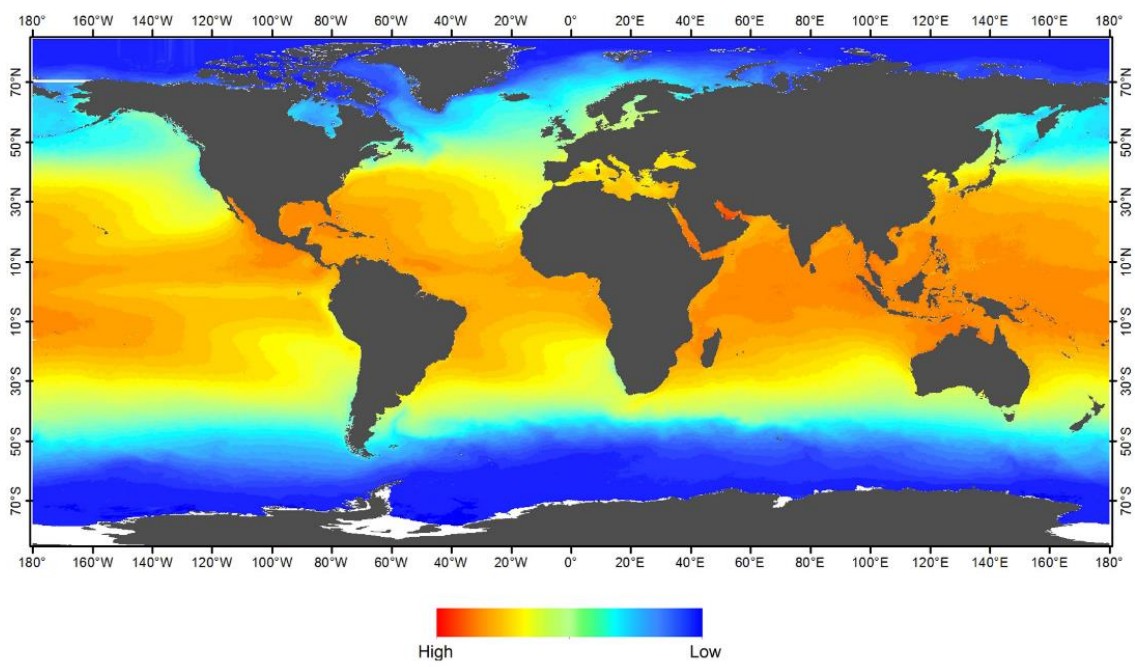

Figure A18. Sea Surface Temperature Minimum



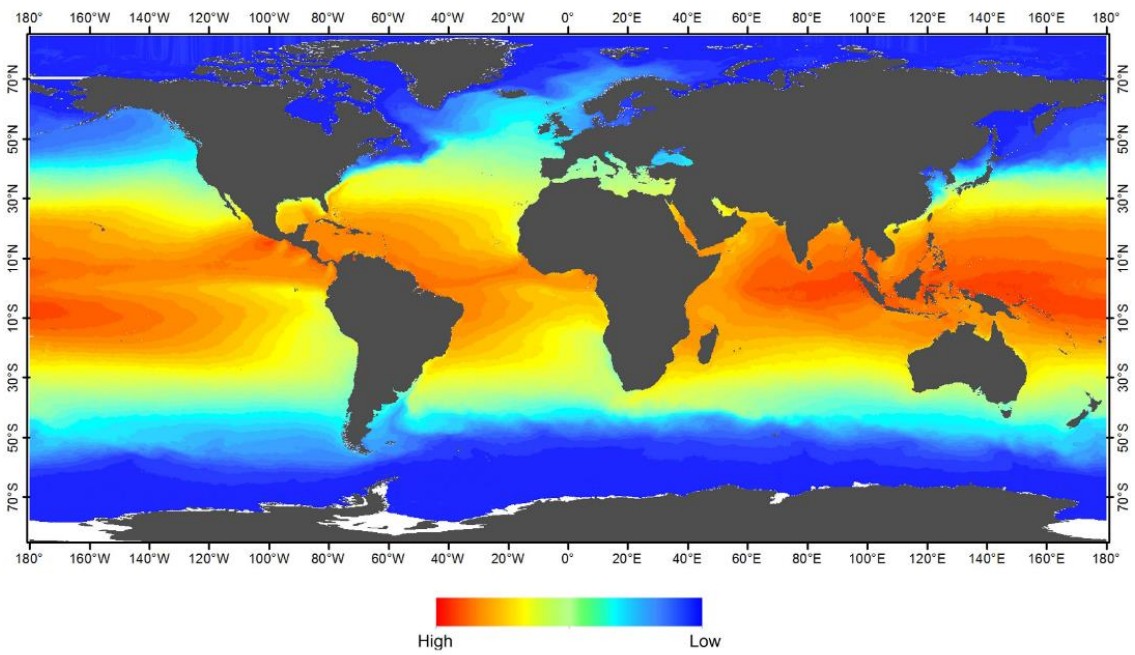

Figure A19. Sea Surface Temperature Range

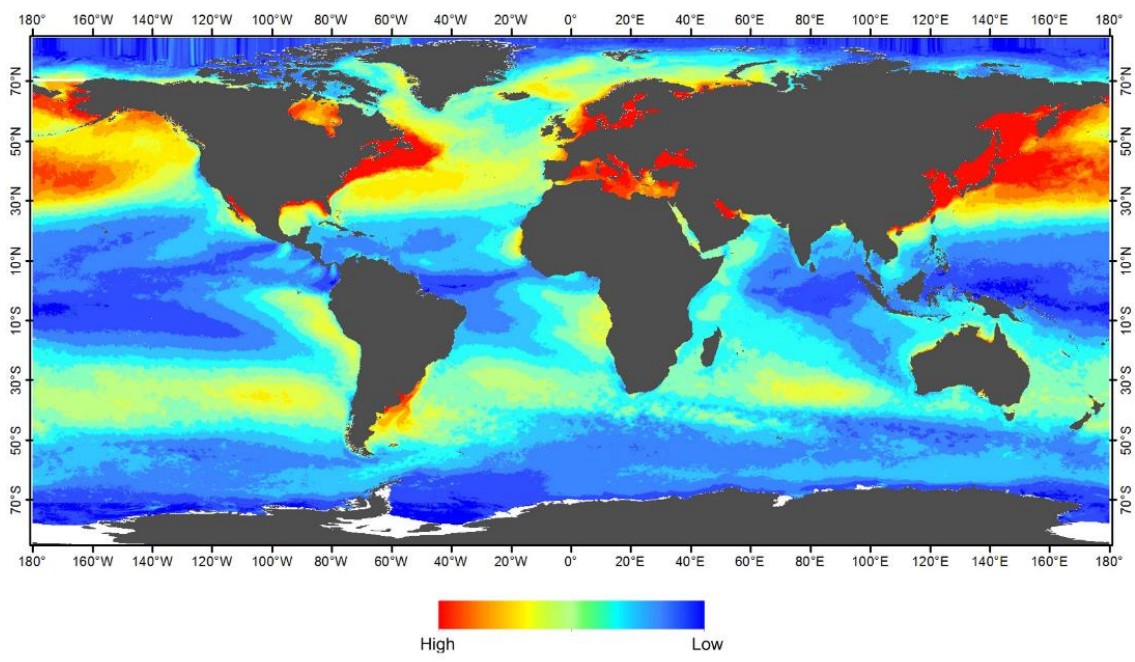



Figure A20. Sea Surface Temperature (May-Oct)

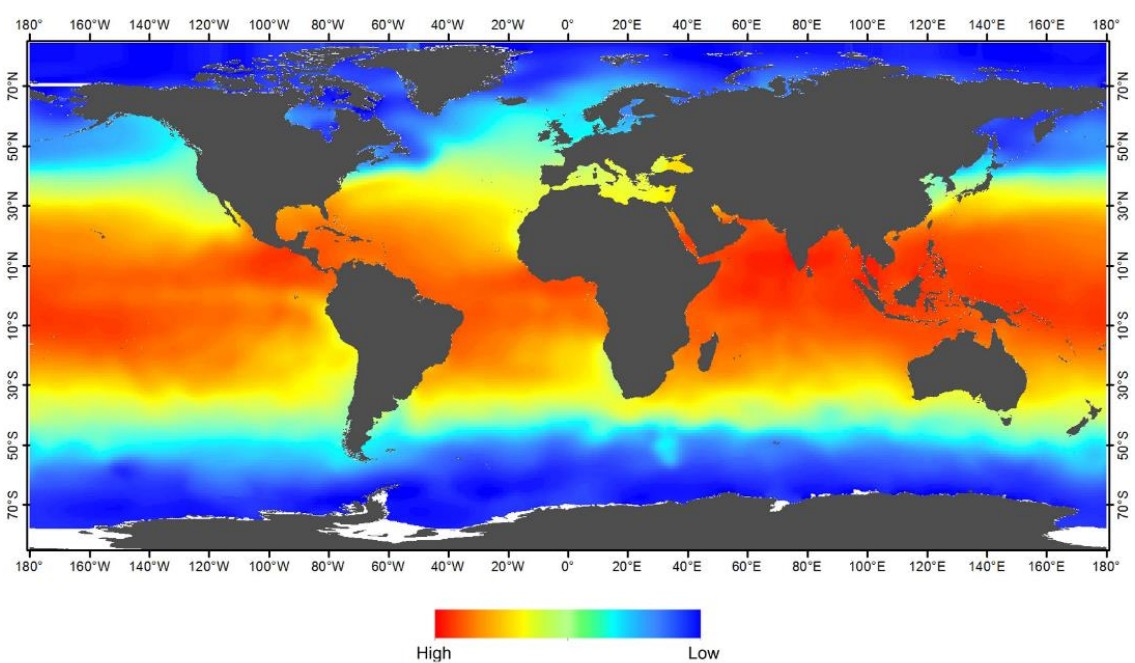

Figure A21. Sea Surface Temperature (Nov-Apr)

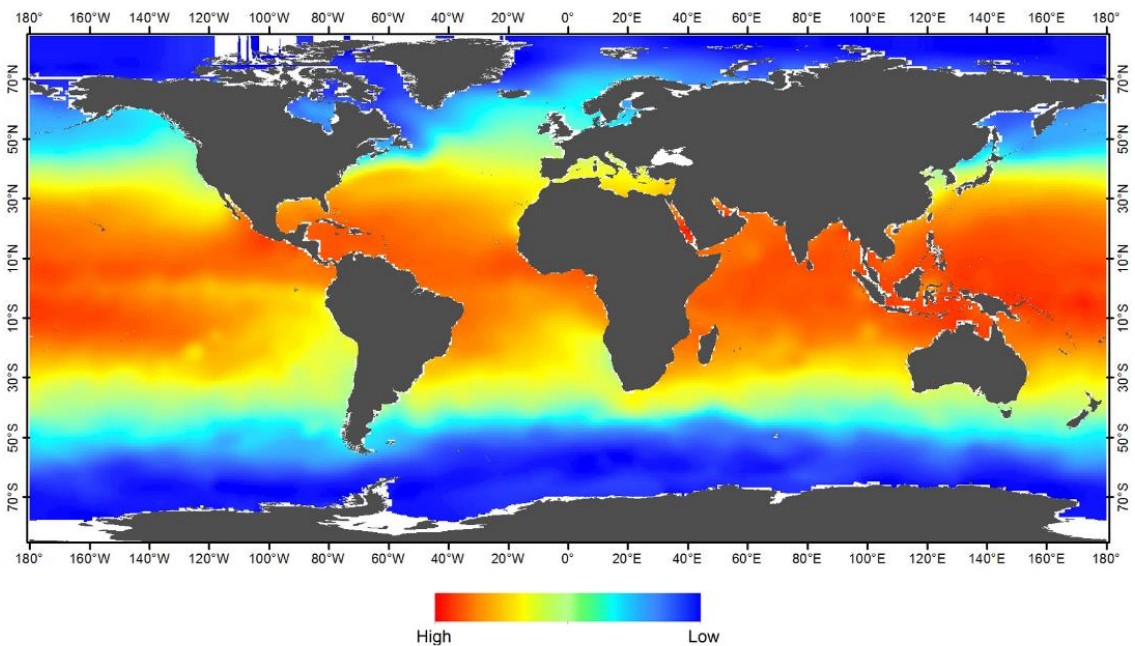

Figure A22. Seabed Temperature

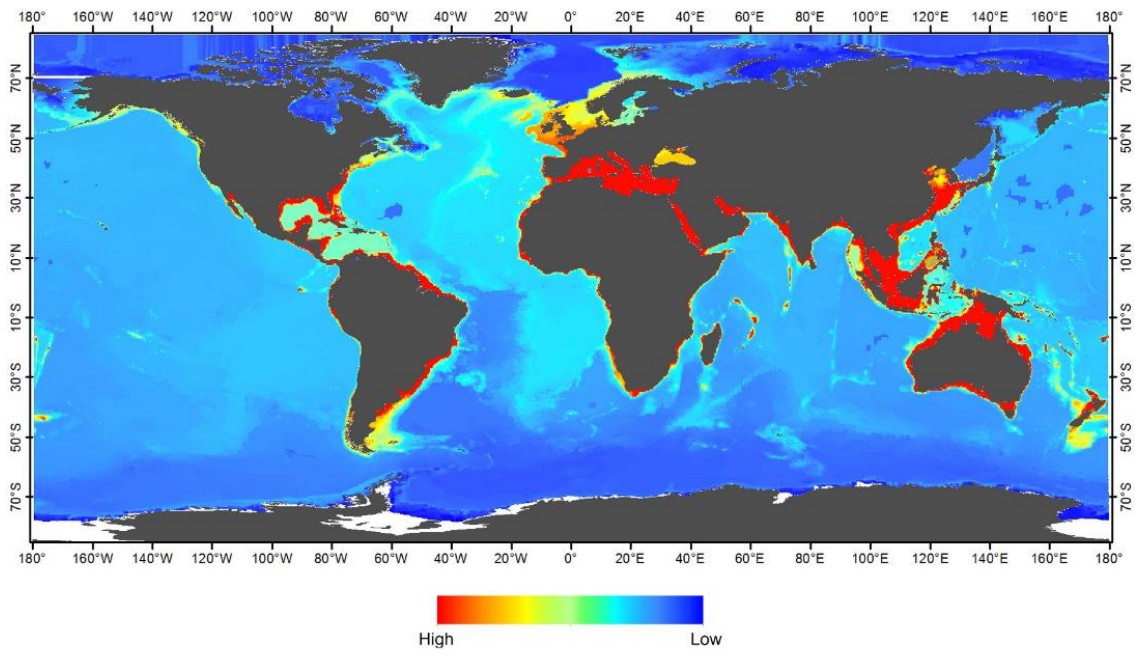

Figure A23. Water Column Temperature



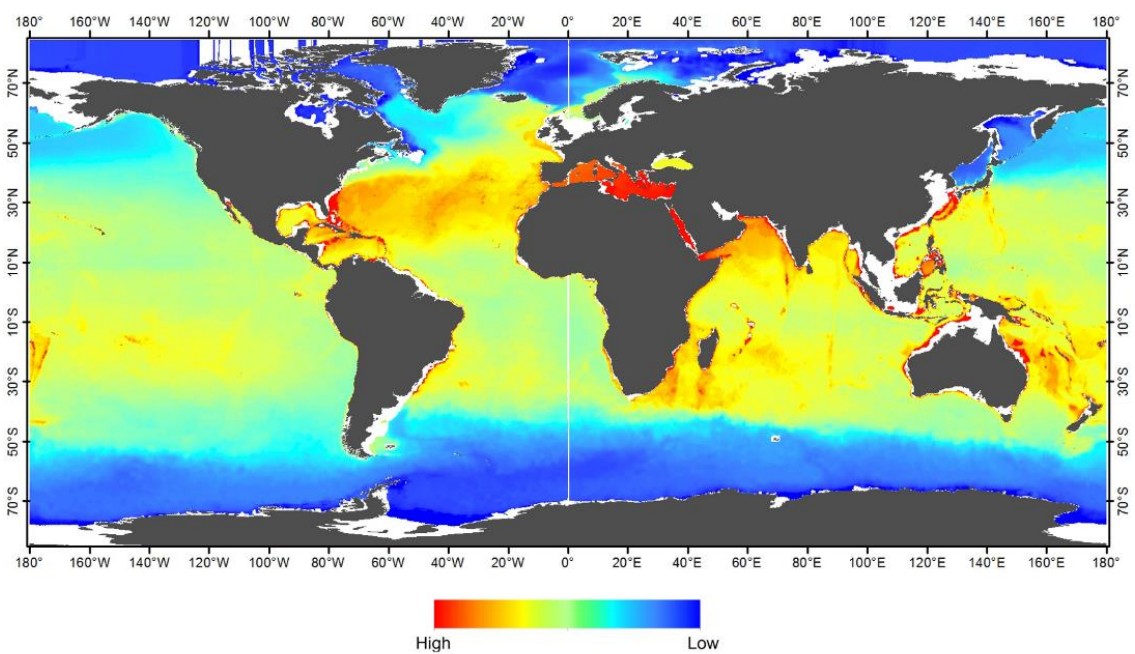

Figure A24. Surface Salinity

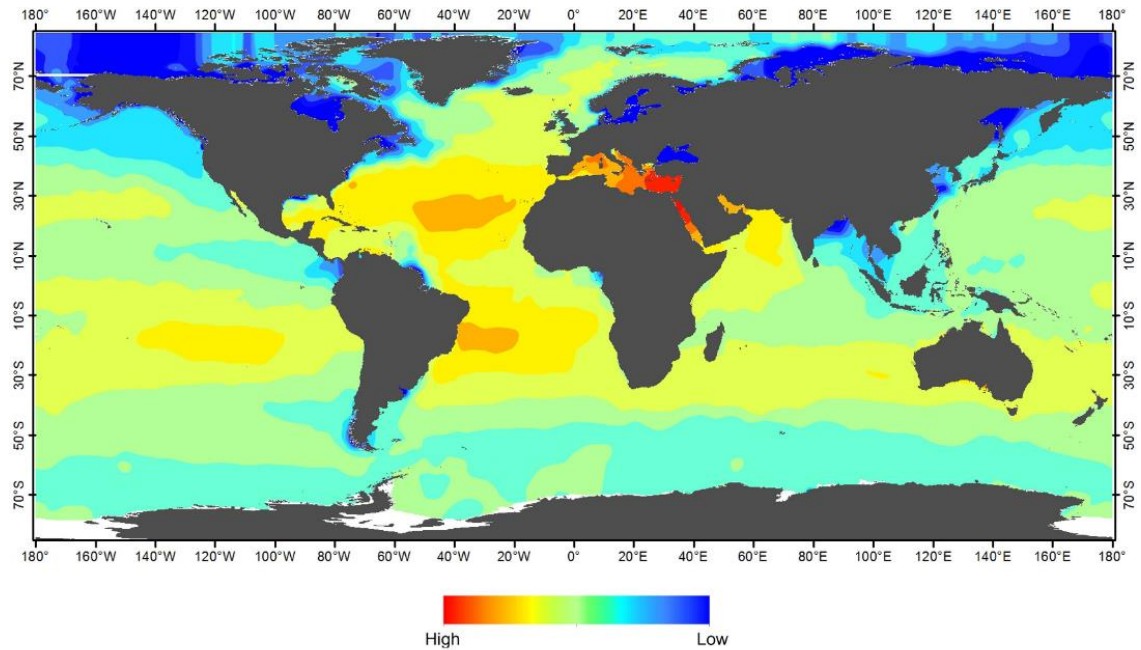



Figure A25. Water Column Salinity

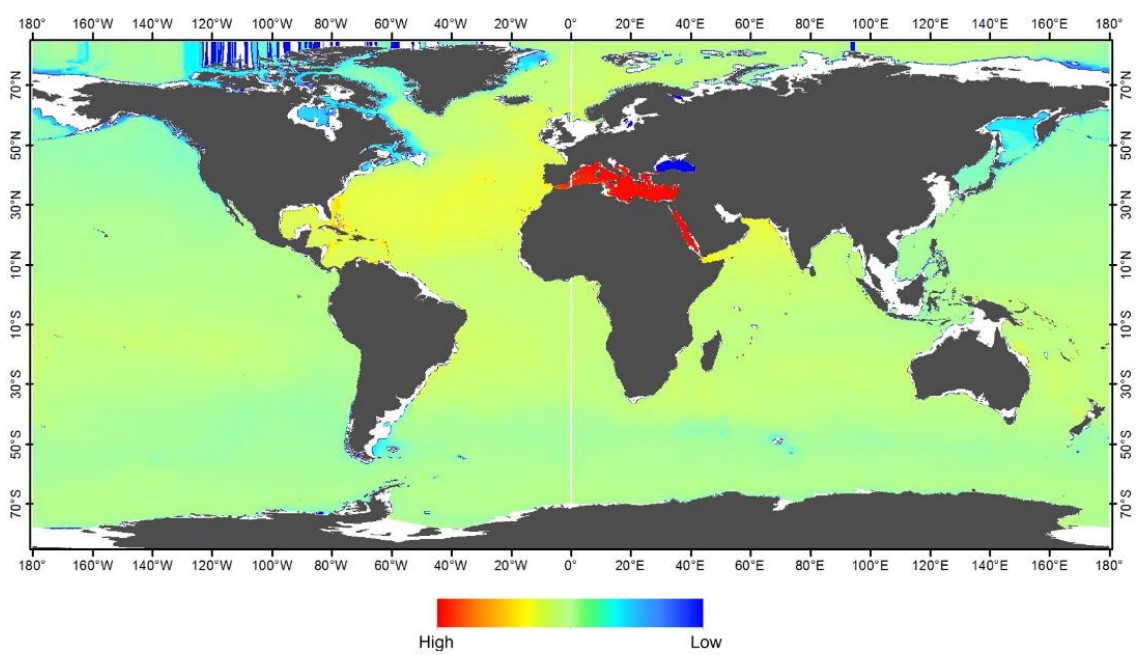

Figure A26. Photosynthetically Active Radiation



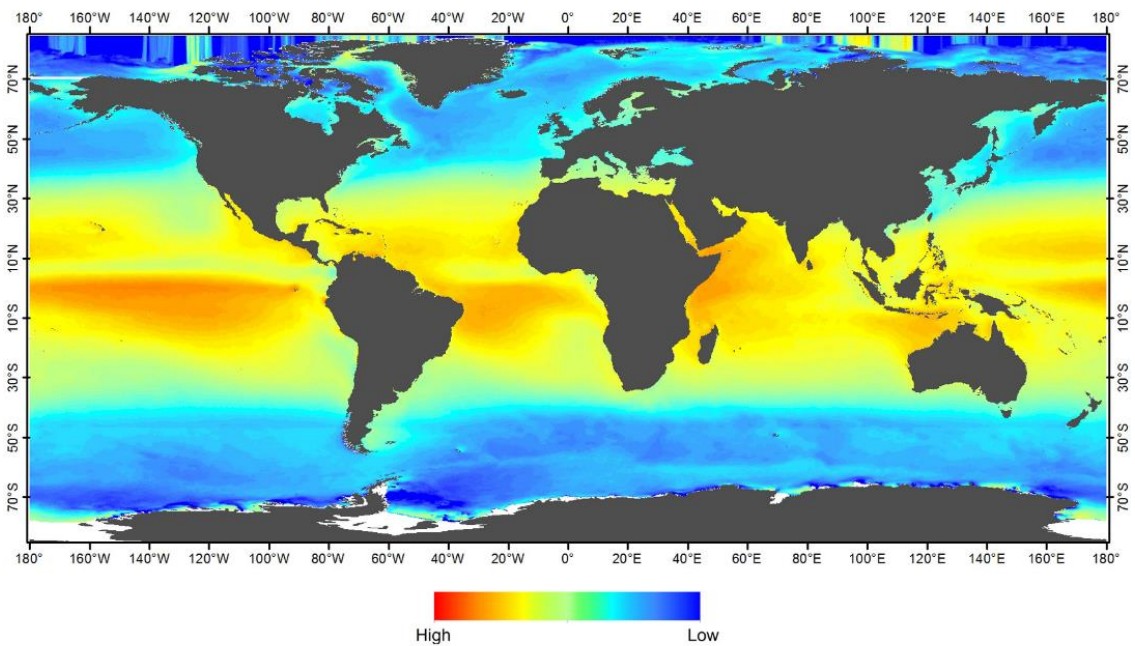

***Chemical***

Figure A27. Chlorophyll-a Mean

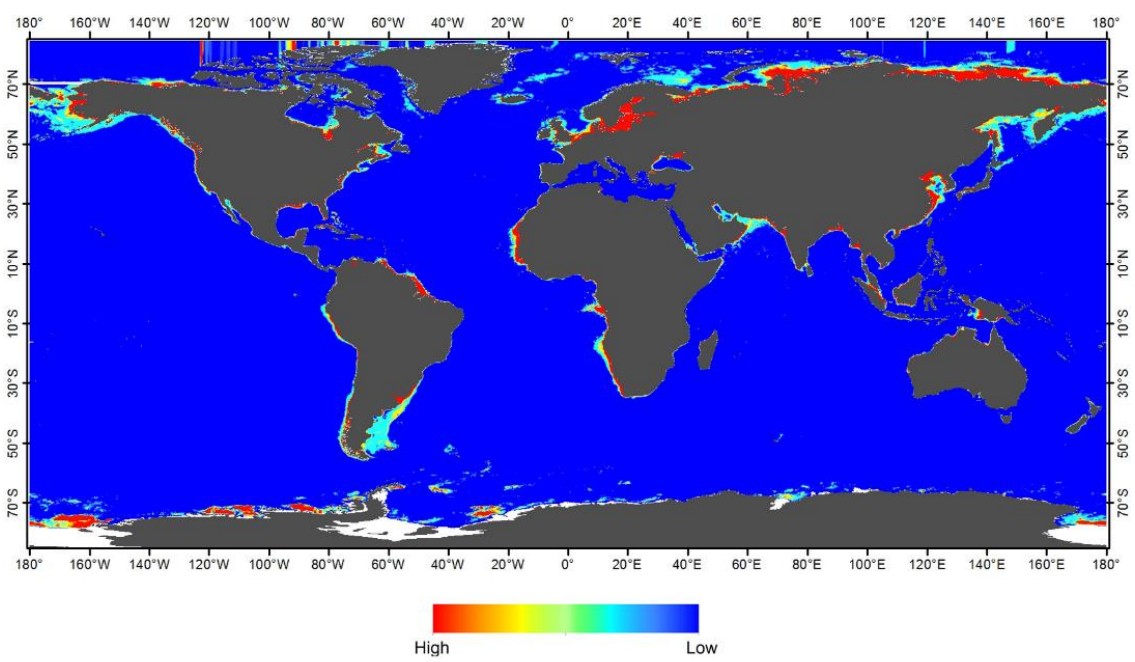





Figure A28. Chlorophyll-a Maximum

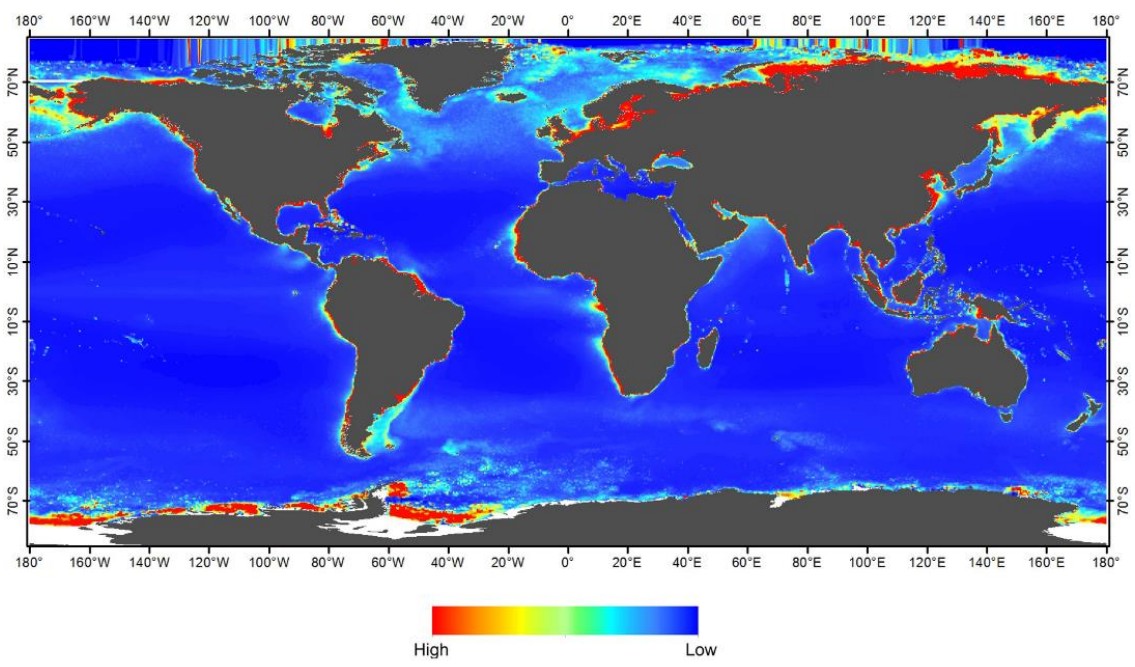

Figure A29. Chlorophyll-a Minimum

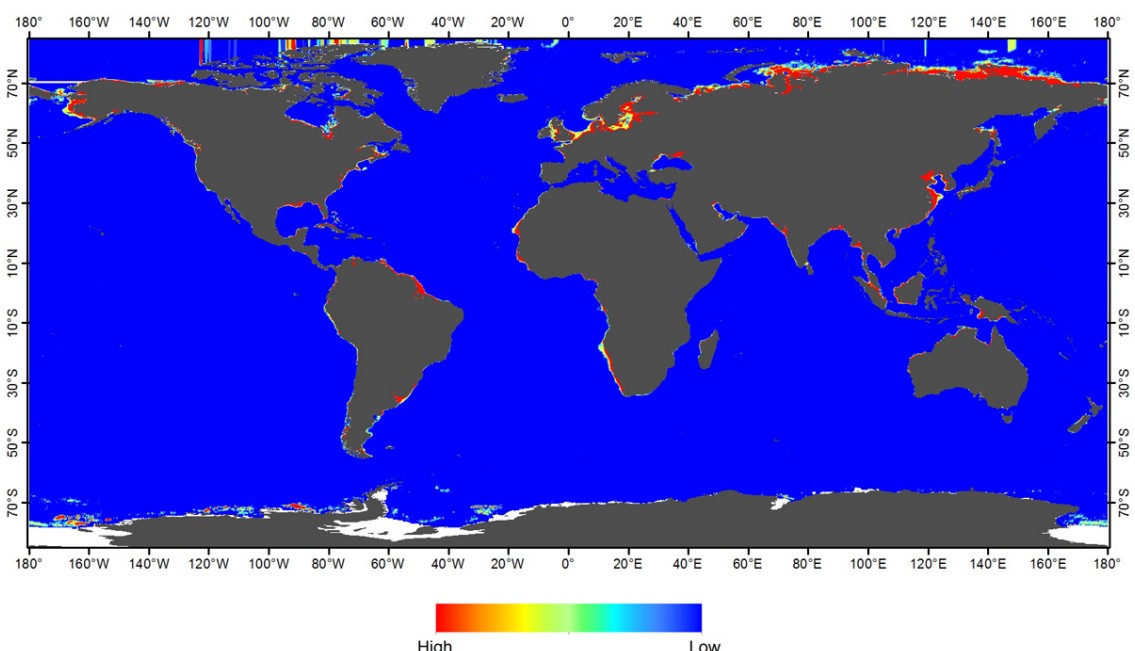

Figure A30. Chlorophyll-a Range

Figure A31. Chlorophyll-a (May-Oct) Maximum

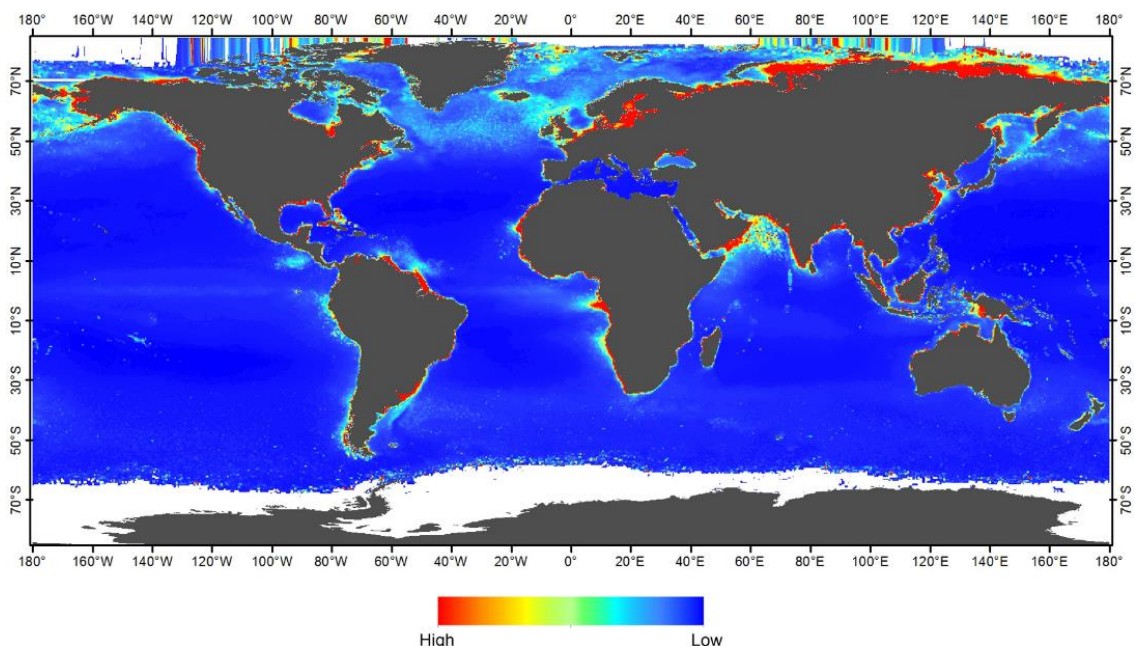



Figure A32. Chlorophyll-a (Nov-Apr) Maximum

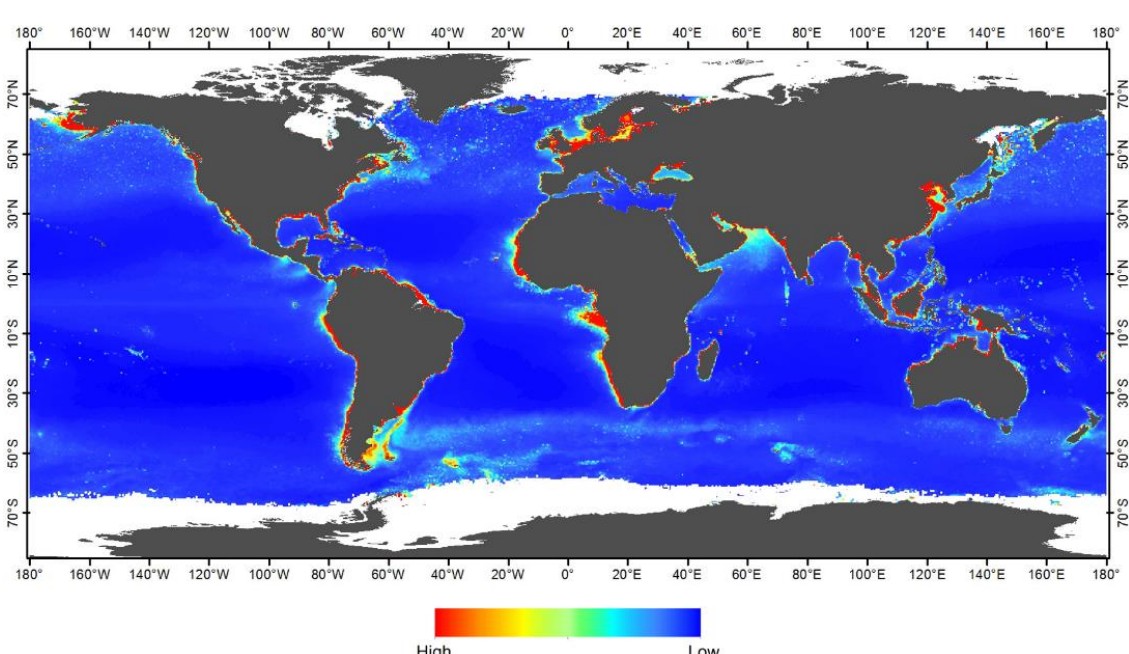

Figure A33. Primary Productivity

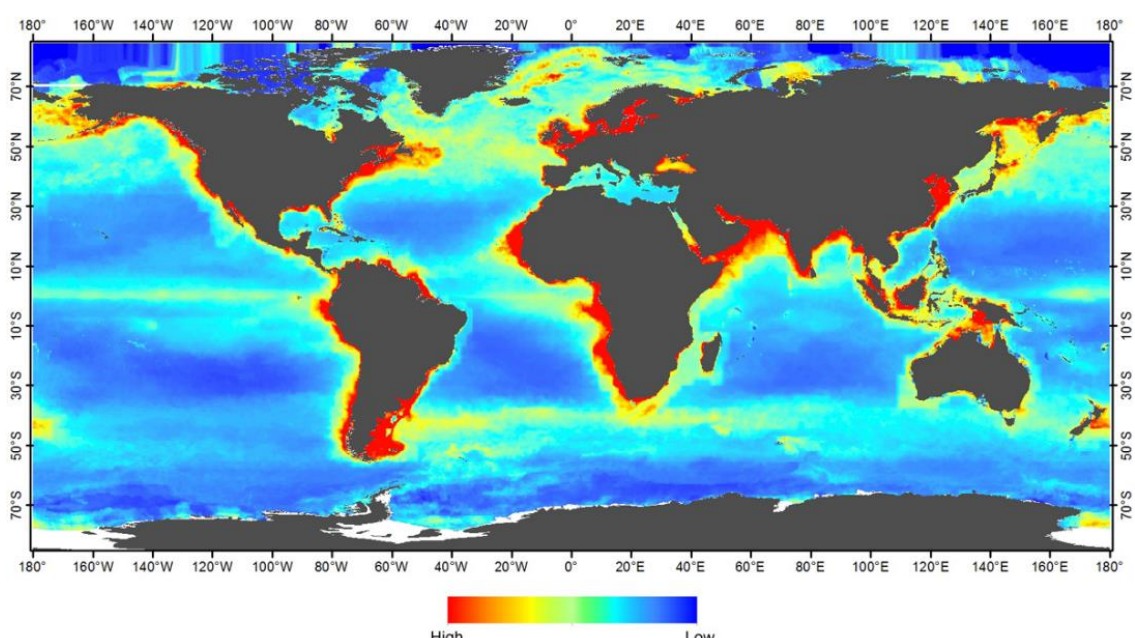



Figure A34. pH

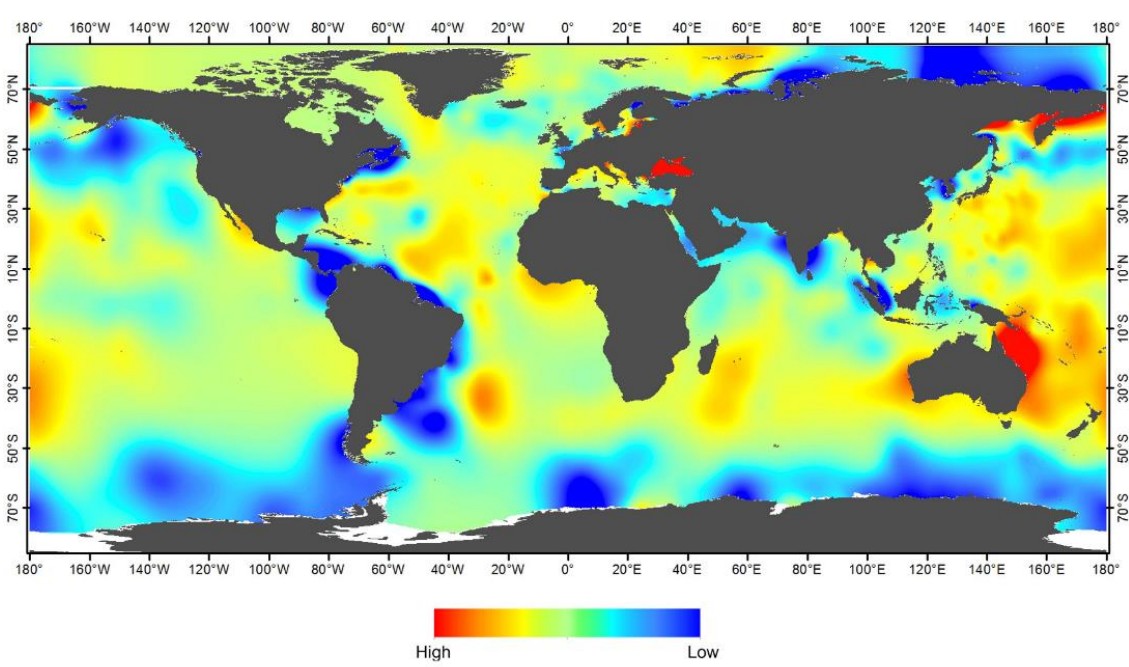

Figure A35. Total Suspended Matter



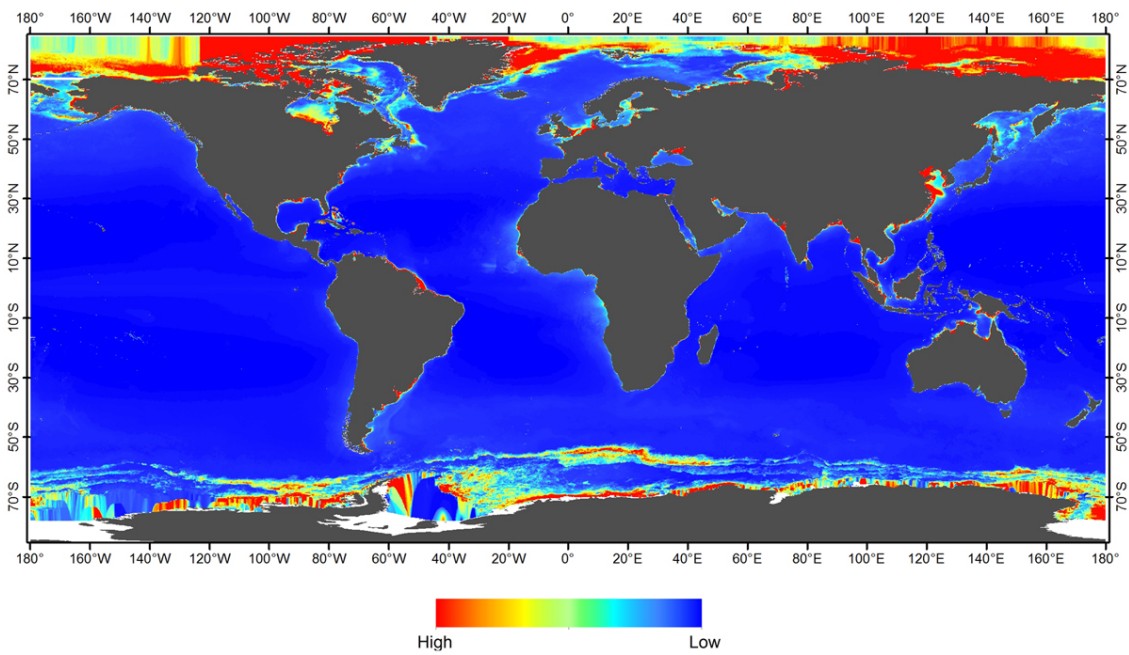

*Nutrients*

Figure A36. Calcite

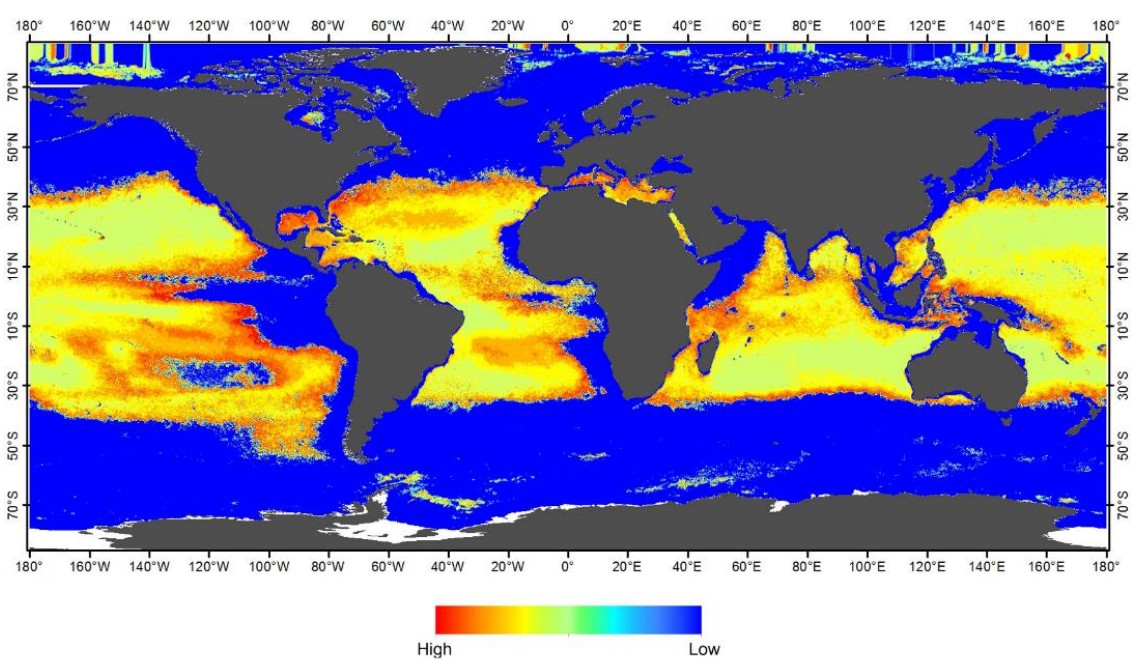

Figure A37. Nitrate

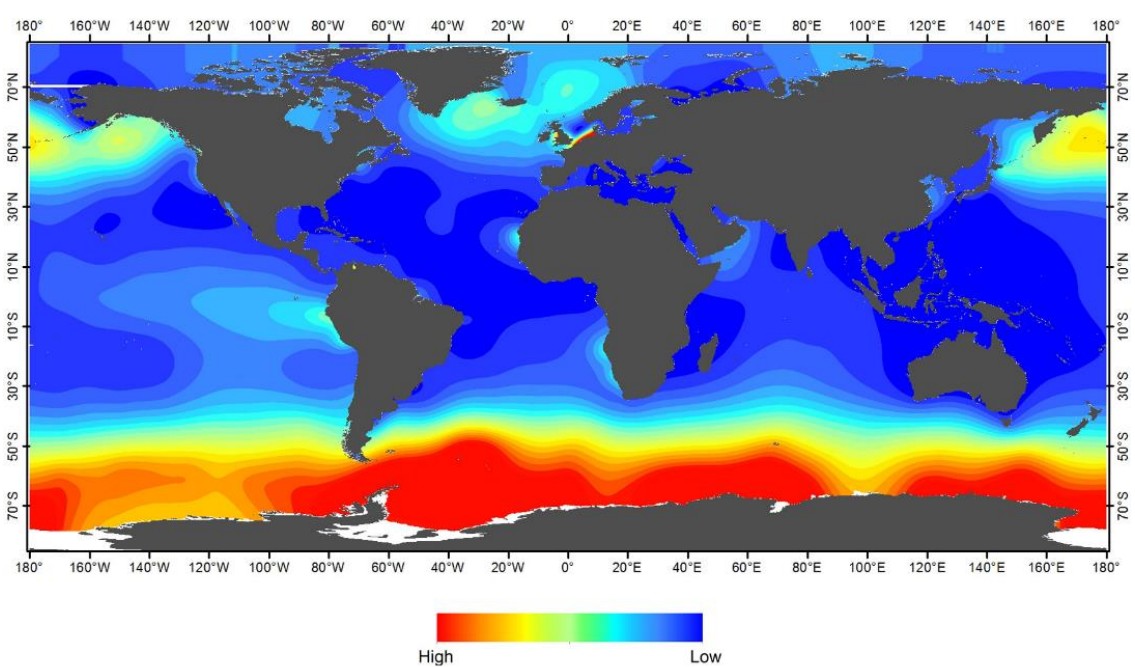

Figure A38. Seabed Nitrate

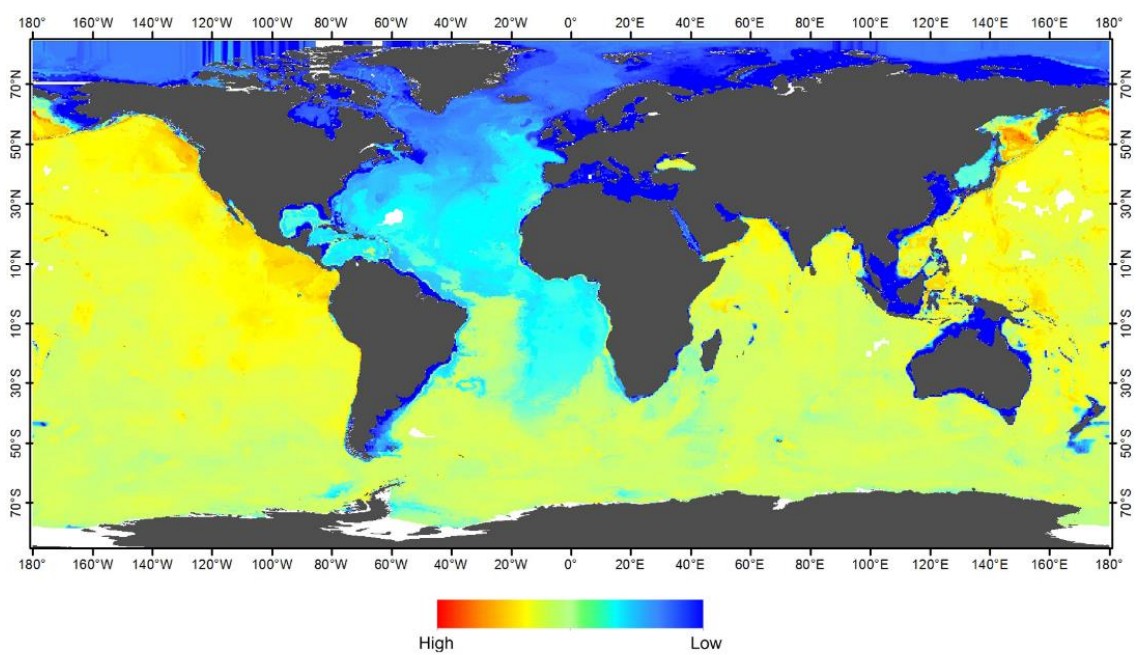

Figure A39. Phosphate

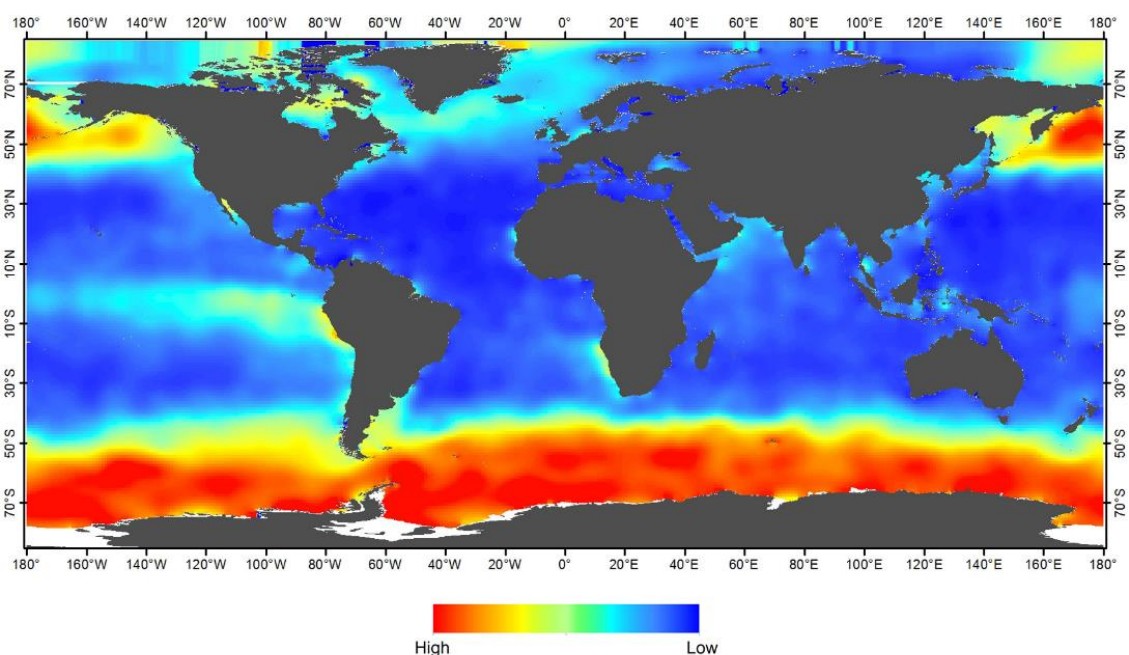

Figure A40. Seabed Phosphate

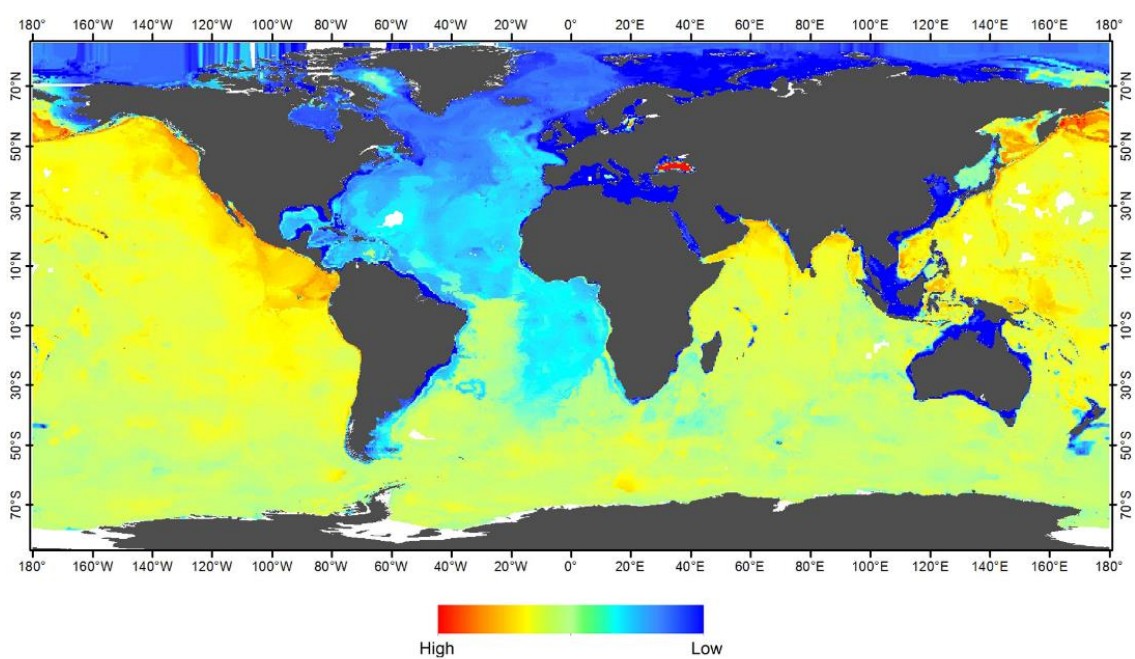

Figure A41. Silicate

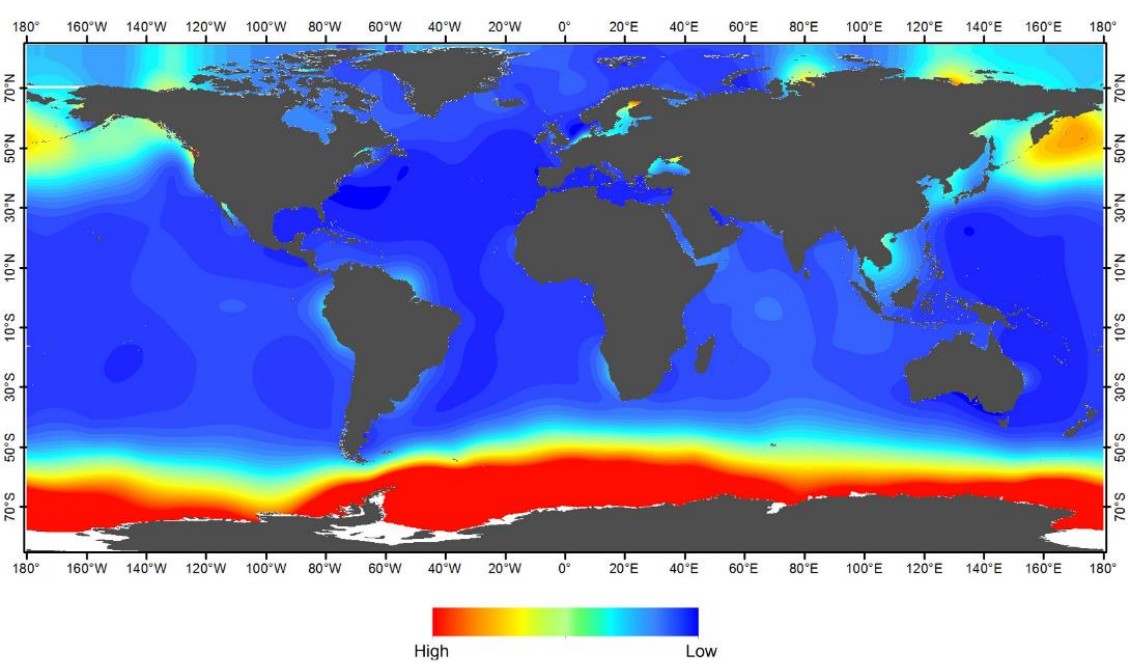



Figure A42. Seabed Silicate

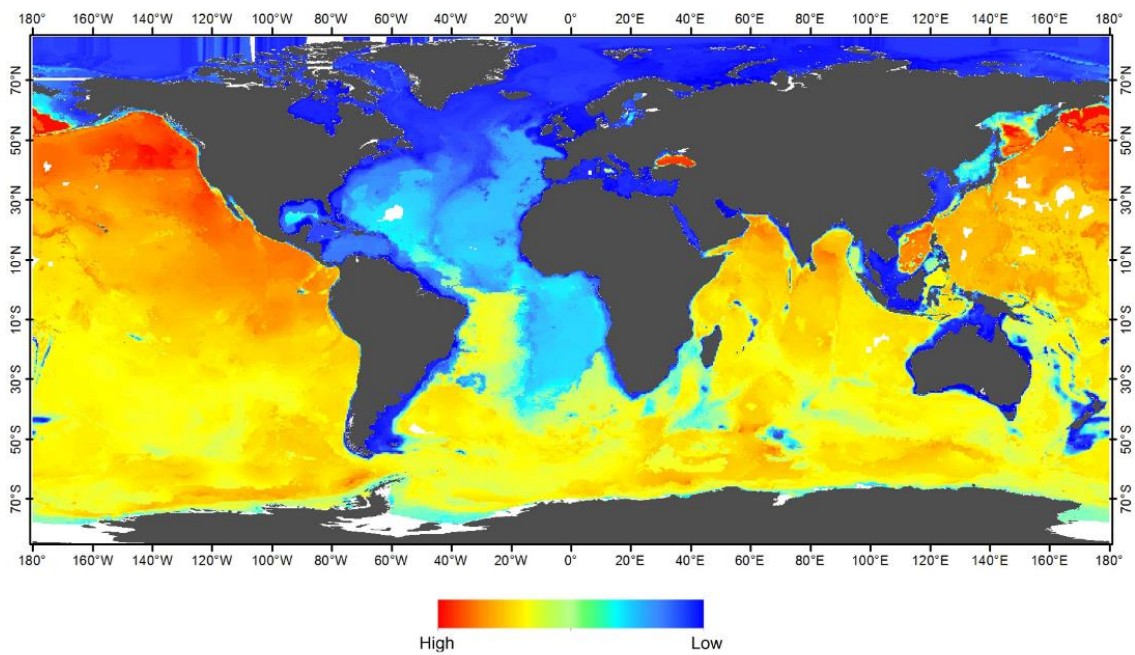

Figure A43. Dissolved O$_2$

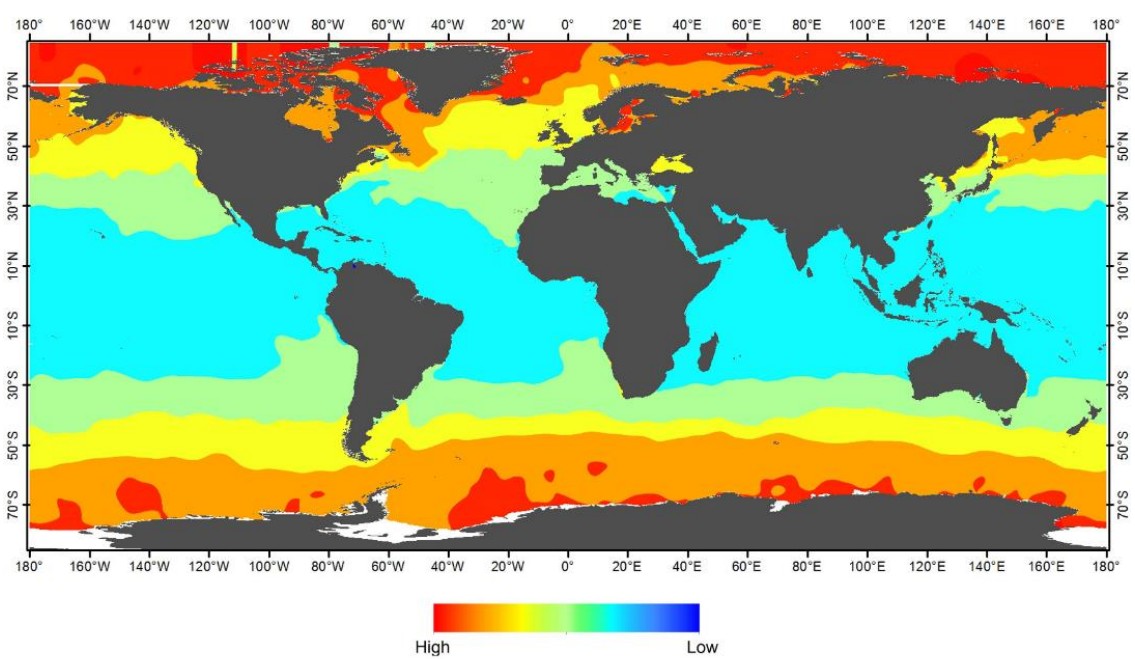


Figure A44. Seabed Dissolved O$_2$

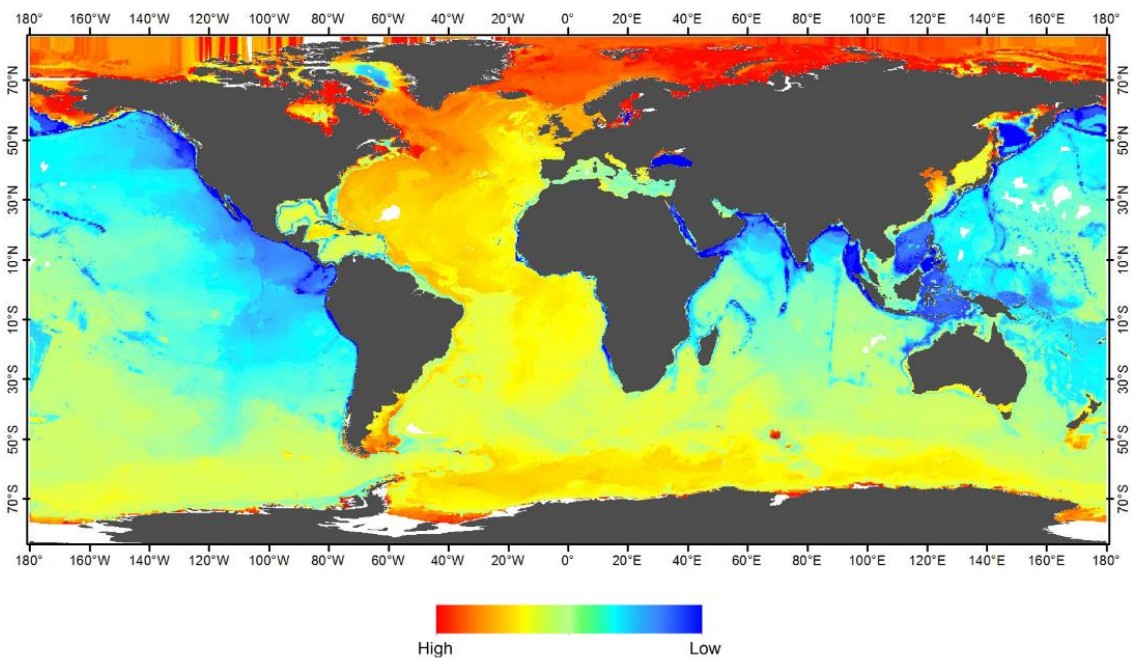

Figure A45. Saturated O$_2$

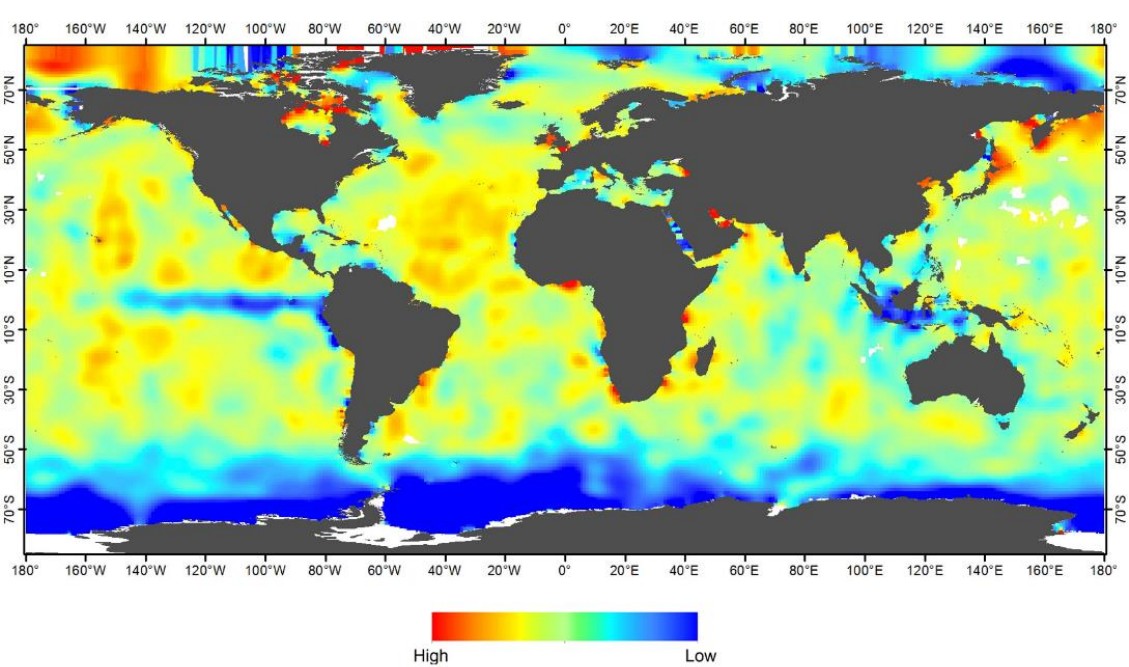



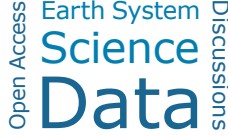

Figure A46. Seabed Utilized O$_2$

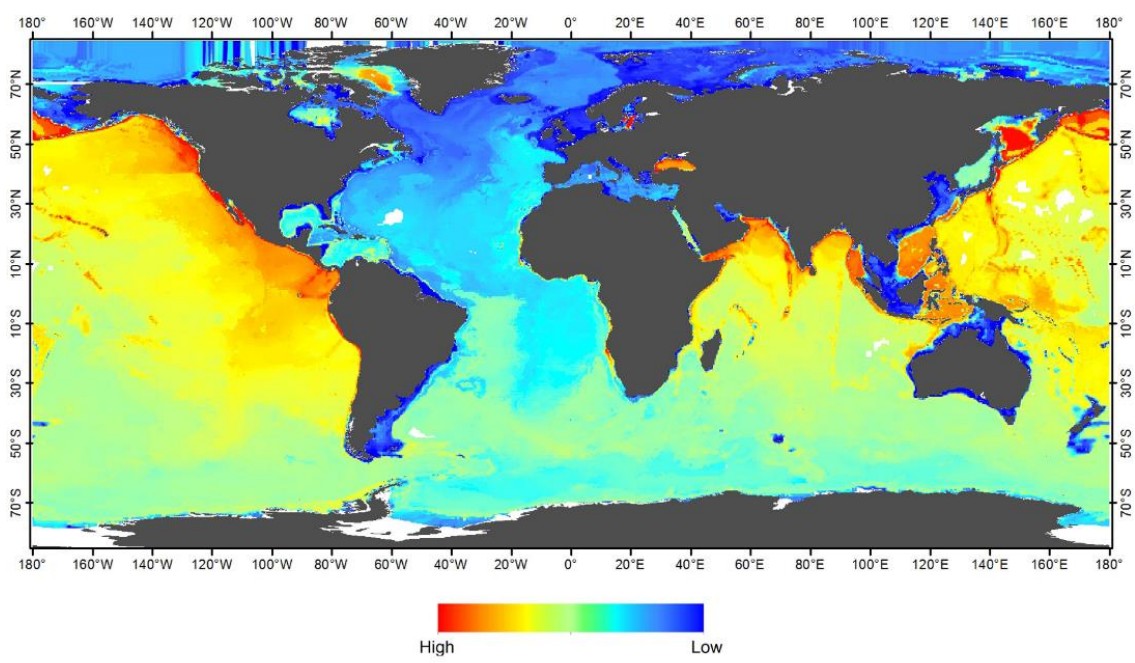

Figure A47. Particulate Organic Carbon

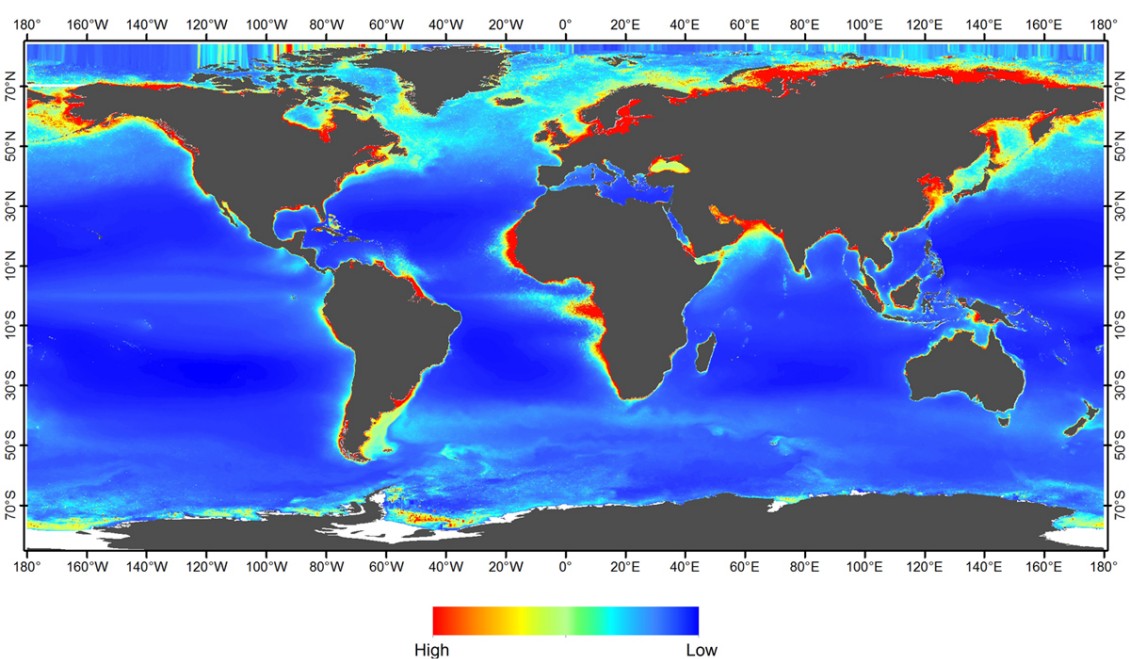





Figure A48. Particulate Inorganic Carbon

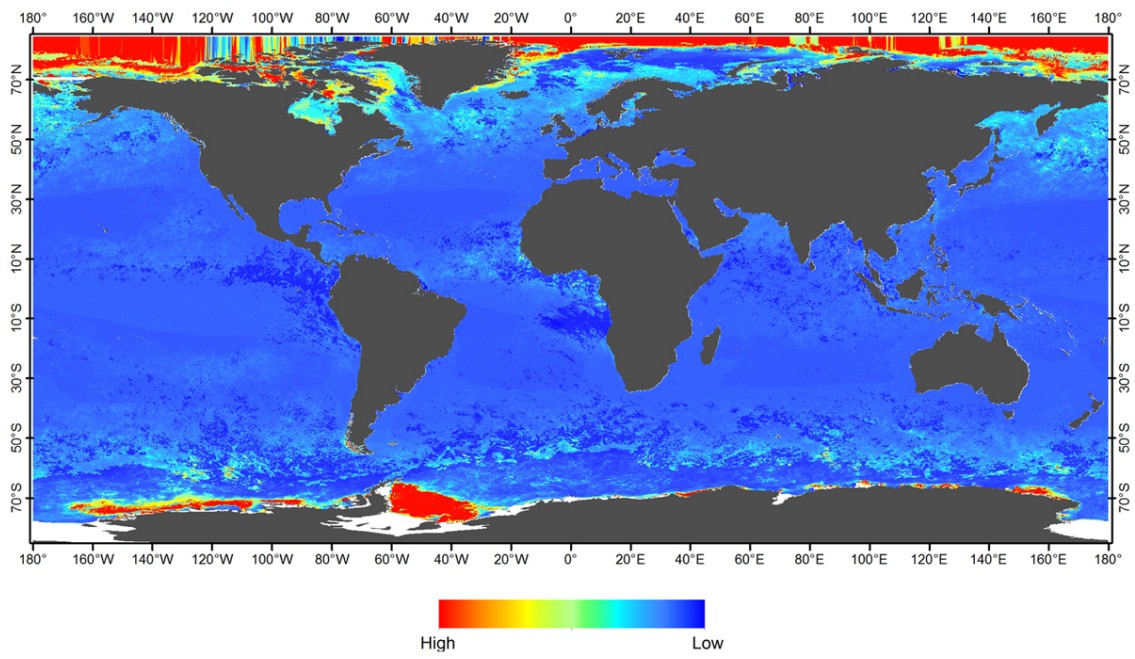



*Past (Last Glacial Maximum, 22 mya)*

Figure A49. Depth

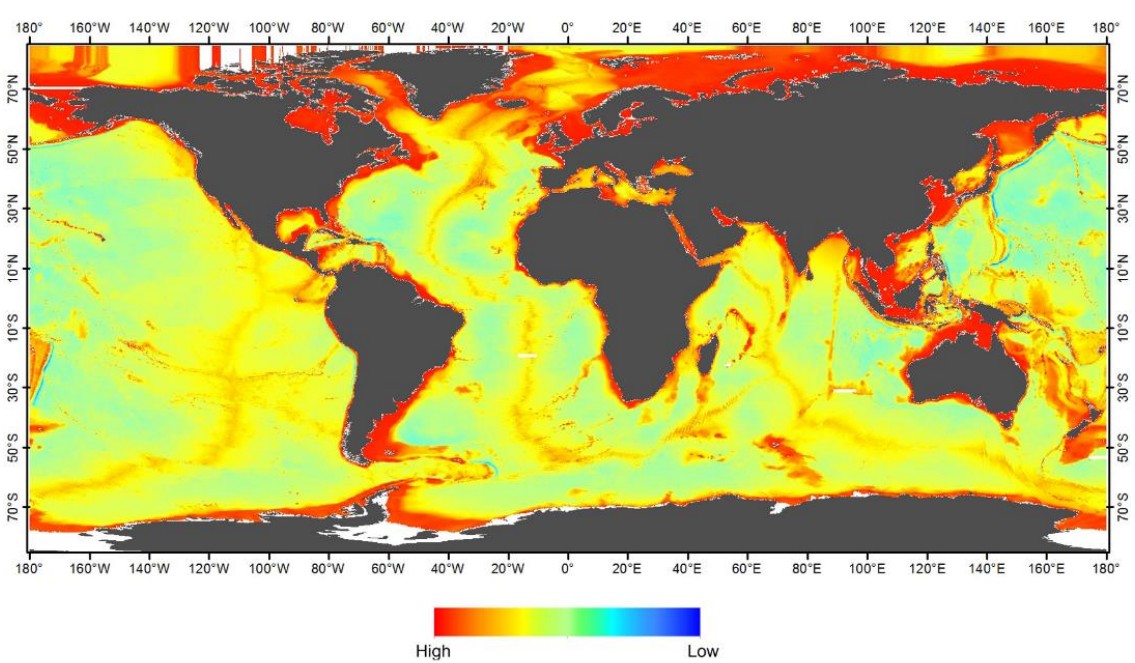

Figure A50. Temperature

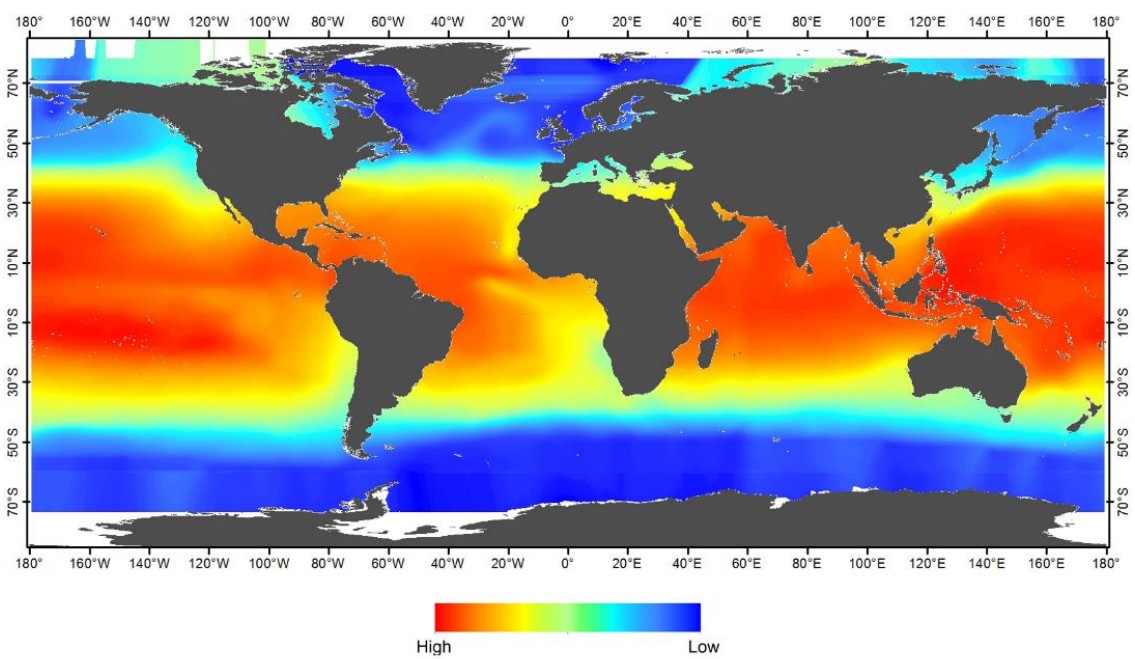



Figure A51. Salinity

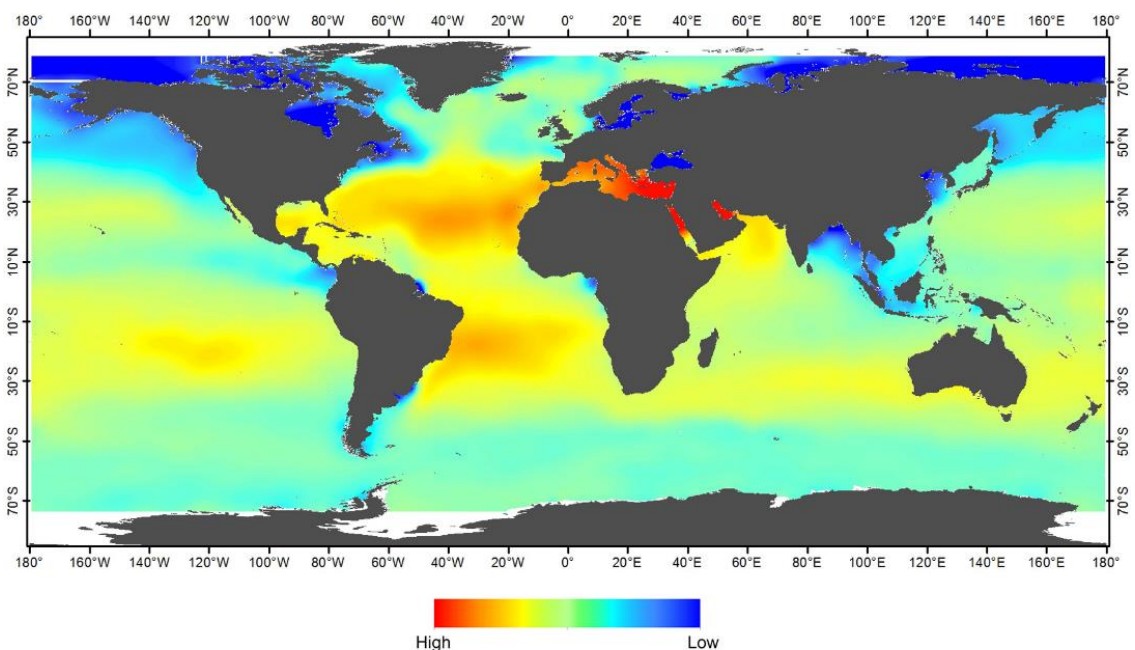

Figure A52. Ice Thickness

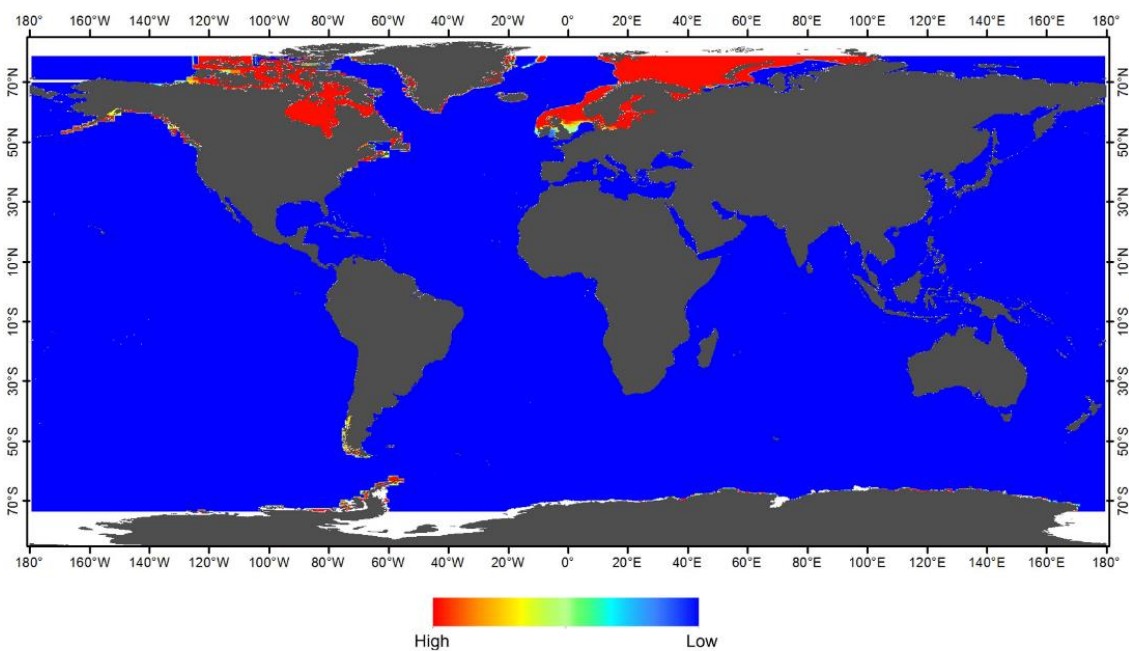



***Future (Year 2100)***

Figure A53. Temperature A1B Scenario

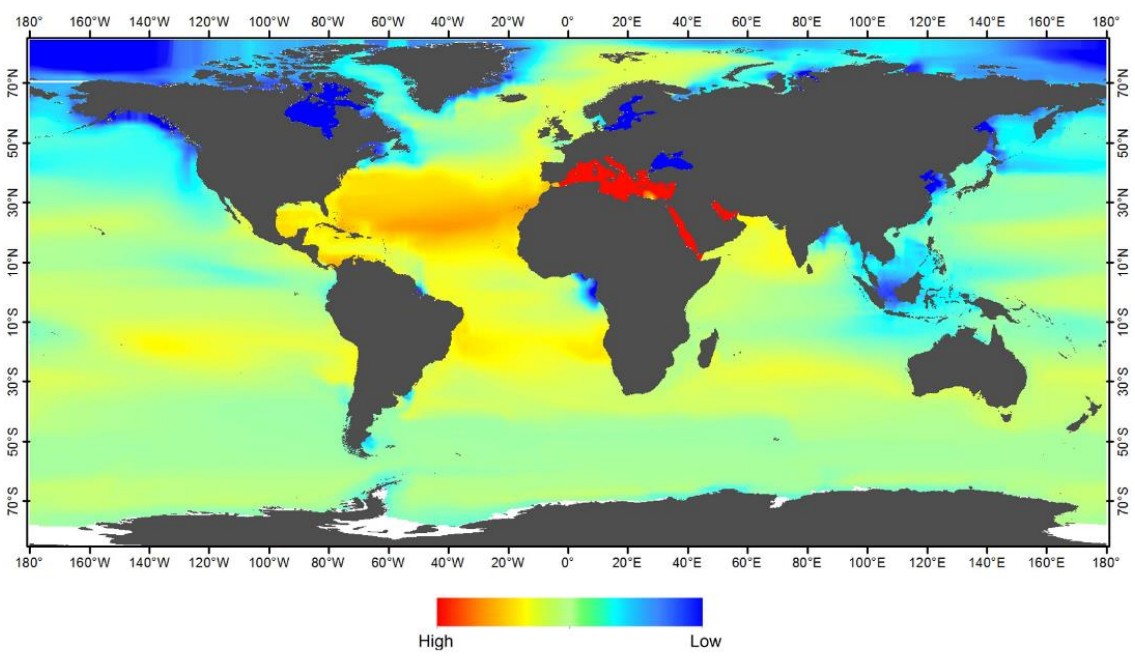

Figure A54. Temperature A2 Scenario

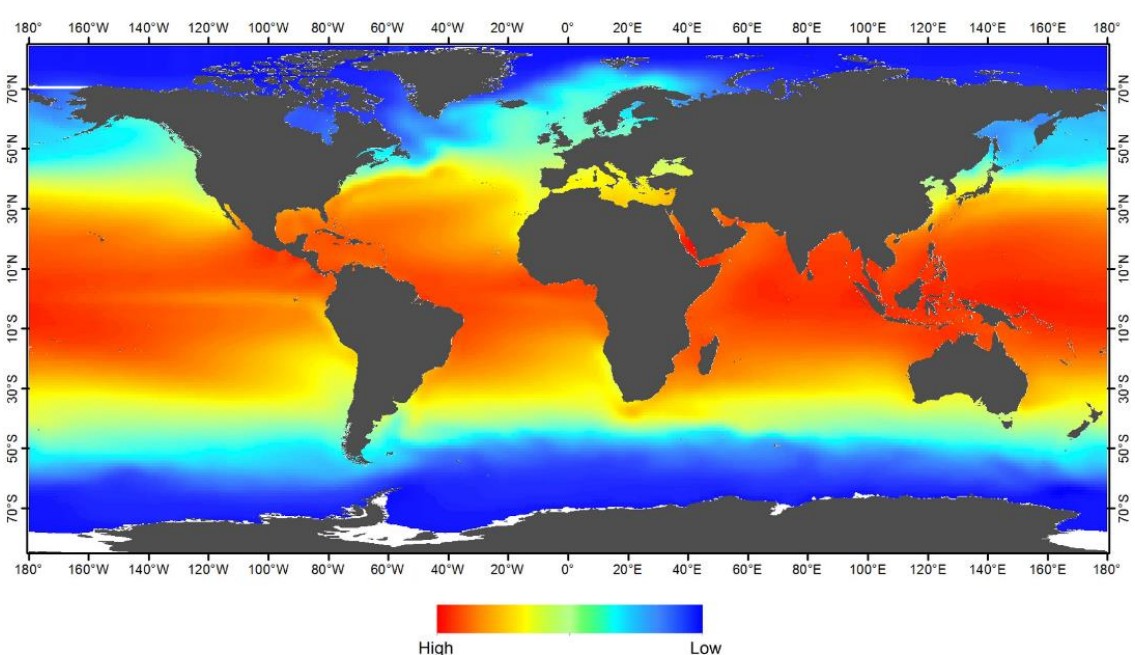

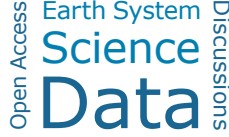

Figure A55. Seabed Temperature

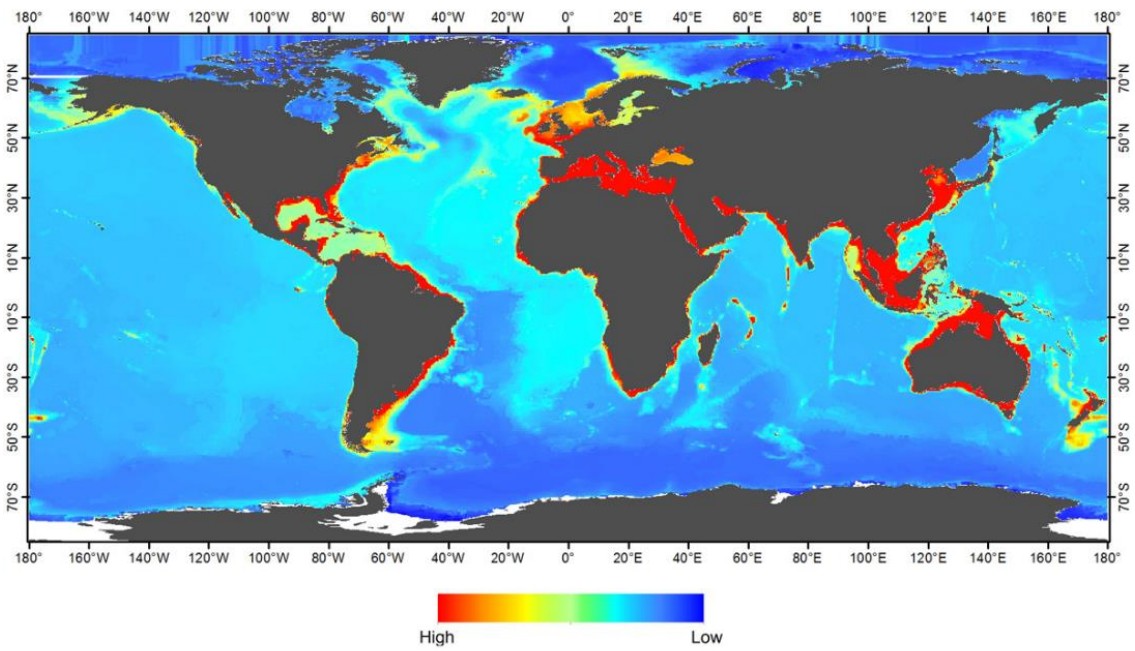

Figure A56. Salinity A1B Scenario

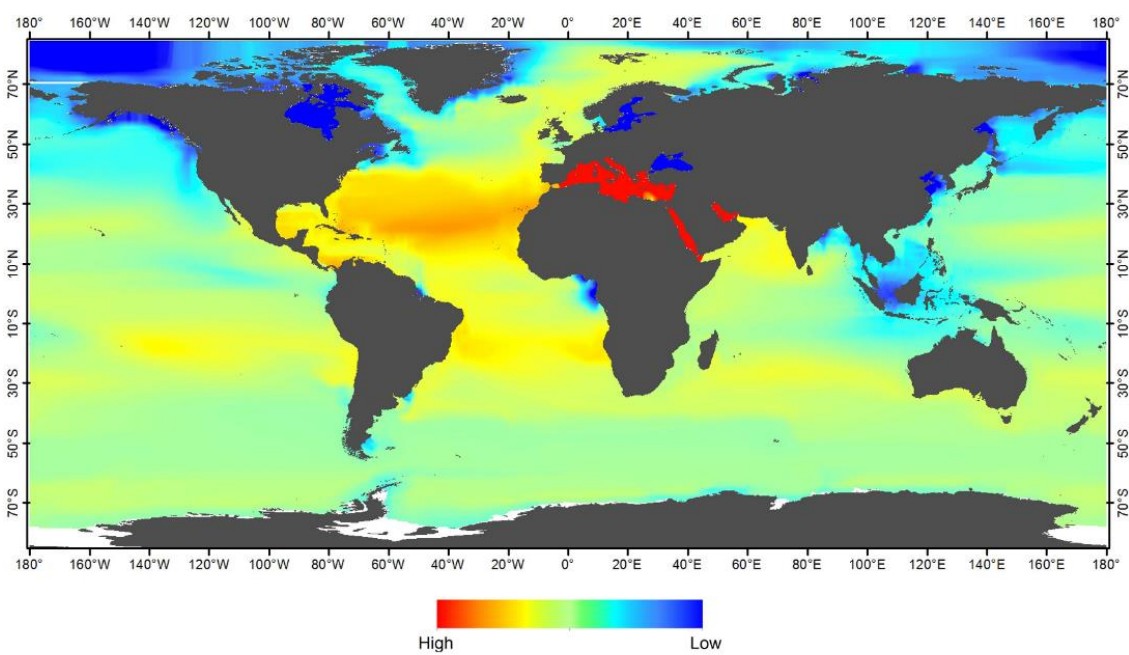




Figure A57. Salinity A2 Scenario

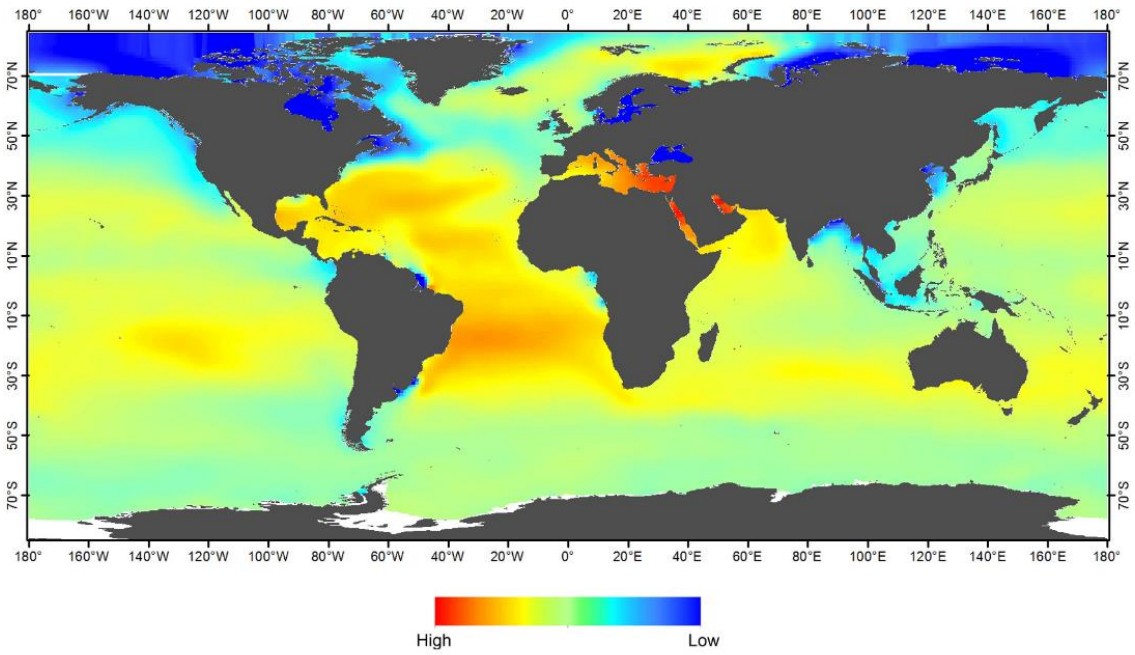

Figure A58. Seabed Salinity

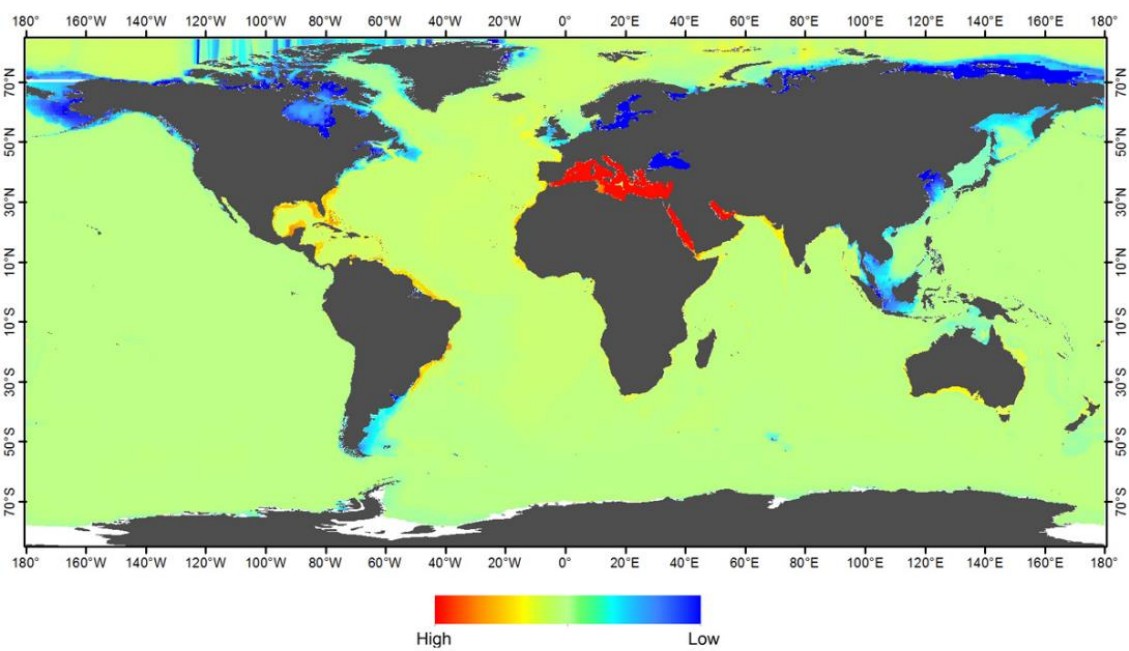



Figure A59. Primary Productivity

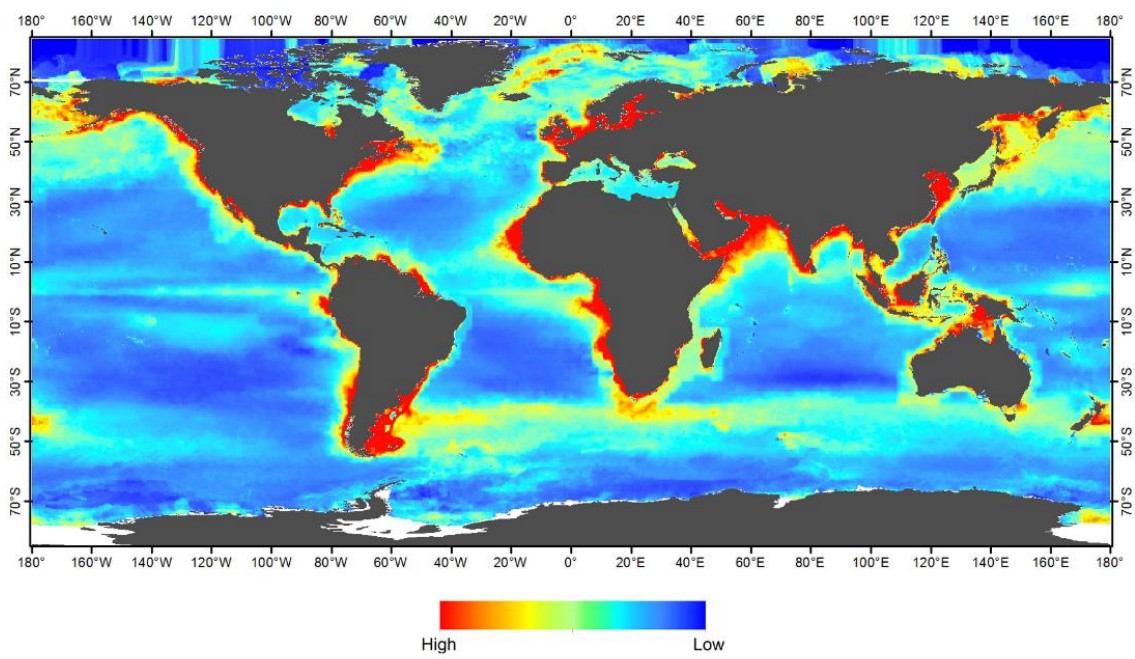

Figure A60. Ice concentration

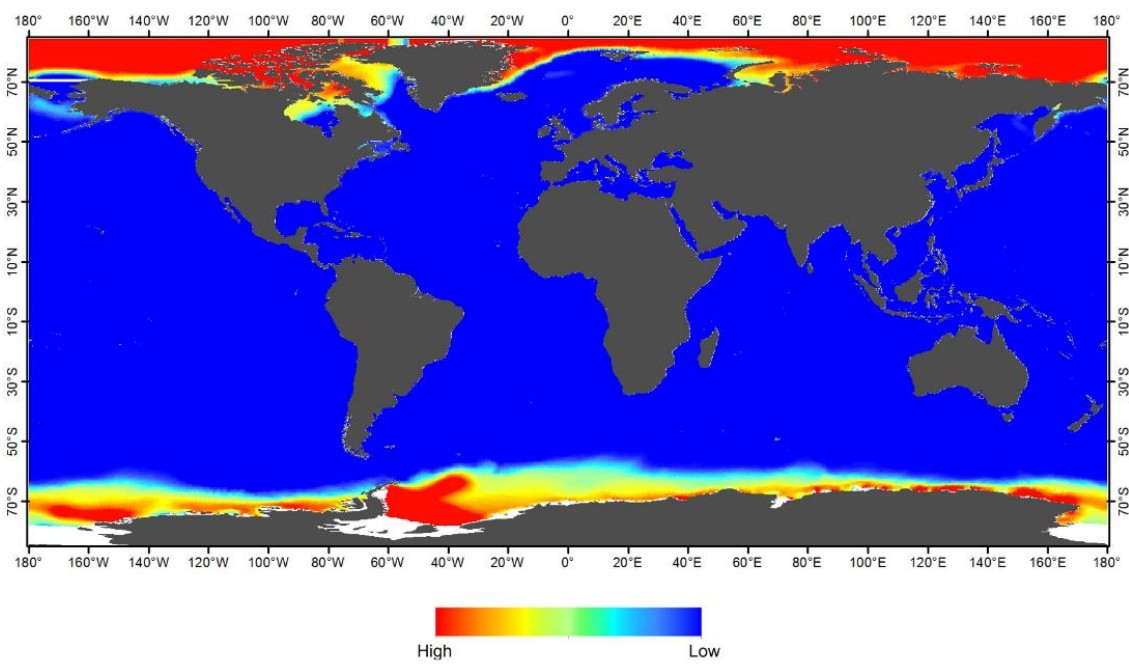