# Peer review of "Earth System Discussion Science usions"

_Earth System Science Data, 2018_

## Referee Comment (RC1) · Anonymous Referee #1 · 2 Oct 2018

Review ESSD-2018-64, Marine Environment Data Sets

Demonstrating minimal knowledge of ocean data, the authors have relied on obsolete sources and a proprietary (ESRI ArcGIS) software to 'produce' a series of unreliable data layers. Their product demonstrates neither quality nor reproducibility, the hallmarks of a good (ESSD) data set. Surprisingly, these authors seem substantially unaware of fundamental deficiencies.

Should we as readers hold these authors accountable for the very poor quality of their source data? Yes, in this case! The authors represent themselves (below) as competent data processors. They demonstrate their confidence by submitting their new compilation of previously un-reviewed data products for review and public comment in a prominent data journal.

These authors imply that based on "recent experience with developing compatible, comprehensive, environmental layers" they have "developed an extensive on-line repository of marine environmental data layers." Assembled, perhaps. Developed, no. Later, these authors admit that all data sets used for GMED "had undergone quality control checks by the primary data collectors and processors". These authors have applied a canned statistical interpolation (from ArcGIS), used some additional ArcGIS tools to check whether their interpolation introduced errors, and then served 60 layers of environmental … what? Reprocessed, improved and consistent data? Or, garbage amplified? Because these authors demonstrate minimal understanding of the quality, temporal extent or spatial coverage of their source data - as evidenced by the appalling array of old and obsolete sources - this reader regards their product as garbage, and old garbage at best. Apologies for strong language but one feels a need to 'wake up' these authors to a world of ocean data outside of their narrow view.

If in fact the ecology community accepts and relies on obsolete deficient ocean data sources then these authors may have done us an unintended favour by exposing the vast gap remaining between what oceanographers use and what ecologists use.

Data Access

The GMED url works, leads one to a useful landing page. The GMED version 2.0 link, at the upper right of that landing page, does not work. The 'DataSets' link does work and presumably leads to the layers discussed in the manuscript.

For this reader the figshare doi did not work in any configuration. ESSD generally does not use figshare for exactly these reliability issues. If the authors have a GMED snapshot they can deposit at figshare, they could easily - and more usefully - deposit it at Zenodo, Pangaea, 4TU, etc? Not reliable as presented, needs a change.

Page 3 starting line 59: I think you mean to say 'Differences in applications of SDM to marine environments compared to terrestrial environments include fewer observation records, extensive spatial-temporal variability of oceanic environments, and complexities in processing marine environmental data for SDM purposes'? But you would need to back up this statement with facts? From a carbon point of view, we might know less about land sources and sinks than we know about ocean sources and sinks? The SDM community takes a different view than the biogeochemical carbon cycle community? The authors seem blissfully ignorant of these larger issues, as perhaps they should, but to publish their data in the same journal that publishes (land and ocean) carbon cycle work they need to assure readers of an awareness of larger context?

Page 3 line 61: the phrase "spatial-thermal" seems strange here (I can find it used in other marine ecology papers) and I doubt the supposed differences marine to terrestrial. Spatial: ocean has surface to deep, land has sea level to alpine. Both land and ocean have strong latitudinal gradients and high variability in immediate coastal zones. If you truly mean thermal rather than temporal, ocean has -1 to 32C, land (even if we exclude ice) has -40 to +45), with ocean temperatures much more consistent than those of land. Ocean proper has very small range of salinity compared to large range of humidity on land. Ocean has no light to full light but so does land. Both have distinct daily and seasonal patterns that strongly impact biology. If you mean to imply range of habitats, we don't know much about 3-D patterns in the full-depth ocean but we

know that land includes a wide array of habitats. You will need to help the reader understand what you mean here because as written you have allowed a too-wide variety of misconceptions.

Page 3 lines 65 to 67: two biggest improvements in ocean monitoring probably come from ocean colour by satellite (e.g. MODIS and follow-ons) and globally-distributed profiling floats (e.g ARGO and, one hopes, soon bio-ARGO). (Some will also argue for the strong impact from GRACE.) This discussion of ocean observations seems to miss these crucial developments?

Page 3 line 70: many other groups have done a substantially better job at compiling and quality-controlling ocean data sets than Tyberghein and the bio-ORACLE crowd. To understand ocean properties from a biogeochemical or ecological perspective, a modern unbiased user would probably not start from bio-ORACLE? This entire discussion (lines 70 to 75) seems impossibly simple and surprisingly ignorant of other substantial efforts. Large literature exists on ocean SST. Substantial literature exists on ocean Chl A, much of it in ESSD, and extending quite far back before 1997. Have these authors never heard of GLODAP? Remote areas from an ocean data point of view might point to the central Pacific rather than (or as well as) the Arctic? This discussion pretty much convinced this reader that these authors lack working knowledge of current ocean data efforts.

Page 4 lines 78,79 "continuous, global, layers for such variables are predicted from ocean circulation models and by extrapolation of in situ sample data." A very large and very active community of ocean observationalists and modelers would strongly disagree with this glib uninformed attempt at summary. These authors provide no citations to back their contentions?

Page 4 lines 80, 81: resolution of ocean models represents a hot and active topic, with variety of useful approaches driven by both physics (eddy-resolving, mixing parameterisations) and biology. The authors seem aware of none of that current vital work. Redfern 2006 represents a weak, almost irrelevant, reference.

Page 4 line 85: www.worldclim.org represents a weak, slow, out-of-date data source? Who uses it? The site provides no quality control nor uncertainties, they only repackage data from other unspecified sources. No climate modeller uses these products. Everything in one format but at what cost? Data only goes to year 2000? I wouldn't touch it. GMED wants to emulate this?

Nice to see (lines 99,100) GBIF and OBIS mentioned but those do not include physical or chemical parameters. Typical of all GEO efforts, GEOBON provides nothing itself but only tries to ride on data sets produced and quality controlled by others.

Based on this introductory discussion a reader comes to doubt whether the authors have any knowledge of ocean observational (in situ, satellite or re-analysis) data sets.

Page 5 lines 115 to 117: basically, land-masking, then interpolation using standard ArcGIS tools, followed by comparisons with data sets already dismissed as deficient. Somehow this represents a useful contribution?

Figure 1 of terrible resolution, basically not even readable. Better figure in Supplement but why should reader need to look in Supplement to find a readable first figure of the paper?

Above, the authors dismissed AquaMAP and KGS Mapper as inadequate due to (apparently) complexity. Here they use the same two sources as their primary data sources? Later they then compare back to AquaMap and KGS Mapper. Can one imagine a more circular process, with less actual quality control contribution?

Page 5 line 128: Jungclaus 2006 - one run of one version of ECHAM5 at T63 downloaded from WDC, now more than 10 years out of date? Run at SRESA1B - an optimistic (and now also out-of-date) emission scenario? Do the authors somehow need to prove their ignorance of global ocean and climate data sets? Kaschner references merely a back-door route back to AquaMAPS.

Page 5 line 128: Kaschner et al 2013 url link does not work. And no wonder: 5 years since last access? Why insult the reader with useless out-of-date links?

In Table 1 authors claim to have used climate projections from CMIP3, one iteration of the UK Met Office model - with no emissions scenario information (as it turns out, already chosen by bioORACLE according to Table S1) - and one from IPSL, configuration unknown but evidently with a full depth green ocean and a coupled ice model, run at A2 (again from Table S1, already selected by and available from AquaMaps). Does reader follow the text (line 128) or the table? No guidance and too little information for anyone to attempt to replicate and confirm. Do the authors even know what they have used? Panels on the GMED landing page, for year 2100, credit UK Met / Hadley and IPSL, so the text at line 128 is wrong! Here we see Hadley run at A1B while IPSL runs at A2. Even if we accept those old CMIP3 scenarios as still valid, those two particular scenarios diverge widely at year 2100. Comparing apples with oranges? Do the authors even know what they present?

Page 5 line 129 and many following instances: for every processing step and every data challenge the authors turn to a standard ArcGIS tool. A reader never finds evaluation of alternatives or citations about how other researchers have confronted and solved these issues. ArcGIS represents an expensive proprietary tool, not suitable for ESSD. Graduate students around the world use open access GIS tools for the simple reason that they can't afford ESRI products. Reviewers and users who likewise prefer R or QGIS will have no ability to test, replicate or confirm procedures and tools used by these authors. Fails completely the open access repeatability expectations of ESSD. Also suggests that the authors lacked skill, interest or motivation to explore other tools. Reads like an ESRI advert.

Page 5 line 132: "average value of the 12 surrounding (ocean) cells." If done in 2-D (e.g. on the same depth or pressure surface), immediately adjacent cells would total 4 or perhaps 8. If done in three dimensions, 'adjacent' cells would total 6 or 26. Please can the authors explain and justify this gap filling? Did they simply chose a value number from ArcGIS raster calculator?

Page 5 line 134: the authors had an old CD of GEBCO data files so they used that? Several more recent, more accurate versions exist.

Page 6 lines 141 to 154: whatever tool ArcGIS has, these authors use it with no questions asked. A substantial literature exists on interpolation techniques for environmental data sets, particularly in meteorology and oceanography. These authors apparently look only in the ArcGIS tools manual. if the authors want to understand gridding techniques - and cautions - for ocean data sets they should look at the series of SOCAT products published in ESSD. Or, if they really want nuts and bolts of gridding and quality control for assimilation into global models, they should wade through the observations for model intercomparisons (Obs4MIPS) literature and guidelines. Normally I would not inflict Obs4MIPS but these authors seem in desperate need of a reality check.

Page 6, line 168: Again, the authors apparently pay no attention to unique or difficult properties of the data at hand but simply push the ArcGIS button for 'band statistics'.

Page 6, lines 172, 173: Compare with the same data sets used as sources? In a world of modern data denial and independent validation techniques, this can't be true?

Page 7, line 178: These authors accept all data inputs as already quality controlled? Nonsense; they have no idea what they have included nor why. By their strange choices - or non-choices, as they simply re-assimilated what others had already compiled - they determined final actual quality of their product in a manner that they apparently neither acknowledge nor understand.

Page 7, line 186: "ensure no significant error was introduced with the interpolation process." This is the best they can say about this entire effort? They have no idea of quality of what they put in, they compile this data of unknown quality into new layers at higher interpolated resolution, then assure readers that they introduced no fresh errors by the interpolation itself. Why do they somehow feel that such a weak effort stands up to the quality and reproducibility standards demonstrated by hundreds of valid data processing efforts documented in ESSD?

Page 7, lines 195 to 204: comparisons by looking at ranges of extreme or maximum values? Ludicrous. Do they not understand statistical techniques for data set intercomparisons and validations? They could learn a lot by looking at almost any ESSD paper.

Page 8, line 222: "derived from a more diverse set of sources". The authors intend this sentence to convey a positive asset of the GMED product. In fact it represents a fatal deficiency because a) they have simply copied what others have already compiled and b) the list of what others have compiled conveys an appalling ignorance of modern ocean data sources which these authors fail to recognise. Obsolete and erroneous source data, no matter how skilfully interpolated, still results in obsolete erroneous data layers.

Page 9, starting from line 249: Finally the reader gains a reality check, of the probable quality of the GMED environmental "surfaces". Readers find welcome cautions from these authors about quality of source data, although note in one sentence these authors disparage "raw observational data" but a few sentences later assure us - erroneously, as their own tables prove - that they have only used highest quality "Level 3" data. 'Level 3' primarily refers to satellite data and misses a large discussion (see matrix often reported by Bates) about maturity and quality of satellite data records. Their citation here covered a very small subset of ocean colour data and is now nearly 20 years of out date.

Page 9, lines 275,276: "verification data indicates that the GMED layers are reliable representations of the source data". More true than the authors understand! Use unreliable source data, apply proprietary statistical interpolation tools, then "validate" by comparison back to the same deficient source data? Garbage in, leads to same garbage reliably represented in the end products. Do the authors not realise this inevitable consequence? Did they not expose their work to competent oceanographers and modellers?

Page 10, line 287: Unfortunately, readers find no evidence that these authors understand the "difference between a pure statistical and a more mechanistic expert-driven approach in interpolation". These authors have certainly not shown any indication that they recognise a need for "expert-driven" guidance or advice before embarking on a "pure" but in fact sadly deficient and essentially useless statistical approach.

Page 10, line 293. Have the authors provided a useful improvement of the land masks? After reading the dismal state of ocean data they used as sources, one wishes for some tangible improvements, e.g. of land masks. As usual, however, authors provide minimal evidence and no citations.

Overall, a completely unsatisfying presentation of a basically flawed (one hesitates to say 'incompetent') effort.

To confirm my impressions about the dismal state of GMED source data, I made a quick scan of ocean data sets available from ESSD. (Authors could do - and arguably should have done - a more careful systematic but fundamentally similar scan.) As these authors apparently recognise (because they attempted to submit their own product) these ESSD-published sources provide up-to-date, well documented, permanently identified, easy access data in standard formats, known and used in the ocean community. My quick scan exposed data sets that cover 60 to 80% of the GMED parameters. These ESSD papers also include many references to other available data. Why the authors did not at least check their AquaMAP and KGS sources against these recent openly-accessible data remains a mystery.

Sources (*with my comments*):

The MAREDAT Special Issue
(*especially*) doi:10.5194/essd-5-109-2013, 25.3.2013, The MAREDAT global database of high performance liquid chromatography marine pigment measurements (*much better than any chl a product you reference*)

SOCAT *(nearly 15 million data points, 1957 to 2014, the definitive way to compile, grid and quality control ocean data)*
doi:10.5194/essd-5-125-2013, A Uniform, Quality Controlled Surface Ocean CO2 Atlas (SOCAT),
doi:10.5194/essd-5-145-2013, Surface Ocean CO2 Atlas (SOCAT) Gridded Data Products
doi:10.5194/essd-8-383-2016, A multi-decade record of high-quality fCO2 data in version 3 of the Surface Ocean CO2 Atlas (SOCAT)

doi:10.5194/essd-5-295-2013, 12.8.2013, Global database of surface ocean particulate organic carbon export fluxes diagnosed from the 234Th technique *(better POC than anything you have)*

doi:10.5194/essd-7-261-2015, 5.10.2015, Vertical distribution of chlorophyll a concentration and phytoplankton community composition from in situ fluorescence profiles: a first database for the global ocean *(instructive about challenges of compiling global ocean data)*

doi:10.5194/essd-8-15-2016, 1.2.2016, A gridded data set of upper-ocean hydrographic properties in the Weddell Gyre obtained by objective mapping of Argo float measurements *(example of the richness and processing of Argo data)*

doi:10.5194/essd-8-165-2016, 28.4.2016, A long-term record of blended satellite and in situ sea-surface temperature for climate monitoring, modeling and environmental studies *(arguably now the definitive SST data set)*

doi:10.5194/essd-8-235-2016, 3.6.2016, A compilation of global bio-optical in situ data for ocean-colour satellite applications *(interesting quality control, covers all your bio-optical parameters)*

GLODAP *(the definitive ocean data set, with extensive well-documented quality control)*
doi:10.5194/essd-8-297-2016, 15.8.2016, The Global Ocean Data Analysis Project version 2 (GLODAPv2) – an internally consistent data product for the world ocean
doi:10.5194/essd-8-325-2016, 15.8.2016, A new global interior ocean mapped climatology: the 1°× 1° GLODAP version 2

doi:10.5194/essd-8-531-2016, 20.10.2016, Global ocean particulate organic carbon flux merged with satellite parameters *(for ESA CCI, much-used and well-documented)*

doi:10.5194/essd-8-679-2016, 29.11.2016, C-GLORSv5: an improved multipurpose global ocean eddy-permitting physical reanalysis *(interesting re-analysis product, instructive on how they assimilate sparse data)*

https://doi.org/10.5194/essd-10-251-2018, 6.2.2018, Photosynthesis–irradiance parameters of marine phytoplankton: synthesis of a global data set *(another good data assembly example, highly relevant to your intended uses)*

https://doi.org/10.5194/essd-5-311-2013, A long-term and reproducible passive microwave sea ice concentration data record for climate studies and monitoring *(sea ice data at the highest quality level)*

---

## Referee Comment (RC2) · F. Benedetti (Referee) · 4 Oct 2018

Comments for Basher et al. (2018) - ESSD Discussion

The present manuscript aims to present a novel digital atlas of environmental (meaning physical, chemical, biogeochemical) climatologies, from which scientists may download numerous environmental layers that are typically used for developing spatial statistical models, such as species distribution models (SDMs). The authors did a fine job in compiling many published datasets, and gathering all of them in a homogeneous and central atlas. Consequently, the Global Marine Environment Dataset (GMED) is the online platform with the widest range of environmental layers. The GMED proposes

environmental data at a finer spatial resolution compared to previous comparable at-lases (mainly AQUAMAPS, MARSPEC and Bio-ORACLE). The GMED also supplies past and future fields for some of the environmental layers (temperature, salinity, ice cover), thus allowing the community to quickly test long-term changes in species distri-butions and diversity. Consequently, it might attract marine ecologists aiming to easily model the niches and distributions of marine taxa, whether those are benthic, pelagic, coastal or inhabiting offshore conditions. But that might also be an issue. In spite of the added value of the dataset might present, I have identified some major points that may help improve the completeness, the quality of the atlas and the manuscript. I will now give my step-by-step review of the manuscript and data access based on the ESSD review guidelines. Then I will detail my major concerns and comments regarding the GMED itself.

Please see my full comments in the .pdf file attached as supplementary.

Yours,

Please also note the supplement to this comment:
https://www.earth-syst-sci-data-discuss.net/essd-2018-64/essd-2018-64-RC2-supplement.pdf

———————————————————

[Figure]

**Supplement:**

**Comments for Basher et al. (2018) - ESSD Discussion**

The present manuscript aims to present a « novel » digital atlas of environmental (meaning physical, chemical, biogeochemical) climatologies, from which scientists may download numerous environmental layers that are typically used for developing spatial statistical models, such as species distribution models (SDMs). The authors did a fine job in compiling many published datasets, and gathering all of them in a homogeneous and central atlas. Consequently, the Global Marine Environment Dataset (GMED) is the online platform with the widest range of environmental layers. The GMED proposes environmental data at a finer spatial resolution compared to previous comparable atlases (mainly AQUAMAPS, MARSPEC and Bio-ORACLE). The GMED also supplies past and future fields for some of the environmental layers (temperature, salinity, ice cover), thus allowing the community to quickly test « long-term » changes in species distributions and diversity. Consequently, it might attract marine ecologists aiming to easily model the niches and distributions of marine taxa, whether those are benthic, pelagic, coastal or inhabiting offshore conditions. But that might also be an issue as I will develop below.

In spite of the added value of the dataset might present, I have identified some major points that may help improve the completeness, the quality of the atlas and the manuscript. I will now give my step-by-step review of the manuscript and data access based on the ESSD review guidelines. Then I will detail my major concerns and comments regarding the GMED itself.

1. The data presented consist of a compilation of pre-existing datasets so the data themselves are not « new » , but the atlas is clearly more exhaustive than previous and comparable ones, even though some of the data used are clearly outdated (but see major comments below). Also, the data presented here have been interpolated to follow a higher resolution grid, so they represent an improvement for end users (SDM users). I do like that the authors added variables such as distance to land, or to closest port, because these often have to be calculated separately and can be very useful to account for sampling biases in marine species distributions. Therefore, I agree that this dataset could be useful for future studies. Although I consider the methods description to be thorough, I do miss proper uncertainty estimates in the layers provided online, and especially for the past and future environmental layers (but see major comments below). For controlling the quality of the data, the authors completely rely on the controls undergone y other authors for the first publication of the data compiled. Furthermore, the perform some sort of completely circular cross-validation to control the output of their interpolation. Proper quality control would at least require some independent data. Therefore, they do not really perform « quality control » in my humble opinion. As a result, key information are missing for the reader about the way the environmental layers were developed initially. Here, it is not sufficient to simply state that « *All of the primary datasets used in the GMED compilation had undergone quality control checks by the primary data collectors and processors* ».

2. I have identified some gaps in the niche modeling literature (lines 53-59) that I would like the authors to address carefully because I think it may have lead them to forget important predictors in their atlas (but see major comments below). I would also like the authors to mention the update of the Bio-ORACLE v2 dataset (Assis et al., 2018 - DOI: 10.1111/geb.12693) in their manuscript. Yet I acknowledge it might have been published after the authors finished their atlas.

3. The dataset is easily accessible via the online portal. I had no problem downloading, unarchiving and then reading the data with R. The files are encoded in ASCII, which is easy to read with the « *raster* » R package (Hijmans (2017). *raster: Geographic Data Analysis and Modeling*. R package version 2.6-7. https://CRAN.R-project.org/package=raster). I honestly have no experience with reading ASCII tables with other commonly-used languages, such as Matlab or python, but I am convinced the people concerned will not have too much trouble with that. The online dataset seems complete regarding to what is described in the manuscript. However, I am missing error estimates and/or quality flags in the data tables provided. For now, the only « quality flag » consists in the existence of a cropped version of the layers (at 70°N and S because of the satellite data). I would like to know whether it would be possible to add uncertainty estimates (linked to initial observations density biases, or model uncertainties when models are part of the process) to the data tables so one can identify where the less reliable data are geographically located (especially for biogeochemical and future fields)? Nevertheless, I would say the data cleaning, treatment and comparison to previous similar datasets are adequate: what the authors did is clear and well described. One of the authors' main claims is that the finer resolution of their layers should lead to more reliable SDMs compared to previous products. Although I agree this should be the case, a formal test of this assertion is needed (like developing a few standard SDMs for a virtual species based from the present data and then compare them to SDMs built from the previous atlases). But I am not sure this is within the scope of an ESSD paper, which focuses on the data itself.

4. Overall, I find the dataset to be usable in its current format. Maybe others would prefer to be able to download it as a text file (.txt or .csv). The metadata are provided in the appendices and can easily be found online (http://gmed.auckland.ac.nz/layersd.html). The issue when compiling pre-existing datasets is that one may simply refer to the original publication of the data for the full metadata. These are not provided in the present manuscript but the authors do refer to the original publications and website (like in any other publication of this nature, see Tyberghein et al., 2012). Overall, the language and the figures are of good-quality in my opinion. However, I strongly recommend that the authors provide a quantitative scale and the appropriate units with the maps, instead of the rather arbitrary « low » and « high ». I think there is a typo in the caption of Figure A53: *Temperature 1AB Scenario*? This really looks like a salinity map. Also, in Figure 1, the authors need to clearly indicate which steps they performed themselves. Indeed, they did not compile all the satellite/ model/

observation-based products that were used to implement the previous atlases. Therefore, from what I understood, the data processing steps actually performed by the authors are those indicated by the 4th and 5th arrows (after « Raster Grid »). To summarize: the authors interpolate the older layers on a new and finer grid, and then evaluate the interpolation's output by computing variation coefficient and standard error between those and the initial layers. By doing so, the authors do not claim to actually control the quality of the data, but rather the « *interpolation quality* ». I think the authors are right in stating so, but I do find the process a bit circular…

To conclude, I do think the data presented here are complete and could be useful to quickly run and test some SDMs. It does comprise a very comprehensive compilation of different environmental variables that are commonly used as predictors in species distribution modeling. However, I cannot conclude that the data presented here are « unique ».

**Major Comments**

**1) Mixed-layer depth and variables averaged over the mixed-layer?**

In the introduction (l. 53-59), the authors rightfully state that SDMs have been relatively less used for studying marine taxa compared to their terrestrial counterparts. Then, they mention the marine groups that have been studied through SDMs with the associated literature. Here they fail to mention the recent (and less recent) studies that performed niche modeling for the marine plankton (both phytoplankton and zooplankton), apart from Bentlage et al. (2013) whom quickly performed SDMs using climatologies from before 2005…Haphazardly, you should mention some of the following studies:

Beaugrand, G. & Helaouët, P. (2008) Simple procedures to assess and compare the ecological niche of species. *Marine Ecology Progress Series*, **363**, 29-37.

Beaugrand, G., Edwards, M., Brander, K., Luczak, C. & Ibanez, F. (2008) Causes and projections of abrupt climate-driven ecosystem shifts in the North Atlantic. *Ecology Letters*, **11**, 1157-1168.

Beaugrand, G., Lenoir, S., Ibañez, F. & Manté, C. (2011) A new model to assess the probability of occurrence of a species based on presence-only data. *Mar. Ecol. Prog. Ser*, **424**, 175-190.

Reygondeau, G. & Beaugrand, G. (2011) Future climate-driven shifts in distribution of *Calanus finmarchicus*. *Global Change Biology*, **17**, 756-766.

Irwin, A.J., Nelles, A.M. & Finkel, Z.V. (2012) Phytoplankton niches estimated from field data. *Limnology and Oceanography*, **57**, 787-797.

Beaugrand, G., Mackas, D. & Goberville, E. (2013) Applying the concept of the ecological niche and a macroecological approach to understand how climate influences zooplankton: advantages, assumptions, limitations and requirements. *Progress in Oceanography*, **111**, 75-90.

Chust, G., Castellani, C., Licandro, P., Ibaibarriaga, L., Sagarminaga, Y. & Irigoien, X. (2014) Are *Calanus* spp. shifting poleward in the North Atlantic? A habitat modelling approach. *ICES Journal of Marine Science: Journal du Conseil*, **71**, 241-253.

Pinkernell, S. & Beszteri, B. (2014) Potential effects of climate change on the distribution range of the main silicate sinker of the Southern Ocean. *Ecology and Evolution*, **4**, 3147-3161.

Villarino, E., Chust, G., Licandro, P., Butenschön, M., Ibaibarriaga, L., Kreus, M., Larrañaga, A. & Irigoien, X. (2015) Modelling the future biogeography of North Atlantic zooplankton communities in response to climate change. *Marine Ecology Progress Series*, **531**, 121-142.

Brun, P., Vogt, M., Payne, M.R., Gruber, N., O'Brien, C.J., Buitenhuis, E.T., Le Quéré, C., Leblanc, K. & Luo, Y.W. (2015) Ecological niches of open ocean phytoplankton taxa. *Limnology and Oceanography*, **60**, 1020-1038.

Barton, A.D., Irwin, A.J., Finkel, Z.V. & Stock, C.A. (2016) Anthropogenic climate change drives shift and shuffle in North Atlantic phytoplankton communities. *Proceedings of the National Academy of Sciences*, **113**, 2964-2969.

Brun, P., Kiørboe, T., Licandro, P. & Payne, M.R. (2016) The predictive skill of species distribution models for plankton in a changing climate. *Global Change Biology*, **22**, 3170-3181.

Benedetti, F., Vogt, M., Righetti, D., Guilhaumon, F. & Ayata, S.-D. (2018) Do functional groups of planktonic copepods differ in their ecological niches? *Journal of Biogeography*, **45**, 604-616.

But more importantly: in several of these publications, SDMs were developed using mixed-layer depth (MLD) as a predictor, or other variables (PAR, SST, Chlorophyll concentration) integrated over the mixed layer. The noteworthy paper of Brun et al. (2015) even identified MLD as the most important variable for modeling the niches of phytoplankton species. MLD greatly contributes to the temperature, light conditions and nutrients dynamics perceived by the plankton, the basis of evert marine food-web. Its role in controlling Ocean-Amosphere heat fluxes and in shaping bloom dynamics has been studied for decades now. It should always be considered as a potential predictor even for fishes and/or top predators because of its probable effect through bottom-up processes. Overall, MLD is arguably one of the most important oceanographic variable so I was extremely surprised not to see it among the variables compiled. Why is that?

I highly recommend that the authors add at least one MLD product to their atlas. The most recent one I can think of would be: Holte, J., Talley, L.D., Gilson, J. & Roemmich, D. (2017) An Argo mixed layer climatology and database. *Geophysical Research Letters*, **44**, 5618-5626. Which can be found here: http://mixedlayer.ucsd.edu/

I also encourage the authors to compute mixed-layer averages for several other variables such as temperature, irradiance, salinity, nutrients concentrations, Chlorophyll-a concentration etc. It seems like the authors do not benefit from the recent wave of Argo floats data. Which brings me to my second major point.

**2) Outdated data sources.**

While reviewing the sources of the data compiled, I was surprised that many of the layers still rely on data from the World Ocean Atlas of 2009, or from the Sea-WiFS satellite era. Since then, the World Ocean Atlas has undergone not one but two updates (it is currently at the WOA 2013v2 stage: https://www.nodc.noaa.gov/OC5/woa13/) and the MODIS-Aqua sensor has been operational since 2002. The WOA 2013v2 provides monthly/ seasonal/ annual climatologies at a 5°, 1° and sometimes 1/4° resolution, with standard depth levels, and with detailed and proper quality controls. I am very surprised the authors did not take the time to assimilate these layers since they are widely known in the oceanographic community.

For other chemical and biogeochemical variables, way more recent and valuable datasets can be found in ESSD:

https://www.earth-syst-sci-data.net/7/261/2015/essd-7-261-2015.pdf

https://www.earth-syst-sci-data.net/8/325/2016/essd-8-325-2016.pdf

https://www.earth-syst-sci-data.net/8/297/2016/essd-8-297-2016.pdf

https://www.earth-syst-sci-data.net/8/383/2016/essd-8-383-2016.pdf

And, of course, updated and controlled observations and re-analyses can be found on the Copernicus data portal: http://marine.copernicus.eu/services-portfolio/access-to-products/

I am a bit uncomfortable as I do not want to dismiss all of the work carried out by the authors, but I must *strongly* encourage them to go through all these data products and update their data sources. Otherwise the community is just given recycled and outdated data products that do not not reflect the state of the art, nor the efforts of the climate and ocean scientists. This point is also valid for the past and future environmental layers provided in the GMED, which brings me to my third major comment.

**3) Fields of future environmental conditions.**

One of the reasons why SDMs got so popular in the last 20 years is because they allow to handily explore temporal changes in species distribution, and therefore diversity, following climate change (greenhouse gas emissions actually) scenarios. Knowing that, the authors added some predictions of SST, SSS, seabed temperature and salinity, primary productivity and ice concentration. This could have been interesting had the scenarios not been completely outdated. Indeed, the data compiled here were issued for the 4[th] AR of the IPCC (CMIP3 exercice). I do not believe the authors are unaware of the existence of the IPCC's 5[th] AR which presents Representative Concentration Pathways (RCPs) that are now the standard when it comes to model climate change impacts. I know, from my own experience, that RCPs data are not always available for regional models, but this is definitely not the case for the global ocean. Proof is that even the latest version of Bio-ORACLE (Assis et al., 2018 - DOI: 10.1111/geb.12693) provides RCPs outputs, with some uncertainty estimate across AOGCMs. Why did the authors not consider the latest standards?

The authors fail to provide crucial information about model set-up, calibration, configuration, validation, bias correction…The two links provided in the references below Table 1 are not functional. Do the layers presented correspond to the absolute fields obtained for the 2090-2100 period? Or to model biases between the contemporary period and the end-of-century period that were added to the observation-based climatologies? What are the uncertainties within each projection? And then between projections? Why did the authors rely on just two models (IPSL and HadCM3) among all the existing ones? Why are future surface temperature and salinity given for two emission scenarios but not seabed temperature? What is the model configuration that generated the future PP product? There are *tremendous* uncertainties across the suite of coupled ecosystem models that can provide biogeochemical projections (just have a look at Laufkötter et al., 2015 - doi:10.5194/bg-12-6955-2015), and this is well known in the community. I am sorry but these future layers cannot be used as of now. The choice of the climate model can make up a significant part of the uncertainties in SDM-based climate change predictions. Please see:

Diniz-Filho, J.A.F., Bini, L.M., Rangel, T.F.L., Loyola, R.D., Hof, C., Nogués-Bravo, D. & Araújo, M.B. (2009) Partitioning and mapping uncertainties in ensembles of forecasts of species turnover under climate change. *Ecography*, **32**, 897-906.

Buisson, L., Thuiller, W., Casajus, N., Lek, S. & Grenouillet, G. (2010) Uncertainty in ensemble forecasting of species distribution. *Global Change Biology*, **16**, 1145-1157.

Garcia, R.A., Burgess, N.D., Cabeza, M., Rahbek, C. & Araújo, M.B. (2012) Exploring consensus in 21st century projections of climatically suitable areas for African vertebrates. *Global Change Biology*, **18**, 1253-1269.

The risk is that young scientists might implement SDMs based on the contemporary layers provided the GMED, with the default settings user-friendly modeling platforms, and simply project those in the future conditions, without any prior knowledge about the way the data were produced…Knowing the data you use (meaning understanding where it comes from, its limitations and quality, the uncertainties associated) is a crucial part of any modeling experiment, and species distribution modeling is not an exception. And this brings me to my fourth and final major comment.

**4) Compilation of environmental predictors takes time…and maybe it should do so.**

The manuscript's abstract stipulates the following: « *Marine environmental datasets available for species distribution modelling (SDM) have different spatial resolutions and are frequently provided in assorted file formats. This makes data assembly one of the most time-consuming parts of any study using multiple environmental layers for biogeography visualization or SDM applications* ». I assume this motivated the authors (but others also) to implement user-friendly and publicly available compilations of environmental data to facilitate (and accelerate) the process. This could make sense when the quality of the data used and therefore the whole procedure is not affected. But I am confident this cannot be the case when using the present GMED (because of all the previously mentioned reasons).

Instead, I argue it is crucial that students and young scientists take the time that is required to: (i) review the environmental datasets available; (ii) thoroughly examine their origins (metadata), advantages and limitations; and (iii) investigate how alternative choices in the environmental data impact final SDMs outputs. How are they supposed to perform state of the art modeling if they do not even understand the ins and outs of the data they use? Data assembly is time-consuming because it is the process that will determine data quality and thus the quality of any SDM projection. The identity and resolution of the environmental predictors available and suitable for a niche modeling exercice depend on its goals (testing for niche overlap, species distribution visualization, climate change impacts projections), and the type of biological data available (abundance, presence only, presence-absence etc.). I totally get that our community is experiencing increasing pressure because of competition for fundings, pressure to publish, and demands from stakeholders to provide climate change predictions, and therefore tries to gain time when possible. But simplicity and easy-to-use products should not take over the quality that any scientific experiment is entitled to.

To conclude, I would like the authors to know that I am truly sorry that I could not provide more positive comments. I hope they will take it as an encouragement to deeply re-organize and actualize their data so they comply with the quality required for any ESSD dataset. I encourage them to work more closely with oceanographers and climate scientists to help them find better and updated marine environment data.

**Dr. Fabio Benedetti**
**ETH Zürich, D-USYS, IBP, UP Group.**
**On the 03/10/2018.**

---

## Author Comment (AC1) · 3 Jan 2019

**Response to Reviewer 1 comments**

Review ESSD-2018-64, Marine Environment Data Sets

Demonstrating minimal knowledge of ocean data, the authors have relied on obsolete sources and a proprietary (ESRI ArcGIS) software to 'produce' a series of unreliable data layers. Their product demonstrates neither quality nor reproducibility, the hallmarks of a good (ESSD) data set. Surprisingly, these authors seem substantially unaware of fundamental deficiencies.

Should we as readers hold these authors accountable for the very poor quality of their source data? Yes, in this case! The authors represent themselves (below) as competent data processors. They demonstrate their confidence by submitting their new compilation of previously un-reviewed data products for review and public comment in a prominent data journal.

These authors imply that based on "recent experience with developing compatible, comprehensive, environmental layers" they have "developed an extensive on-line repository of marine environmental data layers." Assembled, perhaps. Developed, no. Later, these authors admit that all data sets used for GMED "had undergone quality control checks by the primary data collectors and processors". These authors have applied a canned statistical interpolation (from ArcGIS), used some additional ArcGIS tools to check whether their interpolation introduced errors, and then served 60 layers of environmental … what? Reprocessed, improved and consistent data? Or, garbage amplified? Because these authors demonstrate minimal understanding of the quality, temporal extent or spatial coverage of their source data - as evidenced by the appalling array of old and obsolete sources - this reader regards their product as garbage, and old garbage at best. Apologies for strong language but one feels a need to 'wake up' these authors to a world of ocean data outside of their narrow view.

If in fact the ecology community accepts and relies on obsolete deficient ocean data sources then these authors may have done us an unintended favour by exposing the vast gap remaining between what oceanographers use and what ecologists use.

We appreciate the comments from the reviewer and important suggestions to improve the manuscript and the overall data compilation effort described in the manuscript. We do agree many of the primary data sources mentioned as the raw dataset used for GMED may be outdated but as we like to think GMED as a living data resource instead of a data snapshot we have all the intentions to update the data layers with new sources as the opportunity arises in future  We also like to mention that all the secondary datasets (e.g., BioOracle, Aquamaps) used to compile GMED datasets are very widely used by the ecological modelers and there are hundreds of peer-reviewed publications citing these data sources. These resources have been demonstrated to be useful for research in ecology, ecosystems, and biogeography. Moreover, as mentioned in the Appendix, World Ocean Atlas from NOAA was the primary source dataset for most of these data products. Therefore, we respectfully disagree with the comment that we used 'unverified' data as a source for compiling the GMED data layers. For some other purposes the reviewer may consider the data to be out-of-date or not the best possible but that will always be the case for any datasets "fitness for purpose". We believe there is a good opportunity to address data requirement gap between the oceanographers and ecologists exists with GMED and we will continue to make our best effort to improve the data layers in GMED by integrating uncertainty estimates whenever possible.

Data Access

The GMED url works, leads one to a useful landing page. The GMED version 2.0 link, at the upper right of that landing page, does not work. The 'DataSets' link does work and presumably leads to the layers discussed in the manuscript.

For this reader the figshare doi did not work in any configuration. ESSD generally does not use figshare for exactly these reliability issues. If the authors have a GMED snapshot they can deposit at figshare, they could easily - and more usefully - deposit it at Zenodo, Pangaea, 4TU, etc? Not reliable as presented, needs a change.

Thank you for the suggestion. We have also submitted our datasets to Zenodo as an additional resource. The archive could be accessed from https://doi.org/10.5281/zenodo.1491933
We have also checked the Figshare link and it appears to work ok. We note that it has been used extensively by many publications in recent years and is preferred by open source journals like PloS.

Page 3 starting line 59: I think you mean to say 'Differences in applications of SDM to marine environments compared to terrestrial environment's include fewer observation records, extensive spatial-temporal variability of oceanic environments, and complexities in processing marine environmental data for SDM purposes'? But you would need to back up this statement with facts? From a carbon point of view, we might know less about land sources and sinks than we know about ocean sources and sinks? The SDM community takes a different view than the biogeochemical carbon cycle community? The authors seem blissfully ignorant of these larger issues, as perhaps they should, but to publish their data in the same journal that publishes (land and ocean) carbon cycle work they need to assure readers of an awareness of larger context?

We have removed the comparison to terrestrial environments as we recognize data are limited everywhere in some way.

Page 3 line 61: the phrase "spatial-thermal" seems strange here (I can find it used in other marine ecology papers) and I doubt the supposed differences marine to terrestrial. Spatial: ocean has surface to deep, land has sea level to alpine. Both land and ocean have strong latitudinal gradients and high variability in immediate coastal zones. If you truly mean thermal rather than temporal, ocean has -1 to 32C, land (even if we exclude ice) has -40 to +45), with ocean temperatures much more consistent than those of land. Ocean proper has very small range of salinity compared to large range of humidity on land. Ocean has no light to full light but so does land. Both have distinct daily and seasonal patterns that strongly impact biology. If you mean to imply range of habitats, we don't know much about 3-D patterns in the full-depth ocean but we know that land includes a wide array of habitats. You will need to help the reader understand what you mean here because as written you have allowed a too-wide variety of misconceptions.

By the term 'spatio-thermal' we meant the 3D nature of the ocean environment where based on spatial distance and depth the water temperature can change dramatically (in tropics) or remains relatively consistent (in polar-regions). We will add the clarifying sentence in the revised manuscript.

Page 3 lines 65 to 67: two biggest improvements in ocean monitoring probably come from ocean colour by satellite (e.g. MODIS and follow-ons) and globally-distributed profiling floats (e.g ARGO and, one hopes, soon bio-ARGO). (Some will also argue for the strong impact from GRACE.) This discussion of ocean observations seems to miss these crucial developments?

Text in the lines 65-67 only to let a reader know about the methods broadly used to collect marine environmental data related to GMED data layers, without going into specifics about the collection methods (i.e. discussing ARGO/MODIS in details).

Page 3 line 70: many other groups have done a substantially better job at compiling and quality-controlling ocean data sets than Tyberghein and the bio-ORACLE crowd. To understand ocean properties from a biogeochemical or ecological perspective, a modern unbiased user would probably not start from bio-ORACLE? This entire discussion (lines 70 to 75) seems impossibly simple and surprisingly ignorant of other substantial efforts. Large literature exists on ocean SST.

Substantial literature exists on ocean Chl A, much of it in ESSD, and extending quite far back before 1997. Have these authors never heard of GLODAP? Remote areas from an ocean data point of view might point to the central Pacific rather than (or as well as) the Arctic? This discussion pretty much convinced this reader that these authors lack working knowledge of current ocean data efforts.

Thank you for the suggestions. We understand reviewer's perspective, bio-oracle and AquaMaps were referenced in our manuscript as they are the two most widely used data resource for modeling by marine ecologists to date. We are indeed aware of the GLODAP and ARGO float datasets. As

GMED is a global dataset, we tried to incorporate data sources which have the most extensive global coverage with limited spatial data gaps. We understand data exists before the year 1997 (Wernand et al 2013, PlosOne) for ChlA but as SeaWIFS was the first satellite to provide extensive global coverage of ChlA data from 1997, the year was mentioned in the manuscript to give the readers some perspective about potential time frame for data availability in wider context.

Page 4 lines 78,79 "continuous, global, layers for such variables are predicted from ocean circulation models and by extrapolation of in situ sample data." A very large and very active community of ocean observationalists and modelers would strongly disagree with this glib uninformed attempt at summary. These authors provide no citations to back their contentions?

Thank you for pointing that out. We have added citations and revise the text to clarify this confusion.

Page 4 lines 80, 81: resolution of ocean models represents a hot and active topic, with variety of useful approaches driven by both physics (eddy-resolving, mixing parameterisations) and biology. The authors seem aware of none of that current vital work. Redfern 2006 represents a weak, almost irrelevant, reference.

Thank you for the comment. We recognize that more research is needed and underway to improve ocean model resolution.

Page 4 line 85: www.worldclim.org represents a weak, slow, out-of-date data source? Who uses it? The site provides no quality control nor uncertainties, they only repackage data from other unspecified sources. No climate modeller uses these products. Everything in one format but at what cost? Data only goes to year 2000? I wouldn't touch it. GMED wants to emulate this?

Worldclim has been widely used by terrestrial ecologists (have over 3000 citations) for many years. We mentioned it to highlight a need of a similar resource data marine ecologists. We understand that there is an opportunity exists to utilize the vast amount of oceanographic data available to users but they remained under-used due to the complexity of data processing steps involved that discourage most of the researchers to use them. With GMED our main goal is to increase data accessibility to the wider ecological user community who may not have the expertise to process all these data coming from a variety of sources.

Nice to see (lines 99,100) GBIF and OBIS mentioned but those do not include physical or chemical parameters. Typical of all GEO efforts, GEOBON provides nothing itself but only tries to ride on data sets produced and quality controlled by others.

One of Author (MJC) has been closely involved in OBIS and GEO BON since their earliest days, and GBIF since 2001. The referee is not correct in their understanding of these databases. GEO BON does not duplicate the good work by GBIF and OBIS who are leading participants in GEO BON, and many GEO BON members publish into these databases and make data available in other ways. Many datasets in OBIS and thus GBIF do include in-situ environmental data, particularly depth, temperature, and salinity but also others. There new data schema based on environmental samples (rather than biological specimens) now allows considerably more such data to be published.

Based on this introductory discussion a reader comes to doubt whether the authors have any knowledge of ocean observational (in situ, satellite or re-analysis) data sets.

Page 5 lines 115 to 117: basically, land-masking, then interpolation using standard ArcGIS tools, followed by comparisons with data sets already dismissed as deficient. Somehow this represents a useful contribution?

Figure 1 of terrible resolution, basically not even readable. Better figure in Supplement but why should reader need to look in Supplement to find a readable first figure of the paper?

A good quality version of Figure 1 is available at page 27 of the manuscript.

Above, the authors dismissed AquaMAP and KGS Mapper as inadequate due to (apparently)

complexity.   Here they use the same two sources as their primary data sources?  Later they then compare back to AquaMap and KGS Mapper.  Can one imagine a more circular process, with less actual quality control contribution?

We respectfully disagree with the comment. Both of the mentioned data repositories used WOD (WOD 2005 version) as their primary data source. With GMED as we mentioned in the manuscript, we added more data layers with improved spatial resolution, quality (with more refined land mask) and accessibility. As a live data repository, we do have plans to update the GMED with most recent WOD 2018 dataset in near future.
We do not dismiss AquaMaps, KGS Mapper or other resources as the referee states. Rather we add value to them by providing additional functionality through GMED.

Page 5 line 128: Jungclaus 2006  - one run of one version of ECHAM5 at T63 downloaded from WDC, now more than 10 years out of date? Run at SRESA1B - an optimistic (and now also out-of-date) emission scenario?  Do the authors somehow need to prove their ignorance of global ocean and climate data sets?  Kaschner references merely a back-door route back to AquaMAPS.

Page 5 line 128: Kaschner et al 2013 url link does not work.  And no wonder: 5 years since last access?  Why insult the reader with useless out-of-date links?

Thank you for pointing that out. The AquaMaps URL will be updated to the new address (http://www.aquamaps.org/download/main.php). We will update the link in the revised manuscript.

In Table 1 authors claim to have used climate projections from CMIP3, one iteration of the UK Met Office model - with no emissions scenario information (as it turns out, already chosen by bioORACLE according to Table S1) - and one from IPSL, configuration unknown but evidently with a full depth green ocean and a coupled ice model, run at A2 (again from Table S1, already selected by and available from AquaMaps).  Does reader follow the text (line 128) or the table? No guidance and too little information for anyone to attempt to replicate and confirm.  Do the authors even know what they have used?  Panels on the GMED landing page, for year 2100, credit UK Met / Hadley and IPSL, so the text at line 128 is wrong!  Here we see Hadley run at A1B while IPSL runs at A2. Even if we accept those old CMIP3 scenarios as still valid, those two particular scenarios diverge widely at year 2100.  Comparing apples with oranges?  Do the authors even know what they present?

At Line 128 we indicated the primary data source of the projected data layer for the readers, BioOracle/AquaMaps were the secondary sources from where we extracted the data for GMED. Also, details about all the layers are given in the Appendix S1 and always most updated information about data sources and layer details are available from the website at http://gmed.auckland.ac.nz/layersd.html. We do have plans to update the projection layers with most recent RCP's used by IPCC AR5, and will add a warning notice in the website to infirm users the availability of new RCP's for the future data layers form other sources, user could make informed decision whether to use these old layers or wait until we update the data with newer data.

Page 5 line 129 and many following instances: for every processing step and every data challenge the authors turn to a standard ArcGIS tool.  A reader never finds evaluation of alternatives or citations about how other researchers have confronted and solved these issues.  ArcGIS represents an expensive proprietary tool, not suitable for ESSD. Graduate students around the world use open access GIS tools for the simple reason that they can't afford ESRI products. Reviewers and users who likewise prefer R or QGIS will have no ability to test, replicate or confirm procedures and tools used by these authors.  Fails completely the open access repeatability expectations of ESSD. Also suggests that the authors lacked skill, interest or motivation to explore other tools.  Reads like an ESRI advert.

We respectfully disagree with the reviewer. The methodology followed to create GMED is described in Figure 1 and the process could be easily replicated in QGIS (similar tools exist in QGIS, students just need to search the web to find the right tool). It was not possible to use R when we created GMED first in 2013, R was stalled to handle 5 million points in a desktop computer for any extraction or interpolation tasks.

Page 5 line 132: "average value of the 12 surrounding (ocean) cells." If done in 2-D (e.g. on the same depth or pressure surface), immediately adjacent cells would total 4 or perhaps 8. If done in three dimensions, 'adjacent' cells would total 6 or 26. Please can the authors explain and justify this gap filling? Did they simply chose a value number from ArcGIS raster calculator?

Gap filling was processing using the raster calculator with Focal Statistics tool of ArcGIS. Details about the statistics feature could be viewed at http://desktop.arcgis.com/en/arcmap/10.3/tools/spatial-analyst-toolbox/how-focal-statistics-works.htm. We used the circular (2 cell method, which measured the mean value of to 12 surrounding cells in total) neighborhood method to calculate the missing value of pixels.

Page 5 line 134: the authors had an old CD of GEBCO data files so they used that? Several more recent, more accurate versions exist.

The depth layer will be updated with the most recent version of gebco in next update of GMED.

Page 6 lines 141 to 154: whatever tool ArcGIS has, these authors use it with no questions asked. A substantial literature exists on interpolation techniques for environmental data sets, particularly in meteorology and oceanography. These authors apparently look only in the ArcGIS tools manual. if the authors want to understand gridding techniques - and cautions - for ocean data sets they should look at the series of SOCAT products published in ESSD. Or, if they really want nuts and bolts of gridding and quality control for assimilation into global models, they should wade through the observations for model intercomparisons (Obs4MIPS) literature and guidelines. Normally I would not inflict Obs4MIPS but these authors seem in desperate need of a reality check.

We agree the interpolation techniques used by the oceanographic community is different than the interpolation techniques commonly used by ecologists. The gridded methods described in SOCAT paper is very appropriate if you have an in-situ dataset with a defined type of cruise and observation numbers but that we believe it might not be appropriate for our dataset where we only have a value per grid cell from a single in-situ/modelled source.

Page 6, line 168: Again, the authors apparently pay no attention to unique or difficult properties of the data at hand but simply push the ArcGIS button for 'band statistics'.

We note the reviewer's antipathy to ArcGIS, but it is nevertheless the leading world GIS software with a range of tools used daily by thousands of people, notably government bodies and researchers, worldwide. We are aware of the shortcomings of some of its tools, and have worked with the manufacturers to correct them (e.g., Costello M.J., Smith M., Fraczek W. 2015. Correction to surface area and the seabed area, volume, depth, slope, and topographic variation for the world's seas, oceans, and countries. *Environmental Science and Technology*. DOI: 10.1021/acs.est.5b01942. However, any software is subject to errors and in this case there is excellent support from the manufacturer to correct errors.

Page 6, lines 172, 173: Compare with the same data sets used as sources? In a world of modern data denial and independent validation techniques, this can't be true?

We understand that a minimum test of any analysis is that the outputs accurately reflect the input data. This is to test for errors in data processing and analysis.

Page 7, line 178: These authors accept all data inputs as already quality controlled? Nonsense; they have no idea what they have included nor why. By their strange choices - or non-choices, as they simply re-assimilated what others had already compiled - they determined final actual quality of their product in a manner that they apparently neither acknowledge nor understand.

We only use authoritative sources which have documented their quality control procedures (Table 1 of the MS which listed the primary data source while Table S1 in the supplementary listed the source from where we compiled the data from, i.e. AquaMaps/KGS/Bio-Oracle/MERPECS. All these sources has publications which

listed their quality control procedure of the respective dataset in details). It is thus not true that we have not acknowledged the sources and their procedures. Using these resources is an additional check in their quality, and they have been used in many studies, including our own.

Page 7, line 186: "ensure no significant error was introduced with the interpolation process." This is the best they can say about this entire effort? They have no idea of quality of what they put in, they compile this data of unknown quality into new layers at higher interpolated resolution, then assure readers that they introduced no fresh errors by the interpolation itself. Why do they somehow feel that such a weak effort stands up to the quality and reproducibility standards demonstrated by hundreds of valid data processing efforts documented in ESSD?

In a compilation of this kind, it is necessary to trust the scientific integrity of the source data and its processing. All the data come from scholarly and reputable sources. We note the reviewer is highly critical of these. What we have done is make these datasets more accessible to the ecological community and in standardized formats. As with any data used in any study, users need to be congnisant of the limitations of the data. However, the original data and GMED, have not been used in numerous analyses and found to be very useful in examining global scale environmental and biogeographic patterns.

Page 7, lines 195 to 204: comparisons by looking at ranges of extreme or maximum values? Ludicrous. Do they not understand statistical techniques for data set intercomparisons and validations? They could learn a lot by looking at almost any ESSD paper.

We are surprised at the language of the referee. A simple first step in comparing datasets includes looking at the ranges of the data.

Page 8, line 222: "derived from a more diverse set of sources". The authors intend this sentence to convey a positive asset of the GMED product. In fact it represents a fatal deficiency because a) they have simply copied what others have already compiled and b) the list of what others have compiled conveys an appalling ignorance of modern ocean data sources which these authors fail to recognise. Obsolete and erroneous source data, no matter how skilfully interpolated, still results in obsolete erroneous data layers.

We disagree that the data are obsolete or erroneous as the referee alleges. If this is the case then this merits a well-argued critique in a separate paper because these datasets are, and have been, used by probably thousands of researchers. Furthermore, historic data is increasingly of interest for studying changes over time.

Page 9, starting from line 249: Finally the reader gains a reality check, of the probable quality of the GMED environmental "surfaces". Readers find welcome cautions from these authors about quality of source data, although note in one sentence these authors disparage "raw observational data" but a few sentences later assure us - erroneously, as their own tables prove - that they have only used highest quality "Level 3" data. 'Level 3' primarily refers to satellite data and misses a large discussion (see matrix often reported by Bates) about maturity and quality of satellite data records. Their citation here covered a very small subset of ocean colour data and is now nearly 20 years of out date.

GMED includes many layers which are derived from satellite observations in addition to Ocean Color data that's why we mentioned the Level 3 processing. We will include the discussion about

Page 9, lines 275,276: "verification data indicates that the GMED layers are reliable representations of the source data". More true than the authors understand! Use unreliable source data, apply proprietary statistical interpolation tools, then "validate" by comparison back to the same deficient source data? Garbage in, leads to same garbage reliably represented in the end products. Do the authors not realise this inevitable consequence? Did they not expose their work to competent oceanographers and modellers?

GMED has been used in a range of studies and thus exposed to the oceanographic community e.g., Sayre et al. 2018, Jayathilake et al. 2018, Asad et al. 2017, Basher et al. 2016, Saeedi et al. 2016,

Page 10, line 287: Unfortunately, readers find no evidence that these authors understand the "difference between a pure statistical and a more mechanistic expert-driven approach in interpolation". These authors have certainly not shown any indication that they recognise a need for "expert-driven" guidance or advice before embarking on a "pure" but in fact sadly deficient and essentially useless statistical approach.

During preparation of GMED we did consult with experts who manage the data sources (Data managers of KGS, BioOracle, Aquamps). Rational behind our selected interpolation method was explained in lines 142-152.

Page 10, line 293. Have the authors provided a useful improvement of the land masks? After reading the dismal state of ocean data they used as sources, one wishes for some tangible improvements, e.g. of land masks. As usual, however, authors provide minimal evidence and no citations. Overall, a completely unsatisfying presentation of a basically flawed (one hesitates to say 'incompetent') effort.

We created a new high resolution land mask (as mentioned in Line 133 and 134) which was extracted from GEBCO08 bathymetry dataset. Although the details about that process is not included in the MS, if needed we will include it in the revised version of the Manuscript.

To confirm my impressions about the dismal state of GMED source data, I made a quick scan of ocean data sets available from ESSD. (Authors could do - and arguably should have done - a more careful systematic but fundamentally similar scan.) As these authors apparently recognise (because they attempted to submit their own product) these ESSD-published sources provide up-to-date, well documented, permanently identified, easy access data in standard formats, known and used in the ocean community. My quick scan exposed data sets that cover 60 to 80% of the GMED parameters. These ESSD papers also include many references to other available data. Why the authors did not at least check their AquaMAP and KGS sources against these recent openly-accessible data remains a mystery.

Thank you for the link to other datasets. We will incorporate suitable ones in our next revision of GMED dataset. Others are excellent at regional scales but have significant spatial gaps in the global coverage GMED seeks. Meanwhile we will cite them in the paper so users are aware of alternative sources.

Sources (*with my comments*):
The MAREDAT Special Issue
(*especially*) doi:10.5194/essd-5-109-2013, 25.3.2013, The MAREDAT global database of high performance liquid chromatography marine pigment measurements (*much better than any chl a product you reference*)

SOCAT (*nearly 15 million data points, 1957 to 2014, the definitive way to compile, grid and quality control ocean data*)
doi:10.5194/essd-5-125-2013, A Uniform, Quality Controlled Surface Ocean CO2 Atlas (SOCAT), doi:10.5194/essd-5-145-2013, Surface Ocean CO2 Atlas (SOCAT) Gridded Data Products doi:10.5194/essd-8-383-2016, A multi-decade record of high-quality fCO2 data in version 3 of the Surface Ocean CO2 Atlas (SOCAT)

doi:10.5194/essd-5-295-2013, 12.8.2013, Global database of surface ocean particulate organic carbon export fluxes diagnosed from the 234Th technique (*better POC than anything you have*)

doi:10.5194/essd-7-261-2015, 5.10.2015, Vertical distribution of chlorophyll a concentration and phytoplankton community composition from in situ fluorescence profiles: a first database for the global ocean (*instructive about challenges of compiling global ocean data*)

doi:10.5194/essd-8-15-2016, 1.2.2016, A gridded data set of upper-ocean hydrographic properties in the Weddell Gyre obtained by objective mapping of Argo float measurements *(example of the richness and processing of Argo data)*

doi:10.5194/essd-8-165-2016, 28.4.2016, A long-term record of blended satellite and in situ sea- surface temperature for climate monitoring, modeling and environmental studies *(arguably now the definitive SST data set)*

doi:10.5194/essd-8-235-2016, 3.6.2016, A compilation of global bio-optical in situ data for ocean- colour satellite applications *(interesting quality control, covers all your bio-optical parameters)*

GLODAP *(the definitive ocean data set, with extensive well-documented quality control)* doi:10.5194/essd-8-297-2016, 15.8.2016, The Global Ocean Data Analysis Project version 2 (GLODAPv2) – an internally consistent data product for the world ocean
doi:10.5194/essd-8-325-2016, 15.8.2016, A new global interior ocean mapped climatology: the 1°×
1° GLODAP version 2

doi:10.5194/essd-8-531-2016, 20.10.2016, Global ocean particulate organic carbon flux merged with satellite parameters *(for ESA CCI, much-used and well-documented)*

doi:10.5194/essd-8-679-2016, 29.11.2016, C-GLORSv5: an improved multipurpose global ocean eddy-permitting physical reanalysis *(interesting re-analysis product, instructive on how they assimilate sparse data)*

https://doi.org/10.5194/essd-10-251-2018, 6.2.2018, Photosynthesis–irradiance parameters of marine phytoplankton: synthesis of a global data set *(another good data assembly example, highly relevant to your intended uses)*

https://doi.org/10.5194/essd-5-311-2013, A long-term and reproducible passive microwave sea ice concentration data record for climate studies and monitoring *(sea ice data at the highest quality level)*

Cited References

Sayre, R., Noble, S., Hamann, S., Smith, R., Wright, D., Breyer, S., Butler, K., Van Graafeiland, K., Frye, C., Karagulle, D. and Hopkins, D., (2018). A new 30 meter resolution global shoreline vector and associated global islands database for the development of standardized ecological coastal units. Journal of Operational Oceanography, pp.1-10.

Jayathilake, D. R. M., & Costello, M. J. (2018). A modelled global distribution of the seagrass biome. Biological Conservation, 226, 120–126. doi:10.1016/j.biocon.2018.07.009

Saeedi, H., Dennis, T.E., Costello, M.J., 2016. Bimodal latitudinal species richness and high endemicity of razor clams (Mollusca). J. Biogeogr. 44, 592–604. https://doi.org/10.1111/jbi.12903.

Asaad, I., Lundquist, C. J., Erdmann, M. V., & Costello, M. J. (2017). Ecological criteria to identify areas for biodiversity conservation. Biological Conservation, 213, 309-316.

Basher, Z., & Costello, M. J. (2016). The past, present and future distribution of a deep-sea shrimp in the Southern Ocean. PeerJ, 4, e1713.

Weatherdon LV, Fletcher R, Jones MC, Kaschner K, Sullivan E, Tittensor DP, Mcowen C, Geffert JL, van Bochove JW, Thomas H, Blyth S, Ravillious C, Tolley M, Stanwell-Smith D, Fletcher S, Martin CS (2015). Manual of marine and coastal datasets of biodiversity importance. December 2015 edition. Cambridge (UK): UNEP World Conservation Monitoring Centre. 30 pp.

---

## Author Comment (AC2) · 3 Jan 2019

**Response to Reviewer 2 comments**

**Comments for Basher et al. (2018) - ESSD Discussion**

The present manuscript aims to present a « novel » digital atlas of environmental (meaning physical, chemical, biogeochemical) climatologies, from which scientists may download numerous environmental layers that are typically used for developing spatial statistical models, such as species distribution models (SDMs). The authors did a fine job in compiling many published datasets, and gathering all of them in a homogeneous and central atlas. Consequently, the Global Marine Environment Dataset (GMED) is the online platform with the widest range of environmental layers. The GMED proposes environmental data at a finer spatial resolution compared to previous comparable atlases (mainly AQUAMAPS, MARSPEC and Bio-ORACLE). The GMED also supplies past and future fields for some of the environmental layers (temperature, salinity, ice cover), thus allowing the community to quickly test « long-term » changes in species distributions and diversity. Consequently, it might attract marine ecologists aiming to easily model the niches and distributions of marine taxa, whether those are benthic, pelagic, coastal or inhabiting offshore conditions. But that might also be an issue as I will develop below.

In spite of the added value of the dataset might present, I have identified some major points that may help improve the completeness, the quality of the atlas and the manuscript. I will now give my step-by-step review of the manuscript and data access based on the ESSD review guidelines. Then I will detail my major concerns and comments regarding the GMED itself.

We thank the reviewer for the positive and important comments. We appreciate the inputs on how to improve the overall quality of the manuscript/ data repository with additional datasets by the reviewer. We will implement some of these steps immediately and some in near future to improve the overall quality of the GMED data layers.

1. The data presented consist of a compilation of pre-existing datasets so the data themselves are not « new », but the atlas is clearly more exhaustive than previous and comparable ones, even though some of the data used are clearly outdated (but see major comments below). Also, the data presented here have been interpolated to follow a higher resolution grid, so they represent an improvement for end users (SDM users). I do like that the authors added variables such as distance to land, or to closest port, because these often have to be calculated separately and can be very useful to account for sampling biases in marine species distributions. Therefore, I agree that this dataset could be useful for future studies. Although I consider the methods description to be thorough, I do miss proper uncertainty estimates in the layers provided online, and especially for the past and future environmental layers (but see major comments below). For

controlling the quality of the data, the authors completely rely on the controls undergone y other authors for the first publication of the data compiled. Furthermore, the perform some sort of completely circular cross- validation to control the output of their interpolation. Proper quality control would at least require some independent data. Therefore, they do not really perform « quality control » in my humble opinion. As a result, key information are missing for the reader about the way the environmental layers were developed initially. Here, it is not sufficient to simply state that « *All of the primary datasets used in the GMED compilation had undergone quality control checks by the primary data collectors and processors* ».

Thank you for the understanding and comments about the value for the compilation. The principal objective of GMED is to make marine datasets readily available to ecologists for their modeling work, to reduce the lengthy process of data compilation and standardization. We understand the issue with circulation validation, we would very much like to provide an uncertainty estimate to end users. However, as no other extensive independent global dataset was available at the time when GMED was created, for that reason we only provided a validation method to ensure users we did not introduce any new errors to the new dataset with our interpolation process. We will use the independent data layers you mentioned and by another reviewer to create uncertainty estimates for all GMED data layers in future.

2. I have identified some gaps in the niche modeling literature (lines 53-59) that I would like the authors to address carefully because I think it may have lead them to forget important predictors in their atlas (but see major comments below). I would also like the authors to mention the update of the Bio-ORACLE v2 dataset (Assis et al., 2018 - DOI: 10.1111/geb.12693) in their manuscript. Yet I acknowledge it might have been published after the authors finished their atlas.

Thank you for mentioning about the new BioOracle dataset. We are aware of the recent this recent update, and will add the citation in the revised manuscript.

3. The dataset is easily accessible via the online portal. I had no problem downloading, unarchiving and then reading the data with R. The files are encoded in ASCII, which is easy to read with the « *raster* » R package (Hijmans (2017). *raster: Geographic Data Analysis and Modeling*. R package version 2.6-7. https://CRAN.R-project.org/ package=raster). I honestly have no experience with reading ASCII tables with other commonly-used languages, such as Matlab or python, but I am convinced the people concerned will not have too much trouble with that. The online dataset seems complete regarding to what is described in the manuscript. However, I am missing error estimates and/or quality flags in the data tables provided. For now, the only « quality flag » consists in the existence of a cropped$_8$ version of the layers (at 70°N and S

because of the satellite data). I would like to know whether it would be possible to add uncertainty estimates (linked to initial observations density biases, or model uncertainties when models are part of the process) to the data tables so one can identify where the less reliable data are geographically located (especially for biogeochemical and future fields)? Nevertheless, I would say the data cleaning, treatment and comparison to previous similar datasets are adequate: what the authors did is clear and well described. One of the authors' main claims is that the finer resolution of their layers should lead to more reliable SDMs compared to previous products. Although I agree this should be the case, a formal test of this assertion is needed (like developing a few standard SDMs for a virtual species based from the present data and then compare them to SDMs built from the previous atlases). But I am not sure this is within the scope of an ESSD paper, which focuses on the data itself.

Thank you for the complement. Several studies have conducted SDM with empirical data and using GMED and we will now cite these in the MS, namely: Jayathilake et al. 2018, Asad et al. 2017, Basher et al. 2016, Saeedi et al. 2016

4. Overall, I find the dataset to be usable in its current format. Maybe others would prefer to be able to download it as a text file (.txt or .csv). The metadata are provided in the appendices and can easily be found online (http://gmed.auckland.ac.nz/layersd.html). The issue when compiling pre-existing datasets is that one may simply refer to the original publication of the data for the full metadata. These are not provided in the present manuscript but the authors do refer to the original publications and website (like in any other publication of this nature, see Tyberghein et al., 2012). Overall, the language and the figures are of good-quality in my opinion. However, I strongly recommend that the authors provide a quantitative scale and the appropriate units with the maps, instead of the rather arbitrary « low » and « high ». I think there is a typo in the caption of Figure A53: *Temperature 1AB Scenario*? This really looks like a salinity map. Also, in Figure 1, the authors need to clearly indicate which steps they performed themselves. Indeed, they did not compile all the satellite/ model/ Therefore, from what I understood, the data processing steps actually performed by the authors are those indicated by the 4th and 5th arrows (after « Raster Grid »). To summarize: the authors interpolate the older layers on a new and finer grid, and then evaluate the interpolation's output by computing variation coefficient and standard error between those and the initial layers. By doing so, the authors do not claim to actually control the quality of the data, but rather the « *interpolation quality* ». I think the authors are right in stating so, but I do find the process a bit circular…

To conclude, I do think the data presented here are complete and could be useful to quickly run and test some SDMs. It does comprise a very comprehensive compilation of different environmental variables that are commonly used as

predictors in species distribution modeling. However, I cannot conclude that the data presented here are « unique ».

Thank you for pointing out about the misplaced figure A53. Correct figure of Temperature will be included in the revised manuscript. We would like to iterate we performed all the steps in Figure 1, from Raster Grid onwards. Data was sourced from both existing data layers as well as satellite/modelled data from other sources (i.e., two of the temperature and salinity data layers were sourced from NASA PSD reanalysis datasets via Giovanni interface). We would very much like to have a data quality check step added to the current process but as mentioned earlier due to the lack of comparable global dataset we had to settle with the interpolation quality assurance only.

**Major Comments**

**1) Mixed-layer depth and variables averaged over the mixed-layer?**

In the introduction (l. 53-59), the authors rightfully state that SDMs have been relatively less used for studying marine taxa compared to their terrestrial counterparts. Then, they mention the marine groups that have been studied through SDMs with the associated literature. Here they fail to mention the recent (and less recent) studies that performed niche modeling for the marine plankton (both phytoplankton and zooplankton), apart from Bentlage et al. (2013) whom quickly performed SDMs using climatologies from before 2005…Haphazardly, you should mention some of the following studies:

Beaugrand, G. & Helaouët, P. (2008) Simple procedures to assess and compare the ecological niche of species. *Marine Ecology Progress Series*, **363**, 29-37.

Beaugrand, G., Edwards, M., Brander, K., Luczak, C. & Ibanez, F. (2008) Causes and projections of abrupt climate-driven ecosystem shifts in the North Atlantic. *Ecology Letters*, **11**, 1157-1168.

Beaugrand, G., Lenoir, S., Ibañez, F. & Manté, C. (2011) A new model to assess the probability of occurrence of a species based on presence-only data. *Mar. Ecol. Prog. Ser*, **424**, 175-190.

Reygondeau, G. & Beaugrand, G. (2011) Future climate-driven shifts in distribution of *Calanus finmarchicus*. *Global Change Biology*, **17**, 756-766.

Irwin, A.J., Nelles, A.M. & Finkel, Z.V. (2012) Phytoplankton niches estimated from field data. *Limnology and Oceanography*, **57**, 787-797.

Beaugrand, G., Mackas, D. & Goberville, E. (2013) Applying the concept of the ecological niche and a macroecological approach to understand how climate influences zooplankton: advantages, assumptions, limitations and requirements. *Progress in*

      *Oceanography*, **111**,
      75-90.

Chust, G., Castellani, C., Licandro, P., Ibaibarriaga, L., Sagarminaga, Y. & Irigoien, X. (2014) Are *Calanus* spp. shifting poleward in the North Atlantic? A habitat modelling approach. *ICES Journal of Marine Science: Journal du Conseil*, **71**, 241-253.

Pinkernell, S. & Beszteri, B. (2014) Potential effects of climate change on the distribution range of the main silicate sinker of the Southern Ocean. *Ecology and Evolution*, **4**, 3147-3161.

Villarino, E., Chust, G., Licandro, P., Butenschön, M., Ibaibarriaga, L., Kreus, M., Larrañaga, A.
      & Irigoien, X. (2015) Modelling the future biogeography of North Atlantic zooplankton communities in response to climate change. *Marine Ecology Progress Series*, **531**,
      121-142.

Brun, P., Vogt, M., Payne, M.R., Gruber, N., O'Brien, C.J., Buitenhuis, E.T., Le Quéré, C., Leblanc, K. & Luo, Y.W. (2015) Ecological niches of open ocean phytoplankton taxa. *Limnology and Oceanography*, **60**, 1020-1038.

Barton, A.D., Irwin, A.J., Finkel, Z.V. & Stock, C.A. (2016) Anthropogenic climate change drives shift and shuffle in North Atlantic phytoplankton communities. *Proceedings of the National Academy of Sciences*, **113**, 2964-2969.

Brun, P., Kiørboe, T., Licandro, P. & Payne, M.R. (2016) The predictive skill of species distribution models for plankton in a changing climate. *Global Change Biology*, **22**,
      3170-3181.

Benedetti, F., Vogt, M., Righetti, D., Guilhaumon, F. & Ayata, S.-D. (2018) Do functional groups of planktonic copepods differ in their ecological niches? *Journal of Biogeography*, **45**,
      604-616.

But more importantly: in several of these publications, SDMs were developed using mixed-layer depth (MLD) as a predictor, or other variables (PAR, SST, Chlorophyll concentration) integrated over the mixed layer. The noteworthy paper of Brun et al. (2015) even identified MLD as the most important variable for modeling the niches of phytoplankton species. MLD greatly contributes to the temperature, light conditions and nutrients dynamics perceived by the plankton, the basis of evert marine food-web. Its role in controlling Ocean-Amosphere heat fluxes and in shaping bloom dynamics has been studied for decades now. It should always be considered as a potential predictor even for fishes and/or top predators because of its probable effect through bottom-up processes. Overall, MLD is arguably one of the most important oceanographic variable so I was extremely surprised not to see it among the variables compiled. Why is that?

I highly recommend that the authors add at least one MLD product to their atlas. The most recent one I can think of would be: Holte, J., Talley, L.D., Gilson, J. & Roemmich, D. (2017) An Argo mixed layer climatology and database. *Geophysical*

*Research Letters*, **44**, 5618-5626. Which can be found here: http://mixedlayer.ucsd.edu/

I also encourage the authors to compute mixed-layer averages for several other variables such as temperature, irradiance, salinity, nutrients concentrations, Chlorophyll-a concentration etc. It seems like the authors do not benefit from the recent wave of Argo floats data. Which brings me to my second major point.

Thank you for the suggestions about SDM references and MLD dataset. We consider GMED as a living repository which will be updated with data layers as they became available over time. As suggested, we will add the references to the revised manuscript and add MLD product in next update of GMED dataset.

**2) Outdated data sources.**

While reviewing the sources of the data compiled, I was surprised that many of the layers still rely on data from the World Ocean Atlas of 2009, or from the Sea-WiFS satellite era. Since then, the World Ocean Atlas has undergone not one but two updates (it is currently at the WOA 2013v2 stage: https://www.nodc.noaa.gov/OC5/woa13/) and the MODIS- Aqua sensor has been operational since 2002. The WOA 2013v2 provides monthly/ seasonal/ annual climatologies at a 5°, 1° and sometimes 1/4° resolution, with standard depth levels, and with detailed and proper quality controls. I am very surprised the authors did not take the time to assimilate these layers since they are widely known in the oceanographic community.

For other chemical and biogeochemical variables, way more recent and valuable datasets can be found in ESSD:

https://www.earth-syst-sci-data.net/7/261/2015/essd-7-261-2015.pdf https://www.earth-syst-sci-data.net/8/325/2016/essd-8-325-2016.pdf
https://www.earth-syst-sci-data.net/8/297/2016/essd-8-297-2016.pdf https://www.earth-syst-sci-data.net/8/383/2016/essd-8-383-2016.pdf
And, of course, updated and controlled observations and re-analyses can be found on the Copernicus data portal: http://marine.copernicus.eu/services-portfolio/access-to- products/

I am a bit uncomfortable as I do not want to dismiss all of the work carried out by the authors, but I must *strongly* encourage them to go through all these data products and update their data sources. Otherwise the community is just given recycled and outdated data products that do not not reflect the state of the art, nor the efforts of the climate and ocean scientists. This point is also valid for the past and future environmental layers provided in the GMED, which brings me to my third major comment.

Thank you for the suggestions. We are aware of the WOA 2013 and even the most

recent WOA 2018. As GMED was initially created in 2013, released on 2014, we did not have opportunity to incorporate these recent data layers into the repository. We will be incorporating all these new data layers progressively in future updates of the repository, and most importantly, noting to users that new versions of the primary data are available which when processed would improve the quality of the maps..

**3) Fields of future environmental conditions.**

One of the reasons why SDMs got so popular in the last 20 years is because they allow to handily explore temporal changes in species distribution, and therefore diversity, following climate change (greenhouse gas emissions actually) scenarios. Knowing that, the authors added some predictions of SST, SSS, seabed temperature and salinity, primary productivity and ice concentration. This could have been interesting had the scenarios not been completely outdated. Indeed, the data compiled here were issued for the 4th AR of the IPCC (CMIP3 exercice). I do not believe the authors are unaware of the existence of the IPCC's 5th AR which presents Representative Concentration Pathways (RCPs) that are now the standard when it comes to model climate change impacts. I know, from my own experience, that RCPs data are not always available for regional models, but this is definitely not the case for the global ocean. Proof is that even the latest version of Bio- ORACLE (Assis et al., 2018 - DOI: 10.1111/geb.12693) provides RCPs outputs, with some uncertainty estimate across AOGCMs. Why did the authors not consider the latest standards?

The authors fail to provide crucial information about model set-up, calibration, configuration, validation, bias correction…The two links provided in the references below Table 1 are not functional. Do the layers presented correspond to the absolute fields obtained for the 2090-2100 period? Or to model biases between the contemporary period and the end-of-century period that were added to the observation-based climatologies? What are the uncertainties within each projection? And then between projections? Why did the authors rely on just two models (IPSL and HadCM3) among all the existing ones? Why are future surface temperature and salinity given for two emission scenarios but not seabed temperature? What is the model configuration that generated the future PP product? There are *tremendous* uncertainties across the suite of coupled ecosystem models that can provide biogeochemical projections (just have a look at Laufkötter et al.,
2015 - doi:10.5194/bg-12-6955-2015), and this is well known in the community. I am sorry but these future layers cannot be used as of now. The choice of the climate model can make up a significant part of the uncertainties in SDM-based climate change predictions. Please see:

Diniz-Filho, J.A.F., Bini, L.M., Rangel, T.F.L., Loyola, R.D., Hof, C., Nogués-Bravo, D. & Araújo, M.B. (2009) Partitioning and mapping uncertainties in ensembles of forecasts of species turnover under climate change. *Ecography*, **32**, 897-906.

Buisson, L., Thuiller, W., Casajus, N., Lek, S. & Grenouillet, G. (2010) Uncertainty in ensemble forecasting of species distribution. *Global Change Biology*, **16**, 1145-1157.

Garcia, R.A., Burgess, N.D., Cabeza, M., Rahbek, C. & Araújo, M.B. (2012) Exploring consensus in 21st century projections of climatically suitable areas for African vertebrates. *Global Change Biology*, **18**, 1253-1269.

The risk is that young scientists might implement SDMs based on the contemporary layers provided the GMED, with the default settings user-friendly modeling platforms, and simply project those in the future conditions, without any prior knowledge about the way the data were produced…Knowing the data you use (meaning understanding where it comes from, its limitations and quality, the uncertainties associated) is a crucial part of any modeling experiment, and species distribution modeling is not an exception. And this brings me to my fourth and final major comment.

Thank you for pointing out about the invalid link below Table 1. We will update the correct URL in the revised manuscript. Regarding the quality assurance for projected data layers, all the future data layers were originally published either by Bio-Oracle v1 or AquaMaps (mentioned in the appendix and http://gmed.auckland.ac.nz/layersd.html webpage), and we compiled the LGM data layer ourselves from the primary data source. As discussed in the manuscript all details about the model setting, calibration, and other information should be available from the meta-data from AquaMaps and Bio-Oracle website. We will include these references and link to metadata next to the future data layers in the revised version of the Manuscript. We will also include a warning to the website stating the some maps will be updated using more recent and improved primary data sources (i.e., new RCP's for future data layers).

**4) Compilation of environmental predictors takes time…and maybe it should do so.**

The manuscript's abstract stipulates the following: « *Marine environmental datasets available for species distribution modelling (SDM) have different spatial resolutions and are frequently provided in assorted file formats. This makes data assembly one of the most time-consuming parts of any study using multiple environmental layers for biogeography visualization or SDM applications* ». I assume this motivated the authors (but others also) to implement user-friendly and publicly available compilations of environmental data to facilitate (and accelerate) the process. This could make sense when the quality of the data used and therefore the whole procedure is not affected. But I am confident this cannot be the case when using the present GMED (because of all the previously mentioned reasons).

Instead, I argue it is crucial that students and young scientists take the time that is required to: (i) review the environmental datasets available; (ii) thoroughly examine their origins (metadata), advantages and limitations; and (iii) investigate how alternative choices in the environmental data impact final SDMs outputs. How are they supposed to perform state of the art modeling if they do not even understand the ins and outs of the data they use? Data assembly is time-consuming because it

is the process that will determine data quality and thus the quality of any SDM projection. The identity and resolution of the environmental predictors available and suitable for a niche modeling exercice depend on its goals (testing for niche overlap, species distribution visualization, climate change impacts projections), and the type of biological data available (abundance, presence only, presence-absence etc.). I totally get that our community is experiencing increasing pressure because of competition for fundings, pressure to publish, and demands from stakeholders to provide climate change predictions, and therefore tries to gain time when possible. But simplicity and easy-to-use products should not take over the quality that any scientific experiment is entitled to.

To conclude, I would like the authors to know that I am truly sorry that I could not provide more positive comments. I hope they will take it as an encouragement to deeply re- organize and actualize their data so they comply with the quality required for any ESSD dataset. I encourage them to work more closely with oceanographers and climate scientists to help them find better and updated marine environment data.

We completely agree with the comment and objective of making quality data available for rapid SDM applications. With GMED we want to approach towards that direction. As suggested we will work to include some validation statistics as suggested by the reviewer and incorporate inputs from oceanographers and climate scientists whenever possible to improve the overall quality of the distributed data layers over time. We will also include warnings where appropriate in the website to notify users about datasets which might be available with newer model outputs (i.e., IPCC AR5 RCP's) from primary data providers.

Dr. Fabio Benedetti
ETH Ziirich, 0-USYS, IBP, UP Group.
On the 03/10/2018.

Cited References

Jayathilake, D. R. M., & Costello, M. J. (2018). A modelled global distribution of the seagrass biome. Biological Conservation, 226, 120–126. doi:10.1016/j.biocon.2018.07.009

Saeedi, H., Dennis, T.E., Costello, M.J., 2016. Bimodal latitudinal species richness and high endemicity of razor clams (Mollusca). J. Biogeogr. 44, 592–604. https://doi.org/10.1111/jbi.12903.

Asaad, I., Lundquist, C. J., Erdmann, M. V., & Costello, M. J. (2017). Ecological criteria to identify areas for biodiversity conservation. Biological Conservation, 213, 309-316.

Basher, Z., & Costello, M. J. (2016). The past, present and future distribution of a deep-sea shrimp in the Southern Ocean. PeerJ, 4, e1713.